# Why Adversarially Train Diffusion Models?

**Maria Rosaria Briglia**[†]    **Mujtaba Hussain Mirza**[†]    **Giuseppe Lisanti**[⋆]    **Iacopo Masi**[†]

Sapienza University of Rome [†]      OmnAI Lab[†]      University of Bologna [⋆]

## Abstract

Adversarial Training (AT) is a known, powerful, well-established technique for improving classifier robustness to input perturbations, yet its applicability beyond discriminative settings remains limited. Motivated by the widespread use of score-based generative models and their need to operate robustly under substantial noisy or corrupted input data, we propose an adaptation of AT for these models, providing a thorough empirical assessment. We introduce a principled formulation of AT for Diffusion Models (DMs) that replaces the conventional *invariance* objective with an *equivariance* constraint aligned to the denoising dynamics of score matching. Our method integrates seamlessly into diffusion training by adding either random perturbations–similar to randomized smoothing–or adversarial ones–akin to AT. Our approach offers several advantages: **(a)** tolerance to heavy noise and corruption, **(b)** reduced memorization, **(c)** robustness to outliers and extreme data variability and **(d)** resilience to iterative adversarial attacks. We validate these claims on proof-of-concept low- and high-dimensional datasets with *known* ground-truth distributions, enabling precise error analysis. We further evaluate on standard benchmarks (CIFAR-10, CelebA, and LSUN Bedroom), where our approach shows improved robustness and preserved sample fidelity under severe noise, data corruption, and adversarial evaluation. Code available at github.com/OmnAI-Lab/Adversarial-Training-DM

## 1 Introduction

Large-scale datasets are cornerstones to the success of generative AI, yet they simultaneously present a significant challenge. Often web-scraped and minimally curated, they frequently contain multiple forms of corruption: *inlier noise*–subtle perturbations within samples; *outlier noise*–samples that significantly deviate from the target distribution; *missing or corrupted data*–commonly affected by Gaussian noise; and *adversarial noise*–deliberately crafted perturbations. While recent approaches have attempted to address training under noisy conditions, they are constrained by restrictive theoretical assumptions. For instance, Daras et al. (2024c) relies on precise knowledge of noise variance, Daras et al. (2024d) exclusively targets missing data scenarios, and Daras et al. (2024a) presupposes access to both clean and corrupted samples. These approaches fall under the umbrella of "noise-aware training", solving a problem with strong assumptions: the methods assume access to clean/noisy sample labels at a sample level, assuming to know the applied noise distribution and its intensity, fully exploiting these assumptions at training time. This controlled scenario greatly limits the practical applicability of the proposed methods, as also specified by the authors in the limitation section of their work. The present work aims to define the principles of robust training for Diffusion Models, highlighting how this technique enables generative models to be robust against unknown corruption, whether applied to the training data or as inference-time perturbations, such as adversarial attacks. In the classification domain, Adversarial Training (AT) (Szegedy et al., 2014) yields robust classifiers that maintain performance despite input perturbations. Notably, recent research has revealed that AT confers additional capabilities beyond robustness, including generative capabilities (Mujtaba Hussain et al., 2024). Despite its demonstrated efficacy in classification tasks, AT has not been systematically extended to other families of deep learning models, particularly generative models. Beyond handling noisy data, generative AI models face further challenges, including data memorization (Jagielski et al., 2023; Somepalli et al., 2023; Carlini et al., 2023b), leading to information leakage, and their propensity to learn spurious correlations from training data that do not reflect real underlying patterns. In this work, we address this challenge by extending AT to score-based generative models, in

Figure 1: **Smooth trajectories.** We train the denoising network to follow the score function *i.e.,* $\mathbf{x}_t \mapsto \mathbf{x}_{t-1}$ using just $\epsilon_\theta(\mathbf{x}_t, t)$, but we also perturb locally $\mathbf{x}_t$ as $\mathbf{x}_t + \boldsymbol{\delta}$ inside a $\ell_p$ ball and then imposing equivariance: $\mathbf{x}_t + \boldsymbol{\delta} \mapsto \epsilon_\theta(\mathbf{x}_t, t) + \boldsymbol{\delta} \triangleq \mathbf{x}_{t-1}$. This equals adding an intermediate step in the Markov Chain, behaving as an additional denoising step, making the model resilient to possible outliers or noise in the dataset—$p_{\text{noise}}(\mathbf{x}_0)$—not proper of $p_{\text{data}}(\mathbf{x}_0)$. The local perturbation can be implemented as adversarial or as random (randomized smoothing). Perturbation strength starts large and progressively shrinks when $T \to 0$. $\rightarrow$ indicates the forward process; $\dashrightarrow$ the reverse process.

particular to Diffusion Models (DM) (Ho et al., 2020). Our contribution bridges the gap between AT for classification and the generation paradigm. This extension also reveals new *hidden capabilities* of AT applied to the generative domain, offering a practical and theoretically sound approach to training robust generative models on real, imperfect datasets. Our contribution is then three-fold:

◇ We are the first to reconnect AT to denoising, linking it to Daras et al. (2024d;c;a). Despite some works on adversarial aspects in DM training(Yang et al.; Sauer et al.), we formally introduce adversarial training for DMs, discussing its practical implications on the learned denoising process.

◇ Inspired by Zhang et al. (2019), we develop an AT algorithm tailored for score-based models. Different from classifiers, which require enforcing *invariance*, score-based models require enforcing *equivariance* to properly learn the data distribution, as formalized in our key finding in Eq. (14).

◇ We show our method's flexibility in handling noisy data, facing extreme variability like outliers, preventing memorization, and improving robustness. Besides low-dimensional (3D) controlled data, we test our method on CIFAR-10, CelebA, LSUN and ImageNet, achieving strong performance.

## 2 ADVERSARIAL TRAINING SMOOTHS TRAJECTORIES

### 2.1 PRELIMINARIES

Diffusion Models (DMs) aim to learn a data distribution, $p_{\text{data}}(\mathbf{x})$ by noising data with a fixed procedure, mapping them to $\mathcal{N}(\mathbf{0}, \mathbf{I})$ using a Markov Chain $q(\mathbf{x}_T, \ldots, \mathbf{x}_1 | \mathbf{x}_0) = \prod_{t=1}^T q(\mathbf{x}_t | \mathbf{x}_{t-1})$, where, given a noisy input $\mathbf{x}_{t-1}$, the next state $\mathbf{x}_t$ is reached through the following gaussian transition:

$$q(\mathbf{x}_t | \mathbf{x}_{t-1}) = \mathcal{N}\big(\mathbf{x}_t; \sqrt{1 - \sigma(t)}\,\mathbf{x}_{t-1}, \sigma(t)\mathbf{I}\big), \tag{1}$$

$\sigma(t)$ is the noise scheduler: a monotonically decreasing time-varying function chosen s.t. $\sigma(0) = \sigma_{\min}$, $\sigma(T) = \sigma_{\max}$ and $0 < \sigma_{\min} < \sigma_{\max} < 1$. The generation is achieved with a learnable "decoding step" that reverts data from noise estimation $p(\mathbf{x}_{t-1} | \mathbf{x}_t)$. If the noise scheduler is chosen carefully to take small noising steps, then the approximation $q(\mathbf{x}_T | \mathbf{x}_0) \approx \mathcal{N}(\mathbf{0}, \mathbf{I})$ and the following equation holds:

$$q(\mathbf{x}_t | \mathbf{x}_0) = \mathcal{N}\big(\mathbf{x}_t; \sqrt{\alpha_t}\mathbf{x}_{t-1}, (1 - \alpha_t)\mathbf{I}\big) \quad \text{where} \quad \alpha_t \doteq \prod_{s=1}^t 1 - \sigma(t)$$

This means we can encode directly from $\mathbf{x}_0 \mapsto \mathbf{x}_t$ as:

$$\mathbf{x}_t = \sqrt{\alpha_t}\mathbf{x}_0 + \sqrt{1 - \alpha_t}\,\epsilon \quad \text{where} \quad \epsilon \sim \mathcal{N}(\mathbf{0}, \mathbf{I}). \tag{2}$$

Samples generation is then performed by solving the probability flow ODE (PF-ODE) Song et al. (2021b), from $t = T$ to $0$ and starting from $\mathbf{x}_T \sim \mathcal{N}(0, \sigma_{\max}^2 I)$, whose solution is learned from the DM. For a given $\mathbf{x}_0$, the training objective $\mathcal{L}_{\text{DM}}$ reported in Ho et al. (2020) is thus defined as:

$$\mathcal{L}_{\text{DM}} = \mathbb{E}_{\substack{\epsilon \sim \mathcal{N}(\mathbf{0}, \mathbf{I}) \\ t \sim \mathcal{U}(\mathbf{0}, \mathbf{I})}} \left[ \left\| \epsilon - \epsilon_\theta\big(\mathbf{x}_t(\mathbf{x}_0, \epsilon), t\big) \right\|_2^2 \right] \tag{3}$$

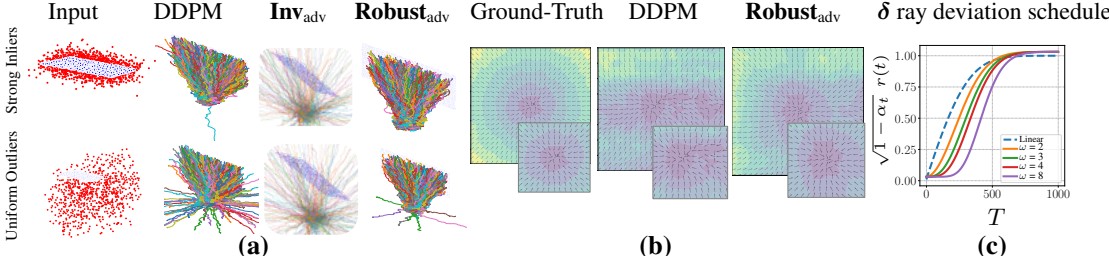

Figure 2: **(a)** The plot shows leftmost training data either with strong inlier noise *(top)* or uniform outliers *(bottom)*. The trajectories reveal that DDPM struggles with both, while if you train with invariance (**Inv**$_{\text{adv}}$) the process diverges. Instead, ours (**Robust**$_{\text{adv}}$) is more robust, avoiding diverging trajectories and better reaching the data centroid. **(b)** Score vector fields: versors represent the score field, colormap shows magnitude, ■ less ■ more intense. *(left)* Ground-truth *(middle)* DDPM; *(right)* Our **Robust**$_{\text{adv}}$. AT yields smoother, more consistent scores, better matching the data shape, shrinking variability and increasing field intensity. **(c)** Perturbation ray. The parameter $\omega$ controls the slope of $\sqrt{1 - \alpha_t} \ r(t)$ to shorten the content phase and reduce the curve's steepness in DDPM.

whose objective is to infer the noise $\epsilon$ applied to the initial image, ensuring that the starting point $\mathbf{x}_0$ is correctly reconstructed, enabling the model—the denoising network $\epsilon_\theta$—to correctly generate in-distribution data during inference. For inference we solve the SDE using $\epsilon_\theta$ and the recurrency:

$$\mathbf{x}_{t-1}(\theta) = \frac{1}{\sqrt{1 - \sigma(t)}} \left( \mathbf{x}_t(\theta) - \frac{\sigma(t)}{\sqrt{1 - \alpha_t}} \epsilon_\theta \big( \mathbf{x}_t(\theta), t \big) \right) + \sigma(t)\mathbf{z}, \ \mathbf{z} \sim \mathcal{N}(\mathbf{0}, \mathbf{I}), \forall t \in [0, \cdots, T]. \quad (4)$$

## 2.2 MOTIVATION, "IN VITRO" EXPERIMENTS, AND NOISE TYPES

**Motivation and overview.** Adversarial training has proven highly effective in the classification domain for handling perturbed training data, imposing a model invariant response across genuine and adversarially manipulated inputs. Unlike classifiers, its application to DMs requires fundamental reformulation due to their regression-based nature. Our work aims to investigate the properties and applications of adversarially trained DMs, with particular emphasis on the case of corrupted training data. Fig. 2 (a,b) describes the investigated settings of uniform outlier and strong inlier noise, where our approach was demonstrated to be learning the correct data distribution and a smoother score field.

**"In vitro" analysis setup.** We propose a first analysis of the framework on synthetic 3D data, spanning from "linear" and unimodal to more complex multi-modal ones. `oblique-plane` assumes the data distribution $p_{\text{data}}$ lives on a 2D subspace with equation $x + y + z = 30$, while `3-gaussians`, a multi-modal 3D Mixture of Gaussians defined as $\frac{1}{3}\mathcal{N}([10, 10, 10], \sigma) + \frac{1}{3}\mathcal{N}([20, 20, 20], \sigma) + \frac{1}{3}\mathcal{N}([10, 30, 30], \sigma)$ and $\sigma = 0.25$. Regarding higher-dimensional data, we built the analysis on the data generated after linearizing, using PCA (Abdi & Williams, 2010), the "Smithsonian Butterflies"[1] image dataset. After fitting a 25 dimensional subspace, retaining 70% of the sample's variance, we sampled data according to $\mathbf{x}' = \boldsymbol{\mu} + \sum_i \lambda_i \boldsymbol{\alpha}_i \mathbf{U}_i$. Sampling stochasticity comes from $\boldsymbol{\alpha} \sim \mathcal{N}(0; \sigma)$, while $\boldsymbol{\mu} \in \mathbb{R}^{3072}$, $\mathbf{U} \in \mathbb{R}^{25 \times 3072}$, and $\lambda_i$ are the mean, dataset's principal components and its singular values. Finally, we discard the real data, fitting the DM on $\{\mathbf{x}'\}_{i=1}^N$. This allows to *perfectly measure* distance between the DM generated samples and the linearized distribution, measuring the **closed-form reconstruction error** $\rho = \left\| \mathbf{x}_0(\theta) - \mathbf{U}\mathbf{U}^\top \mathbf{x}_0(\theta) \right\|$ between the data subspace and the generations, where $\mathbf{x}_0(\theta)$ is generated iterating on Eq. (4). We also use the measure Peak Signal-to-Noise Ratio (PSNR) from image processing (Hore & Ziou, 2010).

**Noise model tested.** We analyze the framework on different noise models. First, we consider inlier noise, implemented by increasing the sampling variance $\sigma$ or, in the case of subspace $\boldsymbol{\mu} + \sum_i \lambda_i \boldsymbol{\alpha}_i \mathbf{U}_i$, increasing the $\boldsymbol{\alpha}$. We then include outliers by adding strong noise in the ambient space: for 3D data, we add a point cloud with dense, grid-like, uniform noise; for `butterflies`, we add Gaussian noise on the linearized data as $\mathbf{x}' + \mathbf{z}$ where $\mathbf{z} \sim \mathcal{N}(\mathbf{0}, \sigma\mathbf{I})$. Figs. 2 and 3 show the proposed ablations.

---

[1] huggingface.co/datasets/huggan/smithsonian_butterflies_subset

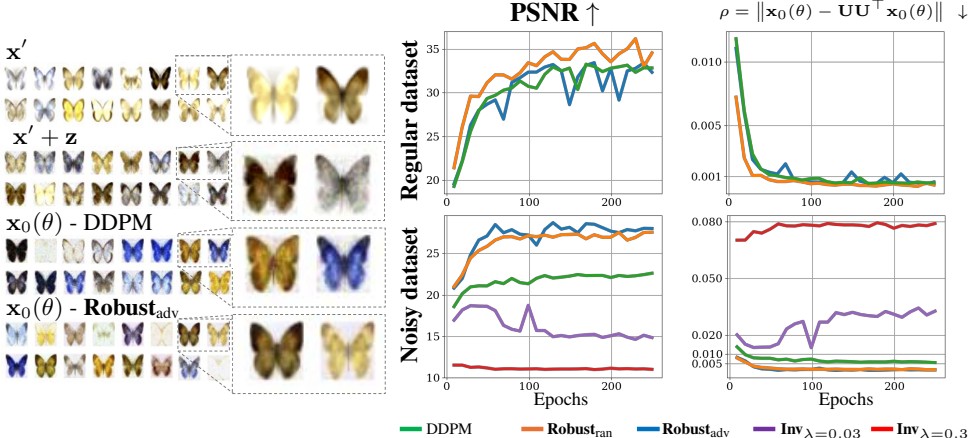

Figure 3: (*left*) On `butterflies`, we report the closed-form reconstruction error. From top to bottom: training data, corrupted data, DDPM-generated samples, and **Robust**$_{adv}$ results. (*right*) The chart columns display PSNR and closed-form reconstruction error measured on clean data (*top*) and on data corrupted at 90% with Gaussian noise ($\sigma = 0.1$) (*bottom*). We also include results for *invariance regularization* with $\lambda = \{0.3, 0.03\}$; these settings prevent the model from properly $p_{data}$.

## 2.3 ADVERSARIAL TRAINING FOR DIFFUSION MODELS

Diffusion processes rely on the denoising function mapping the noisy distribution $q_t$ to the data distribution $p_{data}$, learned from optimizing $\mathcal{L}_{DM}$ (see Eq. (3)), which ensures the model to learn the score field correctly, guiding the trajectories toward the data distribution. When defining the AT procedure, applying the standard AT (Szegedy et al., 2014) could hinder the learning process. We, indeed, propose an AT technique that inherits its principles from TRADES (Zhang et al., 2019), and accordingly acts as a distribution-level regularization.

**Naïve invariance does not work.** As for AT, we aim to guarantee that the model maintains a constant behavior in its predictions, whether the input sample is corrupted or not. To accomplish this, standard AT imposes invariance in the classification domain; conversely, for DMs, performing a regression task, invariance does not guarantee the same result. Fig. 2 shows that applying classical AT invariance as in Eq. (5), causes DMs

$$\mathcal{L}_{inv} = \lambda \arg\min_{\theta} \left\| \boldsymbol{\epsilon}_\theta (\mathbf{x}_t + \boldsymbol{\delta}, t) - \boldsymbol{\epsilon}_\theta (\mathbf{x}_t, t) \right\|_2^2 \tag{5}$$

to learn a different distribution than $p_{data}(\mathbf{x})$, rooting the generation to produce noisy data. This finding translates also to higher-dimensional data: Fig. 3 (*right, bottom row*) shows both qualitative samples and quantitative results applying the invariance to `butterflies` noisy data (90% corrupted samples, $\sigma = 0.1$). The model, indeed, only recovers the distribution knowledge as the weight $\lambda$ decreases. When moving to real data, applying invariance resulted in worse FID, achieving 356.9 on 50K generated samples from a DM trained on the CIFAR-10 dataset.

**Key change is equivariance.** Starting from an $\boldsymbol{\epsilon}$-predicting DM, we defined the AT taking into account the need for input sensitivity of the model by enforcing *equivariance*. The intuition is depicted in the introductory Fig. 1 and a theoretical discussion is given in Section A.1. Since the main aim is to keep the model rooted to the data distribution, despite the additional perturbations $\boldsymbol{\delta}$, the network must learn to correctly recover the previous state $\mathbf{x}_{t-1}$ starting from $\mathbf{x}_t + \boldsymbol{\delta}$. This objective is reached by taking into account $\boldsymbol{\delta}$ in the AT loss as $\arg\min_{\theta} \left\| \boldsymbol{\epsilon}_\theta (\mathbf{x}_t + \boldsymbol{\delta}, t) - [\boldsymbol{\epsilon} + \boldsymbol{\delta}] \right\|_2^2$. While this equation enforces equivariance, it does not yet enforce smoothness, since two outputs of the network do not interact with each other.

---

**Algorithm 1** AT for Diffusion Models

**Input:** dataset $\mathcal{D}$, model $\theta$, max timestep $T$, scheduler $\alpha_t$, strength $\lambda$, ray scheduler $r_\beta(t)$
**repeat**
    Sample $\mathbf{x}_0 \sim \mathcal{D}$, $\boldsymbol{\epsilon} \sim \mathcal{N}(\mathbf{0}, \mathbf{I})$,
    $t \sim \mathcal{U}(\{0, \dots, T\})$, $\beta \sim \mathcal{U}[0.5, 2]$,
    $\boldsymbol{\delta} \sim \mathcal{U}[-r_\beta(t), r_\beta(t)]$
    $\mathbf{x}_t = \sqrt{\bar{\alpha}_t}\mathbf{x}_0 + \sqrt{1 - \bar{\alpha}_t}\boldsymbol{\epsilon}$
    Compute $\boldsymbol{\delta}_{adv}$ using Eq. (8)
    $\mathbf{x}_t^{adv} = \sqrt{\bar{\alpha}_t}\mathbf{x}_0 + \sqrt{1 - \bar{\alpha}_t}(\boldsymbol{\delta}_{adv} + \boldsymbol{\epsilon})$
    $\theta \leftarrow \theta - \eta \nabla_\theta \mathcal{L}_{AT}(\mathbf{x}_t, \mathbf{x}_t^{adv}, t, \boldsymbol{\epsilon})$ Eq. (6)
**until** convergence

---

**Our Training.** In this work, we propose an adversarial loss suited for $\epsilon$-predicting DMs. Given a timestep $t$ and an initial sample $\mathbf{x}_0 \sim p_{\text{data}}$, we define $\mathbf{x}_t$ as in Eq. (2), and its perturbed counterpart $\mathbf{x}_t + \boldsymbol{\delta}$. AT is then defined as a regularization of the standard DM objective. The adversarial regularization term, $\mathcal{L}_{\text{reg}}$, aims to *promote local equivariance and smoothness* along the regular DM trajectories, which is achieved by locally minimizing the difference between the model's prediction on $\mathbf{x}_t$ and $\mathbf{x}_t + \boldsymbol{\delta}$. The complete loss is given in Eq. (6), where the adversarial component is weighted by a time-dependent coefficient $\lambda_t$. AT procedure is detailed in Algorithm 1.

$$\mathcal{L}_{\text{AT}}(\mathbf{x}_t, \mathbf{x}_t + \boldsymbol{\delta}, t, \boldsymbol{\epsilon}) = \arg\min_{\theta} \underbrace{\left\| \epsilon_\theta(\mathbf{x}_t, t) - \boldsymbol{\epsilon} \right\|_2^2}_{\mathcal{L}_{\text{DM}} \text{ to fit data distr.}} + \underbrace{\lambda_t \left\| \epsilon_\theta(\mathbf{x}_t + \boldsymbol{\delta}, t) - [\epsilon_\theta(\mathbf{x}_t, t) + \boldsymbol{\delta}] \right\|_2^2}_{\mathcal{L}_{\text{reg}} \text{ to enforce smoothness}} \quad (6)$$

### 2.4 ADVERSARIAL PERTURBATION IN THE DIFFUSION PROCESS

**Injecting noise in the trajectory space.** Learning a diffusion process requires itself to corrupt natural data iteratively during its training. In this context, the adversarial perturbation can be considered as an additional noise component injected into $\mathbf{x}_t$ during the training. Therefore, defining the sample's adversarial counterpart $\mathbf{x}_t + \boldsymbol{\delta}$ so that it does not interfere with the diffusion process, requires a proper and careful tuning of the perturbation parameters, accounting for both the DM objective and the intermediate noisy data distributions $\mathbf{x}_t \sim \mathcal{N}(\mathbf{x}_{t-1}, \sigma_t I)$. To ensure compatibility with the diffusion process (Wang & Vastola, 2023; 2024), we bound the adversarial noise by a time-varying radius $r(t) = \left\| \boldsymbol{\delta}(t) \right\|_p$, dependent on $\sigma(t)$ values, to maintain model stability and avoid mode collapse due to diffusion trajectories merging. Specifically, allowing the ray to grow too large in some diffusion phases, like the content phase of generation (Choi et al., 2022), can lead to data over-smoothing, causing the model not to capture the correct distribution. We define $\mathbf{x}_t + \boldsymbol{\delta}$ as:

$$\mathbf{x}_t + \boldsymbol{\delta} = \sqrt{\alpha_t}\mathbf{x}_0 + \sqrt{1 - \alpha_t}(\boldsymbol{\epsilon} + \boldsymbol{\delta}), \text{ where } \boldsymbol{\delta} \in \left[-r_\beta(t), r_\beta(t)\right], r_\beta(t) \doteq \frac{(\sqrt{1 - \alpha_t})^\omega + \gamma \cdot \beta}{\sqrt{1 - \alpha_t}} \quad (7)$$

where $\boldsymbol{\epsilon} \sim \mathcal{N}(\mathbf{0}, \mathbf{I})$ and the exponent $\omega \geq 1$ guiding the ray scheduling, whose effect is shown in Fig. 2 (c). Finally, we propose retaining a randomized bias term $\gamma \cdot \beta$ with $\beta \sim \mathcal{U}[0.5, 2], \gamma \in \mathbb{R}^+$, whose aim is to prevent regularization annealing as $t \to 0$ and avoid data under-smoothing.

**Smoothing perturbations.** The adopted smoothing perturbation could be either random $\boldsymbol{\delta}_{\text{ran}}$, akin to randomized smoothing Cohen et al. (2019), or adversarial $\boldsymbol{\delta}_{\text{adv}}$, as in AT Goodfellow et al. (2015).

*Random:* This approach requires the perturbation $\boldsymbol{\delta}_{\text{ran}}$ to be sampled randomly in a uniform distribution, limited by $r_\beta(t)$, defined as in Eq. (7). Being the ray itself randomized through the variable $\beta$, $\boldsymbol{\delta}_{\text{ran}}$ would then be a uniform random variable whose standard deviation is $r_\beta(t)/\sqrt{3}$, proof in Section C.3.

*Adversarial:* In the adversarial setting, we employ the Fast Gradient Sign Method (FGSM) with a random start (Kurakin et al., 2017). The perturbation is first initialized as $\boldsymbol{\delta}_{\text{ran}}$, then followed by a single FGSM step. The resulting perturbation is then projected back onto the $\ell_\infty$ ball of radius $r_\beta(t)$ to ensure $\|\boldsymbol{\delta}_{\text{adv}}\|_\infty \leq r_\beta(t)$. The optimization $\boldsymbol{\delta}_{\text{adv}}$ considers the following cost function:

$$\mathcal{J}_\theta(\mathbf{x}_t, \boldsymbol{\delta}, t) = \left\| \epsilon_\theta(\mathbf{x}_t + \boldsymbol{\delta}, t) - \epsilon_\theta(\mathbf{x}_t, t) \right\|_2^2, \quad \boldsymbol{\delta}_{\text{adv}} = \mathbb{P}_{r_\beta(t)}\left[ \boldsymbol{\delta}_{\text{ran}} + \frac{r_\beta(t)}{\sqrt{3}} \mathcal{S}\big(\nabla_{\mathbf{x}_t} \mathcal{J}_\theta(\mathbf{x}_t, \boldsymbol{\delta}_{\text{ran}}, t)\big) \right] \quad (8)$$

where $\mathbb{P}_{r_\beta(t)}$ projects the adversarial perturbation onto the surface of $\mathbf{x}_t$'s neighbor $\ell_\infty$-ball , $\mathcal{S}$ is the sign operator and $r_\beta(t)/\sqrt{3}$ is the standard deviation of the attack. Once the attack magnitude is defined, we define the AT regularization strength as $\lambda_t = \frac{\lambda \cdot \sqrt{3}}{\beta \cdot r(t)}$, dependent on the perturbation norm via its standard deviation and on a global constant $\lambda \in \mathbb{R}^+$.

## 3 EXPERIMENTAL RESULTS

**Experimental setup.** We present results on datasets ranging from controlled synthetic 3D data to complex, real-world multi-modal data, presenting results "in vitro" to precisely measure errors in both low- and high-dimensional settings We further offer results on real datasets such as CIFAR-10 (Krizhevsky et al., 2009) (50K images, $32 \times 32$ pixels), CelebA (Liu et al., 2015) (202K images, $64 \times 64$ pixels), LSUN Bedroom (Yu et al., 2015) (303K images, $256 \times 256$ pixels), and ImageNet (1.28M images,

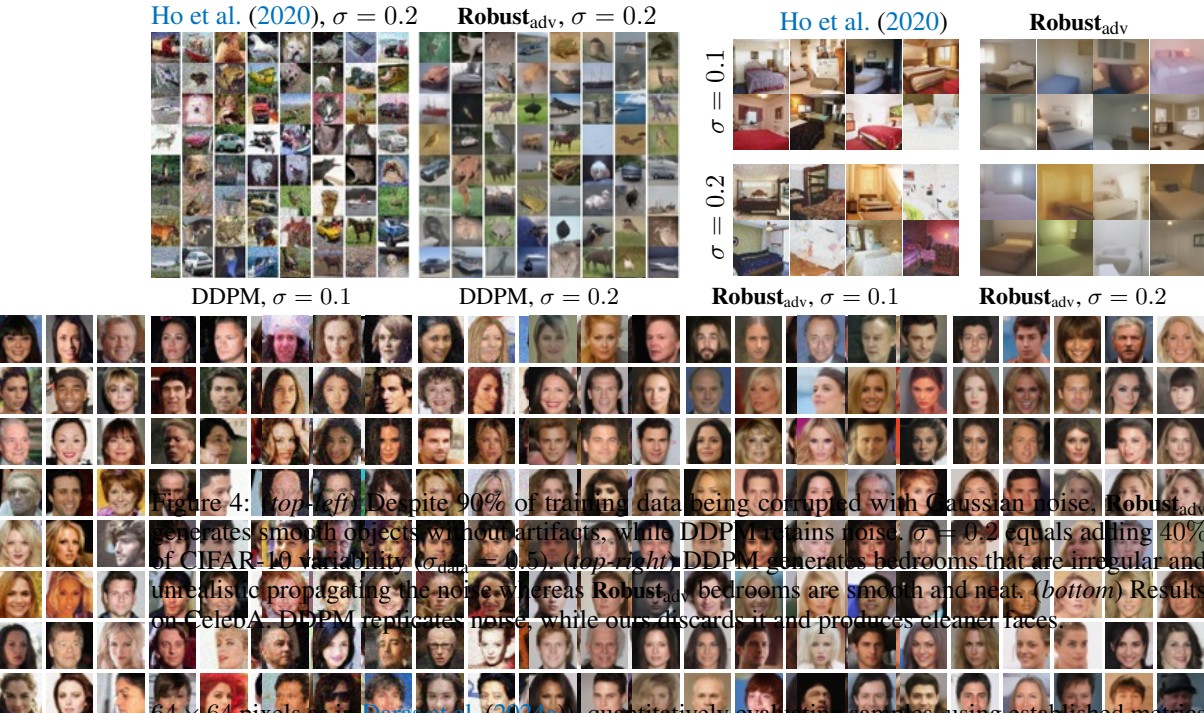

Figure 4: *(top-left)* Despite 90% of training data being corrupted with Gaussian noise, **Robust**adv generates smooth objects without artifacts, while DDPM retains noise. $\sigma = 0.2$ equals adding 40% of CIFAR-10 variability ($\sigma_{\text{data}} = 0.5$). *(top-right)* DDPM generates bedrooms that are irregular and unrealistic propagating the noise whereas **Robust**adv bedrooms are smooth and neat. *(bottom)* Results on CelebA. DDPM replicates noise, while ours discards it and produces cleaner faces.

$64 \times 64$ pixels as in Daras et al. (2024a)), quantitatively evaluating samples using established metrics such as IS (Salimans et al., 2016) and FID (Heusel et al., 2017). Following Daras et al. (2024c;a), we experiment with Gaussian noise as corruption $p_{\text{noise}}(\mathbf{x})$ and only work in challenging settings, testing a percentage $p$ of corrupted data of $p = 90\%$ with two levels of $\sigma = \{0.1, 0.2\}$. However, our method does not take into account the distinction between clean and noisy samples nor requires knowledge of the corruption variance $\sigma$. When computing FID, we always test on the *clean dataset* despite training with noisy datasets. Our methods are indicated by **Robust**adv when using adversarial perturbation and **Robust**ran if random. We show additional experiments that support our claims on less memorization, faster sampling, and robustness to attacks. We set $\omega = 2$, $\gamma = 8/255$ and $\lambda = 0.3$: across datasets, we have observed that when raising it to $0.5$ we get an over-smoothing effect while low values prevent too much denoising. The adopted DDPM baseline is Nichol & Dhariwal (2021), whose available implementation was adopted as codebase.

## 3.1 EVALUATION USING DDPM AND DDIM

**Controlled Experiments.** Fig. 3 *(right)* shows the results when training on high dim. data living on a subspace. When training on the clean, regular dataset, the baseline and our Robust DMs perform similarly though **Robust**ran has slightly better PSNR. When we train on the noisy dataset, $\{\mathbf{x}' + \mathbf{z}\}_{i=1}^{N}$, then both Robust DMs offer superior performance (orange and blue curves) with wide gaps compared to the baseline (green curve) in both PSNR and reconstruction error. Specifically **Robust**adv appears to be better at noise unlearning. DDPM generations often consist in samples with saturated colors that are unlikely to be found in the training set while our method has better fidelity—see Fig. 3*(left)*.

**Random or adversarial perturbation?** We can also reply to this question by ablating on $\boldsymbol{\delta}_{\text{adv}}$ and $\boldsymbol{\delta}_{\text{ran}}$. Table 1 *(top)* shows that the adversarial perturbation can guarantee a much stronger denoising effect than random, yet is more expensive for training. The impact of our Eq. (6) is remarkable even in the case of random perturbation with an FID far below the baselines.

DDPM Ho et al. (2020)      **Robust**adv

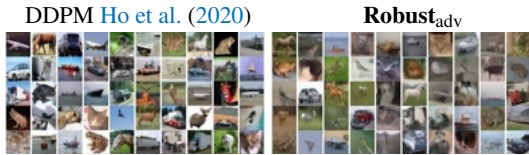

Figure 5: Despite the FID increasing once trained on clean data, images by **Robust**adv appear smoother and background clutter is removed.

**Resistant to noise by design.** Table 2 compares our approach with the baseline DDPM and DDIM on CIFAR-10, CelebA, and LSUN Bedroom, yet corrupted with shite noise. We show that , if we apply our method to the original dataset with no noise ($p = 0\%$), we only get a slight increase in the FID. However, if we visually inspect the results, we discover that ours is actually

Table 1: Top: Random vs adv. noise. Bottom: **Robust**$_{adv}$ allows fewer steps for better FID. Results on CIFAR-10.

| $\sigma \rightarrow$ | 0.1 | | 0.2 | |
|---|---|---|---|---|
| metrics $\rightarrow$ | FID | IS | FID | IS |
| **Robust**$_{ran}$ | 79.21 | 5.21 | 68.04 | 4.34 |
| **Robust**$_{adv}$ | **24.70** | **7.21** | **24.81** | **7.07** |

| steps $\rightarrow$ | 300 | | 500 | |
|---|---|---|---|---|
| metrics $\rightarrow$ | FID | IS | FID | IS |
| DDPM | 224.38 | 3.33 | 28.07 | **8.46** |
| **Robust**$_{adv}$ | **37.89** | **6.39** | **24.34** | 7.53 |

Table 2: Performance under different noise levels on different real datasets. Values indicate FID $\downarrow$ / IS $\uparrow$.

| p % | $\sigma$ | DDPM | Robust$_{adv}$ | DDIM | Robust$_{adv}$ |
|---|---|---|---|---|---|
| | | **CIFAR-10** | | | |
| 0 | 0 | **7.2 / 8.95** | 28.68 / 7.04 | **11.62 / 8.36** | 31.20 / 6.38 |
| 0.9 | 0.1 | 58.05 / 6.93 | **24.70 / 7.21** | 59.28 / 6.89 | **25.48 / 6.85** |
| 0.9 | 0.2 | 102.68 / 4.19 | **24.81 / 7.07** | 105.43 / 4.09 | **24.93 / 6.69** |
| | | **CelebA** | | | |
| 0 | 0 | **3.49 / 2.61** | 19.83 / 2.13 | **6.19 / 2.61** | 17.59 / 2.18 |
| 0.9 | 0.1 | 54.90 / **2.40** | **14.54** / 2.09 | 41.29 / **2.48** | **17.98** / 2.22 |
| 0.9 | 0.2 | 96.03 /**2.65** | **16.53** / 2.11 | 89.28 / **2.62** | **20.24** / 2.20 |
| | | **LSUN Bedroom** | | | |
| 0 | 0 | **9.90** / 2.31 | 57.13 / **2.34** | **27.00** / 3.15 | 48.80 / 2.39 |
| 0.9 | 0.1 | 53.81 / **3.33** | **44.07** / 2.35 | 50.53 / 3.19 | **48.90 / 3.96** |
| 0.9 | 0.2 | 95.85 / **4.08** | **44.27** / 2.50 | 82.20 / **4.39** | **61.98** / 3.66 |

smoothing background features, but still outlines of the objects are visible, as shown in Fig. 5 and Section E. When we switch to noisy settings, we have a large improvement over the baseline for both DDPM and DDIM. We highlight that while the baseline FIDs skyrocket to very high values for $p = 90\%, \sigma = 0.2$, the **Robust**$_{adv}$ can keep it in a reasonable range, generating images unaffected by the noise. We also provide early results on the ImageNet (Russakovsky et al., 2014) dataset, which comprises 1.28M images, downsampled at a resolution of $64 \times 64$ pixels following Daras et al. (2024a). The regularization also works effectively on this more complex dataset, resulting in a decrease in FID from 97.6 to 83.8 for $p = 90\%, \sigma = 0.1$ and from 129.4 to 80.3 for $p = 90\%, \sigma = 0.2$. Quantitative evaluations are provided in Table 2, showing major improvement of the regularized training over standard training. Fig. 4 illustrates our method's benefits on the proposed datasets under noisy data conditions. More results and images are available in the appendix.

**Time complexity.** Training with AT strongly impacts training time due to the overhead of computations needed. DDPM training operations comprehend a single forward pass to get model prediction and a backward pass for weights update. Our regularization adds a backward pass to obtain adversarial loss gradients over the perturbation and doubles the same DDPM operations. The time complexity is $\times 2.5$ for **Robust**$_{adv}$, whereas **Robust**$_{ran}$ is less time-consuming since it does not have to backpropagate for the adversarial perturbation. Despite the training time being higher than the baseline, remarkably, the inference time is the same as other methods, and we can attain faster sampling—see Section 3.3.

## 3.2 ROBUST DIFFUSION MODELS MEMORIZE LESS

Following Daras et al. (2024d) we show that Robust DMs are naturally less prone to memorize the training data. We perform an experiment following Somepalli et al. (2023): using DDPM and our **Robust**$_{adv}$ trained on clean CIFAR-10, we synthesize 50K images from each of them and measure the similarities of those images with the one in the training set, embedding the images with DINO-v2 Oquab et al. (2023). In Daras et al. (2024d) a similar experiment was done yet using DeepFloyd IF instead of U-Net DDPM. Although U-Net has much less parameters than DeepFloyd IF—millions vs billions—one could assume that U-Net will overfit less. Fig. 6 (*left*) shows that still a decent amount of generated samples have similarity higher than 0.90. Similarity $\geq 0.9$ roughly corresponds to the same CIFAR image. Robust models have a histogram that is drastically shifted on the left and the curve of the histogram in the right part decays more rapidly, having less samples in the region $\geq 0.9$.

## 3.3 IMPLICATIONS OF SMOOTH DIFFUSION FLOW

**Smooth diffusion flow.** Fig. 6 (*right*) shows the diffusion flow from the standard normal distribution to the data distribution. To do so, we use DDPM framework and low-dimensional 3D data, projected to 2D for clarity. In `oblique-plane`, we can see how **Robust**$_{adv}$ captures less variability, filtering out noise, while DDPM heatmap is more faded. Moreover, DDPM, misled by the noise, introduces a very subtle additional mode, whereas ours maintains a unimodal generation. The same remarks hold for a multi-modal dataset: in `3-gaussians` DDPM's trajectories are distorted by noise, while ours remain straight, preserving the multi-modal structure (only two modes are visible due to projection). **Trade-off analysis on clean and noisy data.** This sharpening of the trajectories leads to a reduction in

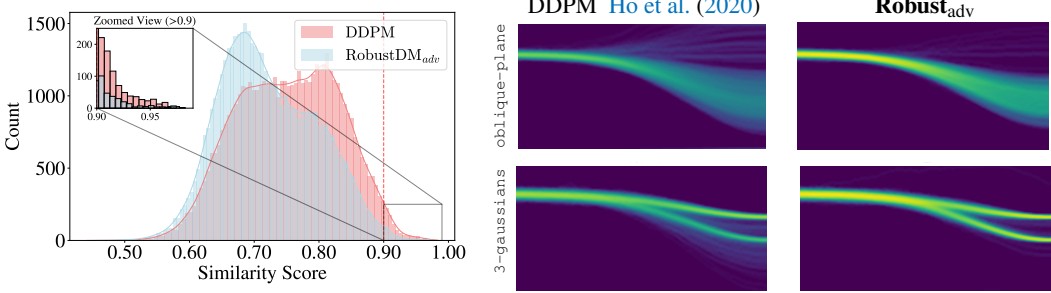

Figure 6: (*left*) The histogram shows similarities between generated samples and CIFAR-10, with values above 0.9 indicating near-duplicates. DDPM memorizes more, while **Robust**$_{\text{adv}}$ reduces near-replicas. (*right*) Regular training tends to have diverging trajectories w.r.t. the data distribution, while **Robust**$_{\text{adv}}$ trades off variability for resilience with trajectories more clustered, sharp, and less faded.

the variance of the generated data, but it does not induce mode collapse. As a result, the generated images may lose some high-frequency noise and fine details, producing outputs that appear smoother overall. Nevertheless, by applying regularization as determined by the parameter $\lambda$, we can effectively modulate its action and thus its smoothing effect. The analysis of the regularization effect, depending on $\lambda$ of Eq. (6), enables us to define an existing trade-off between image quality and robustness, as well as denoising capabilities, similar to the widely examined trade-off between robust and clean accuracy in robust classifiers. Qualitative examples supporting this are provided in Figs. 7 and 25, where we illustrate how model performance varies when the $\lambda$ is changed and how it affects both generation and denoising capabilities.

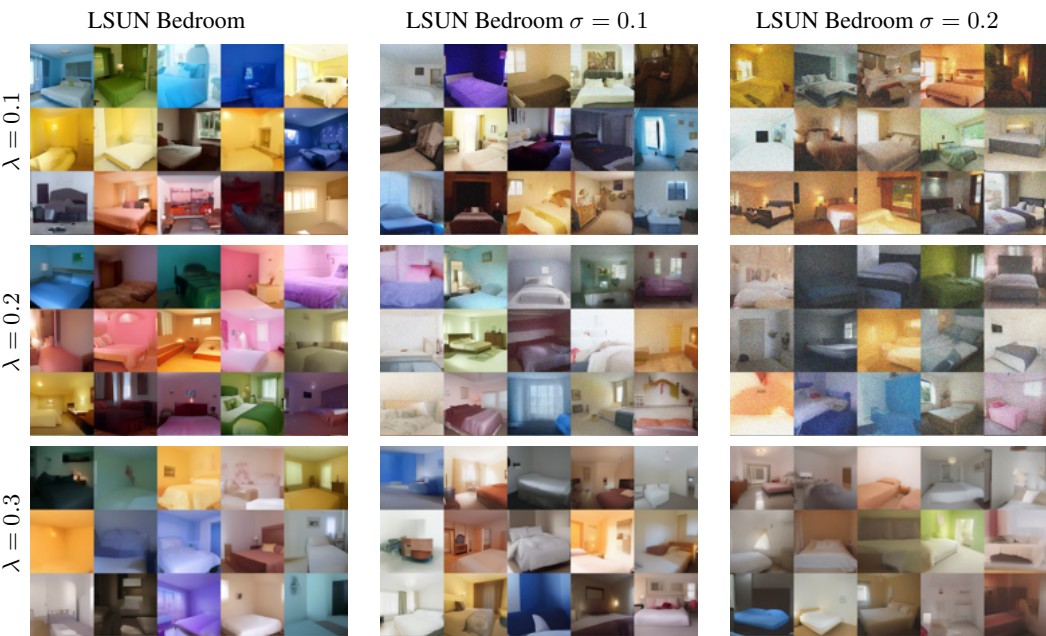

Figure 7: **Robust**$_{\text{adv}}$ trained on LSUN Bedroom dataset, with different noisy data ($p = 90\%$, different $\sigma$ are visible in the image) and varying hyperparameter $\lambda = \{0.1, 0.2, 0.3\}$.

**Faster sampling.** Fig. 6 (*right*) shows that the diffusion flow of **Robust**$_{\text{adv}}$ is more compact and sharp, less faded. This could imply that the inference process may still recover the right path in case the regressed score vector is corrupted or is noisy or in case we deliberately use fewer steps in Eq. (4) for faster sampling. We tested this hypothesis and the trade-off table of FID in function of the number of steps taken is shown in Table 1 (*bottom*). Even more, if we cross compare Table 1 (*bottom*) with Table 2, on clean data **Robust**$_{\text{adv}}$ scores a better FID with 500 steps (24.34) vs 1000 steps (28.68). This experiment supports our claim showing that **Robust**$_{\text{adv}}$ is still able to generate samples with good

fidelity even if using fewer inference steps. The degradation using less steps is widely more graceful than DDPM especially when we take only 300 steps over 1000.

### 3.4 ROBUSTNESS TO ADVERSARIAL ATTACKS

Our method is naturally resistant to attacks. Like classifiers, AT enforces robustness to adversarial perturbations in the diffusion flow. We propose an attack primarily as an analytical tool to better understand the fundamental sensitivity of the generative process to perturbations. The attack takes into account the stochastic nature of DM inference and the fundamental hypothesis of gaussianity for each diffusion stage. We propose attacking a DM in a white-box setting defining a sequence of adversarial perturbations that could maximally disrupt the trajectory at *some* of the intermediate inference steps, defined as described in Algorithm 2. We also propose a procedure to determine the range of values of the perturbation in order to maintain the assumption of the diffusion process; more information can be found in Section A.2. Fig. 8 shows that our method is much more robust to attacks in

---

**Algorithm 2** DM Trajectory Attack

**Input:** attack ratio $p$, max timesteps $T$, model $\epsilon_\theta$, scheduler $\alpha_t$, $\sigma(t)$, attack strength $\phi$;
Sample $\mathbf{x}_T \sim \mathcal{N}(0, I)$
**for** $t = T$ to $0$ **do**
$\quad \mathbf{x}_{t-1} \leftarrow \epsilon_\theta(\mathbf{x}_t, t)$
$\quad \hat{\mathbf{x}}_0 \leftarrow \frac{\mathbf{x}_t - \sqrt{1-\bar{\alpha}_t}\epsilon_\theta(\mathbf{x}_t,t)}{\sqrt{\bar{\alpha}_t}}$
$\quad \tilde{\boldsymbol{\mu}}_t(\mathbf{x}_t, \hat{\mathbf{x}}_0) \leftarrow \frac{\sqrt{\bar{\alpha}_{t-1}}\sigma(t)}{1-\bar{\alpha}_t}\hat{\mathbf{x}}_0 + \frac{\sqrt{\alpha_t}(1-\bar{\alpha}_{t-1})}{1-\bar{\alpha}_t}\mathbf{x}_t$
$\quad \mathbf{x}'_t = \mathbf{x}_t + \boldsymbol{\delta}, \boldsymbol{\delta} \sim \mathcal{N}(0, \phi^2\sigma(t)^2 I)$
$\quad \mathbf{x}'_{t-1} \leftarrow \epsilon_\theta(\mathbf{x}'_t, t)$
$\quad \hat{\mathbf{x}}'_0 \leftarrow \frac{\mathbf{x}'_t - \sqrt{1-\bar{\alpha}_t}\epsilon_\theta(\mathbf{x}'_t,t)}{\sqrt{\bar{\alpha}_t}}$
$\quad L \leftarrow \left\| \tilde{\boldsymbol{\mu}}_t(\mathbf{x}_t, \hat{\mathbf{x}}_0) - \tilde{\boldsymbol{\mu}}_t(\mathbf{x}'_t, \hat{\mathbf{x}}'_0) \right\|_2^2$
$\quad \mathbf{x}_t^{\text{adv}} = \mathbf{x}_t + \sigma(t) \cdot \text{sign}(\nabla_{\mathbf{x}_t} L)$
$\quad \mathbf{x}_{t-1} \leftarrow \epsilon_\theta(\mathbf{x}_t^{\text{adv}}, t)$
**end for**

---

diffusion flow: **Robust**$_{\text{adv}}$ can tolerate up to $50\%$ of time step attacked and still generate samples with decent fidelity. Only at $75\%$ time steps attacked, the generation fails for both. The attack illustrated in Algorithm 2 is a single-iteration attack. In Section A.3, we extend the pool of considered attacks to include iterative PGD (Madry et al., 2018), and provide the model's performance in that setting.

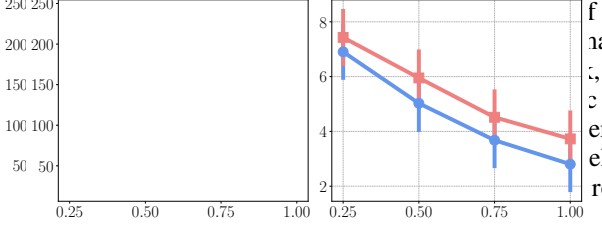

f the diffusion-based inference process, we
nation (EoT) (Athalye et al., 2018). We adopt
, as previously introduced. In particular, the
noise samples. The average gradient is then
ementation can be found in Section A.4. The
el also effectively demonstrates its resilience
robustness to EoT-based attacks.

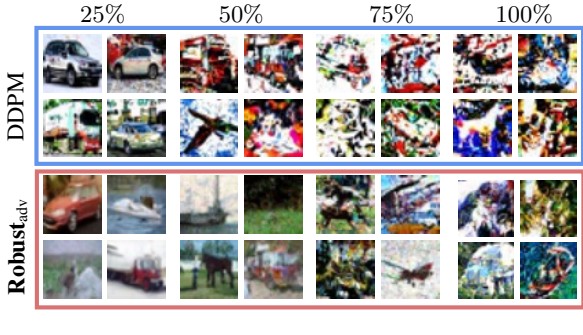

|  | 25% | 50% | 75% | 100% |
|---|---|---|---|---|

**FID ↓ w/ FGSM** — Algorithm 2 and Section A.2

| steps attacked → | 250 | 500 | 750 | 1000 |
|---|---|---|---|---|
| DDPM | 49.8 | 131.7 | 190.4 | 243.4 |
| **Robust**$_{\text{adv}}$ | **19.29** | **52.0** | **90.7** | **127.7** |

**FID ↓ w/ PGD** — Algorithm 3 and Section A.3

| steps attacked → | 250 | 500 | 750 | 1000 |
|---|---|---|---|---|
| DDPM | 55.7 | 134.5 | 200.3 | 248.1 |
| **Robust**$_{\text{adv}}$ | **22.7** | **55.8** | **98.1** | **128.6** |

**FID ↓ w/ EoT-PGD** — Algorithm 4 and Section A.4

| steps attacked → | 250 | 500 | 750 | 1000 |
|---|---|---|---|---|
| DDPM | 57.6 | 132.7 | 195.99 | 248.6 |
| **Robust**$_{\text{adv}}$ | **25.9** | **61.2** | **100.2** | **132.6** |

(a)                                (b)

Figure 8: **(a)** Robustness to Adversarial Attacks. While the baseline DDPM is susceptible to adversarial attacks, Robust DMs better resist them, yielding superior FID and IS for different percentages of time steps attacked (e.g., 25% means 250 out of 1000 DDPM steps are attacked). **(b)** FID under FGSM, PGD and EoT applied to PGD, varying the percentage of attacked timesteps.

## 4 RELATED WORK

**Diffusion models.** Score-based generative models (Song & Ermon, 2019) express the inference process through a Stochastic Differential Equations (SDE) Dhariwal & Nichol (2021). Denoising Diffusion Probabilistic Models (DDPMs) Ho et al. (2020) first introduced diffusion process as a score-based generative framework, becoming a standard algorithm in generative modeling on high-dimensional data, overcoming Goodfellow et al. (2020). DMs not only achieve higher fidelity but also provide a more stable training. DMs have been extensively improved: working on the logarithmic likelihood estimate Nichol & Dhariwal (2021), faster sampling Song et al. (2021a), and performing the diffusion process in the latent space Rombach et al. (2022). Karras et al. (2022; 2024) provide insightful clarifications on several DMs design choices, introducing improved U-Net architectures that ensure consistent activation, weight, and update magnitude, achieving state-of-the-art FID on CIFAR and other benchmarks. Lastly, Song et al. (2023) proposed *consistency* models, a distillation method for one-step inference by directly mapping noise to data. The name *consistency* arises from the fact that they enforce different noisy versions in the same trajectory to map to the same data. Unlike them, we do not aim to distill a model, but rather to train one enforcing local *smoothness* of trajectories within the same timestep $t$, so that their score field remains locally consistent.

**Denoising and inverse problems with DMs.** The attention to apply DMs on corrupted data has increased in recent years (Aali et al., 2023; Xiang et al., 2023; Daras et al., 2024d;a). Given the specific challenges related to training with noisy data, this problem is closely related to inverse problems (Tachella et al., 2024; Kawar et al., 2024). Recently, a line of research focused on applying Stein's Unbiased Risk Estimator (SURE) (Metzler et al., 2020) and its subsequent improvements, including UNSURE (Tachella et al., 2024), GSURE (Kawar et al., 2024), Soft Diffusion (Daras et al., 2024b), and methods leveraging optimal transport for training with noise (Dao et al., 2024).

**Adversarial robustness.** This topic is loosely linked to denoising since AT can be seen as a way to remove spurious correlations (Ye et al., 2024) with improved out-of-domain generalization when transferring to a new domain (Ilyas et al., 2019) or related to causal learning (Zhang et al., 2020; 2022). AT variants have been used to improve domain shift (Salman et al., 2020a) and out of distribution (Wang et al., 2022). While it is reasonable to say that AT has been extensively studied on classifiers, its application to DMs remains unexplored, except for Sauer et al. (2024), where it is applied for fast sampling, Yang et al. (2024), which investigates the batch samples interconnection, and Lorenz et al. (2024), which found adversarial samples do not align with the learned DM manifold. Finally, Wang et al. (2025) proposed a similar equivariant loss to ours, yet they do not demonstrate resilience against attacks or robustness to noisy data.

**Adversarial defenses with denoising or randomized smoothing.** Several adversarial defenses leverage denoising (Salman et al., 2020b; Carlini et al., 2023a) and randomized smoothing (Cohen et al., 2019), mainly in the context of classifiers. Regarding DMs, Song et al. (2024); Liang et al. (2023); Liang & Wu (2023) have shown that adversarial perturbations, if applied at inference time, can significantly disrupt their generative capabilities, leading to deviations from clean data distributions. Further works introduce the concept of robustness when fine-tuning DMs to make them robust in the context of adversarial purification (Song et al., 2018; Nie et al., 2022; Lin et al., 2024). While these methods differ in the definition of adversarial samples (Li et al., 2025; Liu et al., 2025), they share similar underlying objectives. Our work introduces AT in Diffusion Models to enforce local smoothness in the score field, which may help counteract such deviations during inference. Unlike Guo et al. (2024), our work aims to smooth model trajectories, not embeddings (Section A.5), unlike SmoothDiffusion.

## 5 CONCLUSIONS AND FUTURE WORK

We presented the first attempt to incorporate AT into DM training, demonstrating that AT for generative modeling entails smoothing the data distribution and can be effectively utilized for denoising the data. We also show that we need to reinterpret it as *equivariant* property and not *invariance*. Our method has been proven to be highly robust even under 90% of corrupted data with strong Gaussian noise. In terms of future work, we aim to extend this work to a robust fine-tuning technique that is applicable to larger, new models with reduced training costs. Preliminary results are presented in Section F.2. We also plan to extend our method to work in fully corrupted settings ($p = 100\%$) and port our approach to EDM (Karras et al., 2022; 2024) to scale to larger datasets.

**Ethics Statement.** Based on our comprehensive analysis, we assert that this work does not raise identifiable ethical concerns or foreseeable negative societal consequences within the scope of our study. On the contrary, our contributions aim to enhance the robustness of Diffusion Models against attacks.

**Reproducibility.** To ensure reproducibility, we provide a detailed description of our experimental setup in Section 3 including datasets, models, and adversarial attacks, along with their sources. The codebase we adopted for building the AT framework is Nichol & Dhariwal (2021), and the our code is released at github.com/OmnAI-Lab/Adversarial-Training-DM

**LLM Usage.** Large language models were used exclusively for text polishing and minor exposition refinements. All substantive research content, methodology, and scientific conclusions were developed entirely by the authors

**Acknowledgment.** This work was supported by projects PNRR MUR PE0000013-FAIR under the MUR National Recovery and Resilience Plan funded by the European Union - NextGenerationEU, PRIN 2022 project 20227YET9B "AdVVent" CUP code B53D23012830006. It was also partially supported by Sapienza research projects D2QNeT and BEAT (Better dEep leArning securiTy) — bando per la ricerca di Ateneo 2024. Computing was supported by the CINECA clusters under the projects Ge-Di HP10CRPUVC and RDM HP10C7YYL2.

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

## A  Appendix

### A.1  Theoretical considerations on adversarial training for diffusion models

To craft an appropriate adversarial loss, at first, forward and reverse processes are redefined in light of this further intermediate state. The main aim of performing adversarial training on a diffusion model is to enhance the robustness capability against adversarial attacks in its reverse process by providing the algorithm with some data that has previously been corrupted. We model this corruption process as an additional chain state, and in this section, we provide a theoretical discussion for this assumption.

#### A.1.1  The forward process

The theoretical definition of the DDPM forward process is the following:

$$q\left(\mathbf{x}_{1:T} \mid \mathbf{x}_0\right) := \prod_{t=1}^{T} q\left(\mathbf{x}_t \mid \mathbf{x}_{t-1}\right), \quad q(\mathbf{x}_t \mid \mathbf{x}_{t-1}) = \mathcal{N}\left(\mathbf{x}_t; \sqrt{1 - \sigma(t)}\mathbf{x}_{t-1}, \sigma(t)\mathbf{I}\right).$$

where $q(\mathbf{x}_t \mid \mathbf{x}_{t-1})$ represents the transition probability of the process to move from the state $\mathbf{x}_{t-1}$ at the timestep $t-1$ to the state $\mathbf{x}_t$ at the timestep $t$. To achieve the aim of integrating the perturbation in the framework, the forward chain can be redefined considering a different dynamic of the adversarial forward process. A sample at the time step $t$ is first derived as defined above, and then to it is added an adversarial perturbation $\boldsymbol{\delta}_{\theta,t}$ that depends on the model's actual state and on the value of $\mathbf{x}_t$. The overall attack procedure to the model intermediate steps can be represented as a concatenation of two transitions. The primary step is the ordinary DDPM transition from $\mathbf{x}_{t-1}$ to $\mathbf{x}_t$, which is modeled as $q(\mathbf{x}_t \mid \mathbf{x}_{t-1})$. The attack transition can be modeled as the step that goes from $\mathbf{x}_t$ to $\mathbf{x}_t + \boldsymbol{\delta}_t$ in the $t$-th timestep, being defined similarly as above $q'(\mathbf{x}_t + \boldsymbol{\delta} \mid \mathbf{x}_t)$. The two transitions are designed to happen in the same time step $t$ of the chain and, being independent of each other, it is possible to model their interaction as a sub-sequence of steps of a Markov Chain. This is possible since the transition $\mathbf{x}_{t-1} \to \mathbf{x}_t$ is already modeled like this and $\mathbf{x}_t \to \mathbf{x}_t + \boldsymbol{\delta}_t$ depends only on the weights of the model (which are constant when crafting the attack, so considerable as constant within the same evaluation) and the value of $\mathbf{x}_t$ conceived as "previous state". The resulting transition probability $q''(\mathbf{x}_t + \boldsymbol{\delta}_t \mid \mathbf{x}_{t-1})$ is:

$$q''(\mathbf{x}_t + \boldsymbol{\delta}_t \mid \mathbf{x}_{t-1}) = q'(\mathbf{x}_t + \boldsymbol{\delta}_t \mid \mathbf{x}_t) \cdot q(\mathbf{x}_t \mid \mathbf{x}_{t-1}).$$

The overall chain can be written as:

$$q''\left(\mathbf{x}_{1:T} + \boldsymbol{\delta}_{1:T} \mid \mathbf{x}_0\right) = \prod_{t=1}^{T} q'\left(\mathbf{x}_t + \boldsymbol{\delta}_t \mid \mathbf{x}_t\right) \cdot q\left(\mathbf{x}_t \mid \mathbf{x}_{t-1}\right).$$

with $q(\mathbf{x}_t \mid \mathbf{x}_{t-1}) = \mathcal{N}\left(\mathbf{x}_t; \sqrt{1 - \sigma(t)}\mathbf{x}_{t-1}, \sigma(t)\mathbf{I}\right)$. Being $q(\cdot)$ a Gaussian transition and being the perturbation addition still modeled as a Gaussian transition, the DM hypothesis of having only intermediate Gaussian transitions still holds.

#### A.1.2  Reverse process

The reverse process in the diffusion models aims to define an algorithm that approximates the forward function and makes it possible to reconstruct the input. In the DDPM formulation, the backward process is defined as:

$$p_\theta\left(\mathbf{x}_{0:T}\right) := p\left(\mathbf{x}_T\right) \prod_{t=1}^{T} p_\theta\left(\mathbf{x}_{t-1} \mid \mathbf{x}_t\right), \quad p_\theta\left(\mathbf{x}_{t-1} \mid \mathbf{x}_t\right) := \mathcal{N}\left(\mathbf{x}_{t-1}; \boldsymbol{\mu}_\theta\left(\mathbf{x}_t, t\right), \boldsymbol{\Sigma}_\theta\left(\mathbf{x}_t, t\right)\right).$$

Following the previous substitutions, the desired equivalence when applying perturbations in the forward process would be :

$$p(\mathbf{x}_t + \boldsymbol{\delta}_t \mid \mathbf{x}_{t-1}) = q''(\mathbf{x}_t + \boldsymbol{\delta}_t \mid \mathbf{x}_{t-1}).$$

that, if considering its approximation, reduces to:

$$p(\mathbf{x}_{t-1} \mid \mathbf{x}_t + \boldsymbol{\delta}_t) \propto p(\mathbf{x}_t + \boldsymbol{\delta}_t \mid \mathbf{x}_{t-1}) \cdot p(\mathbf{x}_{t-1}).$$

This consideration holds also in this case, so if we substitute the objective distributions $p(\cdot)$ with the desired ones we get:

$$p(\mathbf{x}_{t-1} \mid \mathbf{x}_t + \boldsymbol{\delta}_t) \propto p(\mathbf{x}_{t-1}) \cdot q''(\mathbf{x}_t + \boldsymbol{\delta}_t \mid \mathbf{x}_{t-1}) = p(\mathbf{x}_{t-1}) \cdot q'(\mathbf{x}_t + \boldsymbol{\delta}_t \mid \mathbf{x}_t) \cdot q(\mathbf{x}_t \mid \mathbf{x}_{t-1}).$$

The above equations hold in case the reverse process is defined in closed form, while in our case the reverse function is a learned function by $p_\theta(\cdot)$, which is designed and learned to properly converge to $p_{\text{data}}(\mathbf{x})$ at a specific timestep 0 of the chain. To properly learn this objective, the network is trained to learn to regress the amount of noise added in the forward process by minimizing the following simplified objective:

$$\mathcal{L}(\mathbf{x}_t; \boldsymbol{\theta}) = \left\| \boldsymbol{\epsilon} - \boldsymbol{\epsilon}_{\boldsymbol{\theta}}(\mathbf{x}_t, t) \right\|_2^2 \quad \text{where} \quad \boldsymbol{\epsilon} \sim \mathcal{N}(0, \mathbf{I}) \quad \text{given } t \in [0, \ldots, T]. \tag{9}$$

where $q'(\mathbf{x}_t + \boldsymbol{\delta}_t \mid \mathbf{x}_t)$ represents the transition probability of going from the state $\mathbf{x}_t$ to the state $\mathbf{x}_t' = \mathbf{x}_t + \boldsymbol{\delta}_t$ in the same timestep $t$, the transition from an uncorrupted state to a corrupted one through $\boldsymbol{\delta}_t$. In this case, there is no modeling available as the distribution depends on the kind of attack being performed during the training proces,s but also depends on the state of the model, as the attack is crafted in white box mode:

$$\boldsymbol{\delta}_{\theta,t} = \underset{\|\boldsymbol{\delta}\| \leq \varepsilon}{\arg \max} \left\| \boldsymbol{\epsilon}_\theta(\mathbf{x}_t + \boldsymbol{\delta}, t) - \boldsymbol{\epsilon}_\theta(\mathbf{x}_t, t) \right\|_2^2.$$

Given the proposed setting, the aim is to define a cost function that allows for modeling the correct $\mathbf{x}_{t-1}$ when considering the inverted process. The probability distribution that the reverse process needs to learn is:

$$p_\theta(\mathbf{x}_{t-1} \mid \mathbf{x}_t + \boldsymbol{\delta}_{\theta,t}) \propto p_\theta(\mathbf{x}_{t-1} \mid \mathbf{x}_t) p_\theta'(\mathbf{x}_t \mid \mathbf{x}_t + \boldsymbol{\delta}_{\theta,t}).$$

### A.1.3 VARIATIONAL LOWER BOUND IN CASE OF PERTURBATION

The Diffusion Models loss function is derived from an optimization regarding the variational lower bound. The ELBO is defined canonically as:

$$\mathbb{E}[-\log p_\theta(\mathbf{x}_0)] \leq \mathbb{E}_q\left[ -\log \frac{p_\theta(\mathbf{x}_{0:T})}{q(\mathbf{x}_{1:T} \mid \mathbf{x}_0)} \right] = \mathbb{E}_q\left[ -\log(p_{\mathbf{x}_t}) - \sum_{t \geq 1} \log \frac{p_\theta(\mathbf{x}_{t-1} \mid \mathbf{x}_t)}{q(\mathbf{x}_t \mid \mathbf{x}_{t-1})} \right] := L \tag{10}$$

and the Diffusion Model loss derivation is the following:

$$
\begin{aligned}
L &= \mathbb{E}_q\left[ -\log \frac{p_\theta(\mathbf{x}_{0:T})}{q(\mathbf{x}_{1:T} \mid \mathbf{x}_0)} \right] \\
&= \mathbb{E}_q\left[ -\log p(\mathbf{x}_T) - \sum_{t \geq 1} \log \frac{p_\theta(\mathbf{x}_{t-1} \mid \mathbf{x}_t)}{q(\mathbf{x}_t \mid \mathbf{x}_{t-1})} \right] \\
&= \mathbb{E}_q\left[ -\log p(\mathbf{x}_T) - \sum_{t > 1} \log \frac{p_\theta(\mathbf{x}_{t-1} \mid \mathbf{x}_t)}{q(\mathbf{x}_t \mid \mathbf{x}_{t-1})} - \log \frac{p_\theta(\mathbf{x}_0 \mid \mathbf{x}_1)}{q(\mathbf{x}_1 \mid \mathbf{x}_0)} \right] \\
&= \mathbb{E}_q\left[ -\log p(\mathbf{x}_T) - \sum_{t > 1} \log \frac{p_\theta(\mathbf{x}_{t-1} \mid \mathbf{x}_t)}{q(\mathbf{x}_{t-1} \mid \mathbf{x}_t, \mathbf{x}_0)} \cdot \frac{q(\mathbf{x}_{t-1} \mid \mathbf{x}_0)}{q(\mathbf{x}_t \mid \mathbf{x}_0)} - \log \frac{p_\theta(\mathbf{x}_0 \mid \mathbf{x}_1)}{q(\mathbf{x}_1 \mid \mathbf{x}_0)} \right] \\
&= \mathbb{E}_q\left[ -\log \frac{p(\mathbf{x}_T)}{q(\mathbf{x}_T \mid \mathbf{x}_0)} - \sum_{t > 1} \log \frac{p_\theta(\mathbf{x}_{t-1} \mid \mathbf{x}_t)}{q(\mathbf{x}_{t-1} \mid \mathbf{x}_t, \mathbf{x}_0)} - \log p_\theta(\mathbf{x}_0 \mid \mathbf{x}_1) \right]
\end{aligned}
$$

$$= \mathbb{E}_q \left[ D_{\mathrm{KL}} \left( q\left(\mathbf{x}_T \mid \mathbf{x}_0\right) \| p\left(\mathbf{x}_T\right)\right) + \sum_{t>1} D_{\mathrm{KL}} \left( q\left(\mathbf{x}_{t-1} \mid \mathbf{x}_t, \mathbf{x}_0\right) \| p_\theta \left(\mathbf{x}_{t-1} \mid \mathbf{x}_t\right)\right) - \log p_\theta \left(\mathbf{x}_0 \mid \mathbf{x}_1\right) \right].$$

In light of the previous considerations of the forward and backward process, it is possible to reconsider ELBO derivation as follows:

$$
\begin{aligned}
L &= \mathbb{E}_q \left[ -\log \frac{p_\theta(\mathbf{x}_{0:T})}{q''(\mathbf{x}_{1:T} \mid \mathbf{x}_0)} \right] \\
&= \mathbb{E}_q \left[ -\log \frac{p_\theta(\mathbf{x}_T) \prod_{t=1}^{T} p_\theta(\mathbf{x}_{t-1} \mid \mathbf{x}_t + \delta_t)}{\prod_{t=1}^{T} q(\mathbf{x}_t \mid \mathbf{x}_{t-1}) \, q'(\mathbf{x}_t + \delta_t \mid \mathbf{x}_t)} \right] \\
&= \mathbb{E}_q \left[ -\log p(\mathbf{x}_T) - \sum_{t \geq 1} \log \frac{p_\theta(\mathbf{x}_{t-1} \mid \mathbf{x}_t + \delta_t)}{q(\mathbf{x}_t \mid \mathbf{x}_{t-1}) \, q'(\mathbf{x}_t + \delta_t \mid \mathbf{x}_t)} \right] \\
&= \mathbb{E}_q \left[ -\log p(\mathbf{x}_T) - \sum_{t \geq 1} \log \frac{p_\theta(\mathbf{x}_{t-1} \mid \mathbf{x}_t)}{q(\mathbf{x}_t \mid \mathbf{x}_{t-1})} - \sum_{t \geq 1} \log \frac{p'_\theta(\mathbf{x}_t \mid \mathbf{x}_t + \delta_t)}{q'(\mathbf{x}_t + \delta_t \mid \mathbf{x}_t)} \right] \\
&= \mathbb{E}_q \left[ -\log p(\mathbf{x}_T) - \sum_{t > 1} \log \frac{p_\theta(\mathbf{x}_{t-1} \mid \mathbf{x}_t)}{q(\mathbf{x}_{t-1} \mid \mathbf{x}_t, \mathbf{x}_0)} \frac{q(\mathbf{x}_{t-1} \mid \mathbf{x}_0)}{q(\mathbf{x}_t \mid \mathbf{x}_0)} \right. \\
&\qquad \left. - \sum_{t > 1} \log \frac{p'_\theta(\mathbf{x}_t \mid \mathbf{x}_t + \delta_t)}{q'(\mathbf{x}_t \mid \mathbf{x}_t + \delta_t, \mathbf{x}_0)} \frac{q'(\mathbf{x}_t \mid \mathbf{x}_0)}{q'(\mathbf{x}_t + \delta_t \mid \mathbf{x}_0)} \right] \\
&= \mathbb{E}_q \left[ -\log \frac{p(\mathbf{x}_T)}{q''(\mathbf{x}_T \mid \mathbf{x}_0)} - \sum_{t > 1} \log \frac{p_\theta(\mathbf{x}_{t-1} \mid \mathbf{x}_t)}{q(\mathbf{x}_{t-1} \mid \mathbf{x}_t, \mathbf{x}_0)} \right. \\
&\qquad \left. - \sum_{t > 1} \log \frac{p'_\theta(\mathbf{x}_t \mid \mathbf{x}_t + \delta_t)}{q'(\mathbf{x}_t \mid \mathbf{x}_t + \delta_t, \mathbf{x}_0)} \frac{q'(\mathbf{x}_t \mid \mathbf{x}_0)}{q'(\mathbf{x}_t + \delta_t \mid \mathbf{x}_0)} \right] \\
&\qquad - \mathbb{E}_q \left[ \log p_\theta(\mathbf{x}_0 \mid \mathbf{x}_1) - \log p'_\theta(\mathbf{x}_0 \mid \mathbf{x}_1 + \delta_1) \right] \\
&= \mathbb{E}_q \left[ -\log \frac{p(\mathbf{x}_T)}{q''(\mathbf{x}_T \mid \mathbf{x}_0)} - \sum_{t > 1} \log \frac{p_\theta(\mathbf{x}_{t-1} \mid \mathbf{x}_t)}{q(\mathbf{x}_{t-1} \mid \mathbf{x}_t, \mathbf{x}_0)} \right. \\
&\qquad \left. - \sum_{t > 1} \log \frac{p'_\theta(\mathbf{x}_t \mid \mathbf{x}_t + \delta_t)}{q'(\mathbf{x}_t \mid \mathbf{x}_t + \delta_t, \mathbf{x}_0)} - \log p_\theta(\mathbf{x}_0 \mid \mathbf{x}_1) - \log p'_\theta(\mathbf{x}_0 \mid \mathbf{x}_1 + \delta_1) \right].
\end{aligned}
\tag{11}
$$

The components to be optimized can be seen as two KL-divergences, recalling the formal definition of DDPM optimization. To lower the loss functions the two resulting KL divergences have to be reduced by optimizing both the measure of divergence between the forward $\mathbf{x}_t$ and the approximated one, by correctly estimating the $\epsilon$ and the measure of the $\boldsymbol{\delta}_t$ noise is added to $\mathbf{x}_t$ at the timestep $t$. This distance measure is represented by the second KL divergence. To transition from the notation $q(\mathbf{x}_t \mid \mathbf{x}_{t-1})$ to $q(\mathbf{x}_{t-1} \mid \mathbf{x}_t, \mathbf{x}_0)$ it is first necessary to apply Bayes theorem and the chain rule of probability—the exact same reasoning can be used for the second sum.

1. Start with the conditional probability distribution $q'(\mathbf{x}_t \mid \mathbf{x}_{t-1})$.

2. Apply Bayes' theorem to express $q'(\mathbf{x}_t \mid \mathbf{x}_{t-1})$ in terms of $q'(\mathbf{x}_{t-1} \mid \mathbf{x}_t)$:

$$q(\mathbf{x}_t \mid \mathbf{x}_{t-1}) = \frac{q(\mathbf{x}_{t-1} \mid \mathbf{x}_t) \cdot q(\mathbf{x}_t)}{q(\mathbf{x}_{t-1})}.$$

3. Now, consider conditioning on an additional variable $\mathbf{x}_0$. According to the chain rule of probability, we have:

$$q(\mathbf{x}_{t-1}, \mathbf{x}_t) = q(\mathbf{x}_{t-1} \mid \mathbf{x}_t) \cdot q(\mathbf{x}_t).$$

4. We want to express $q(\mathbf{x}_{t-1} \mid \mathbf{x}_t)$ in terms of $\mathbf{x}_0$ as well. So, we can rewrite the joint distribution $q(\mathbf{x}_{t-1}, \mathbf{x}_t)$ as $q(\mathbf{x}_{t-1} \mid \mathbf{x}_t, \mathbf{x}_0) \cdot q(\mathbf{x}_t, \mathbf{x}_0)$.

5. Use the chain rule again to break down $q(\mathbf{x}_t, \mathbf{x}_0)$:

$$q(\mathbf{x}_t, \mathbf{x}_0) = q(\mathbf{x}_t \mid \mathbf{x}_0) \cdot q(\mathbf{x}_0).$$

6. Substituting these expressions back into our Bayes' theorem-derived expression, we get:

$$q(\mathbf{x}_t \mid \mathbf{x}_{t-1}) = \frac{q(\mathbf{x}_{t-1} \mid \mathbf{x}_t, \mathbf{x}_0) \cdot q(\mathbf{x}_t \mid \mathbf{x}_0)}{q(\mathbf{x}_{t-1} \mid \mathbf{x}_0)}.$$

7. Rearrange terms to isolate $q(\mathbf{x}_{t-1} \mid \mathbf{x}_t, \mathbf{x}_0)$, yielding the desired expression:

$$q(\mathbf{x}_{t-1} \mid \mathbf{x}_t, \mathbf{x}_0) = \frac{q(\mathbf{x}_t \mid \mathbf{x}_{t-1}) \cdot q(\mathbf{x}_{t-1} \mid \mathbf{x}_0)}{q(\mathbf{x}_t \mid \mathbf{x}_0)}.$$

## A.2 ATTACK FORMULATION

In inference mode, it is possible to represent the inverse Markov Chain as the sequence of intermediate realizations of Gaussian distributions with fixed parameters regarding mean scaling and variance scaling. From the paper Ho et al. (2020) in Eqs. 6 and 7 the $t$-th step of the inference can be written as the sampling from the posterior distribution $q(\mathbf{x}_{t-1} \mid \mathbf{x}_t t, \mathbf{x}_0) = \mathcal{N}(\mathbf{x}_{t-1}; \tilde{\boldsymbol{\mu}}_t(\mathbf{x}_t, \mathbf{x}_0), \tilde{\beta}_t \mathbf{I})$, where:

$$\tilde{\boldsymbol{\mu}}_t(\mathbf{x}_t, \mathbf{x}_0) := \frac{\sqrt{\alpha_{t-1}}\sigma(t)}{1 - \alpha_t}\mathbf{x}_0 + \frac{\sqrt{\alpha_t}(1 - \alpha_{t-1})}{1 - \alpha_t}\mathbf{x}_t, \quad \tilde{\beta}_t := \frac{1 - \alpha_{t-1}}{1 - \alpha_t}\sigma(t).$$

This implies that, at each time step, the expected variance and mean of the distribution are defined in a specific manner. During inference, the value of $\mathbf{x}_0$ corresponds to the output obtained after the network's prediction. In the context of the DDPM framework, $\mathbf{x}_0$ is replaced by the estimated value, which depends on the epsilon-predicting network:

$$\hat{\mathbf{x}}_0 = \frac{\mathbf{x}_t - \sqrt{1 - \alpha_t}\boldsymbol{\epsilon}_\theta(\mathbf{x}_t)}{\sqrt{\alpha_t}},$$

To properly craft the attack and still consider it legitimate, it is essential to scale it to the correct standard deviation to align with the diffusion process. Failing to do so would result in the network's inference being affected not by the perturbation itself but by the incorrect range of the perturbation, causing errors due to the inability to maintain the process within its Gaussian assumptions.

In this context, the attack procedure follows the FGSM approach with a random start. However, the perturbation is then scaled to match the appropriate variance at timestep $t$ to maintain consistency with the diffusion process. The FGSM attack generates an adversarial example by perturbing the noisy sample $\mathbf{x}_t$ in the direction of the gradient of a cost function $\mathcal{L}$ with respect to $\mathbf{x}_t$. Specifically, the adversarial perturbation is given by:

$$\mathbf{x}'_t = \mathbf{x}_t + \phi \cdot \text{sign}\big(\nabla_{\mathbf{x}_t}\mathcal{L}(\mathbf{x}_t)\big),$$

where $\phi$ controls the magnitude of the perturbation, $\text{sign}(\cdot)$ represents the element-wise sign function.

The adversarial attack in this approach is integrated into the diffusion process by leveraging the predictive functions, including a variance-handling mechanism defined in the model, in order to guarantee concretely adapting to the Gaussian hypothesis of the reverse Markov Chain. The adversarial attack begins with perturbing the input $\mathbf{x}_t$ defining its $\mathbf{x}'_t$ as:

$$\mathbf{x}'_t = \mathbf{x}_t + \boldsymbol{\delta}, \qquad \boldsymbol{\delta} \triangleq \mathcal{N}(0, \phi^2 \cdot \sigma(t)^2).$$

The cost function for the adversarial attack is theoretically defined based on the mean prediction:

$$\mathcal{L}_{FGSM} = \big\|\tilde{\boldsymbol{\mu}}_t(\mathbf{x}_t, \mathbf{x}_0) - \tilde{\boldsymbol{\mu}}_t(\mathbf{x}'_t, \mathbf{x}_0)\big\|_2^2$$

where $\tilde{\mu}_t$ represents the predicted mean of the diffusion process at time step $t$, which depends on both the input, respectively the clean sample $\mathbf{x}_t$ and the adversarial one $\mathbf{x}'_t$, and the original sample

$\mathbf{x}_0$. The optimization goal is to maximize the discrepancy between the predicted means of the clean and adversarial inputs, ensuring that the perturbation effectively disrupts the reverse diffusion process. This cost function, if considered in light of the model's prediction in the $\epsilon$-prediction setting, can be formulated as:

$$\mathcal{J}_\theta(\mathbf{x}_t, \boldsymbol{\delta}, t) = \left\| \boldsymbol{\epsilon}_\theta(\mathbf{x}_t + \boldsymbol{\delta}, t) - \boldsymbol{\epsilon}_\theta(\mathbf{x}_t, t) \right\|_2^2$$

To compute the adversarial perturbation $\boldsymbol{\delta}$, the gradient of the loss $\mathcal{J}_\theta$ with respect to $\mathbf{x}_t'$ is used:

$$\boldsymbol{\delta} = \sigma(t) \cdot \text{sign}\left( \nabla_{\mathbf{x}_t} \mathcal{J}_\theta(\mathbf{x}_t, \boldsymbol{\delta}, t) \right),$$

where $\sigma(t)$ scales the perturbation to ensure it adheres to the variance of the Gaussian noise in the reverse diffusion process at the $t$-th step. This step aligns the adversarial attack with the stochastic nature of the model, ensuring the perturbation remains consistent with the Gaussian hypothesis.

The final adversarial example is then obtained as:

$$\mathbf{x}_t^{\text{adv}} = \mathbf{x}_t + \boldsymbol{\delta}.$$

The adversarially perturbed sample $\mathbf{x}_t^{\text{adv}}$ is fed back into the reverse diffusion process, following the recurrence of the inference.

### A.3 ITERATIVE ATTACK

In Algorithm 2, we described the attack version that applies a single-step attack procedure applied to each and every inference timestep. In this section, we propose a multi-step attack approach based on the PGD iterative attack that, similarly to what was described in the previous algorithm, aims to attack model generation at the timestep level. We again highlight that this attack is not intended as a practical attack proposed in this paper; the main aim of showing this attack approach is to provide a procedure to assess the abilities of the DM to be resilient against minor perturbations applied to every sampling iteration. In Algorithm 3 we propose the multi-step approach, implemented by applying at every iteration the PGD-20 attack. In this case, being the attack iterative, it is necessary to project at

---

**Algorithm 3** Adversarial Attack on a Diffusion Model.

---

**Input:** percentage of attacked timesteps $p$, total timesteps $T$, model $\boldsymbol{\epsilon}_\theta$, scheduler values $\alpha_t$ and $\sigma(t)$, perturbation strength $\phi$, iterations $\mathbf{N}$, the projection operator $\mathbb{P}$
$\mathbf{x}_T \sim \mathcal{N}(0, I)$
**for** t = $T$ to 0 **do**
 $\sigma(t) \leftarrow \exp\left(\frac{1}{2} \log \sigma_t^2\right)$
 $\mathbf{x}_{t-1} \leftarrow \boldsymbol{\epsilon}_\theta(\mathbf{x}_t, t)$
 $\hat{\mathbf{x}}_0 \leftarrow \frac{\mathbf{x}_t - \sqrt{1 - \bar{\alpha}_t} \boldsymbol{\epsilon}_\theta(\mathbf{x}_t, t)}{\sqrt{\bar{\alpha}_t}}$
 $\tilde{\boldsymbol{\mu}}_t(\mathbf{x}_t, \hat{\mathbf{x}}_0) \leftarrow \frac{\sqrt{\bar{\alpha}_{t-1}} \sigma(t)}{1 - \bar{\alpha}_t} \hat{\mathbf{x}}_0 + \frac{\sqrt{\alpha_t}(1 - \bar{\alpha}_{t-1})}{1 - \bar{\alpha}_t} \mathbf{x}_t$
 $\boldsymbol{\delta}_0 \sim \mathcal{N}(0, \phi^2 \sigma^2(t) I)$
 **for** n = 0 to $N - 1$ **do**
  $\mathbf{x}_{t,n}' = \mathbf{x}_t + \boldsymbol{\delta}_n;$
  $\mathbf{x}_{t-1,n}' \leftarrow \boldsymbol{\epsilon}_\theta(\mathbf{x}_{t,n}', t)$
  $\hat{\mathbf{x}}_{0,n}' \leftarrow \frac{\mathbf{x}_{t,n}' - \sqrt{1 - \bar{\alpha}_t} \boldsymbol{\epsilon}_\theta(\mathbf{x}_{t,n}', t)}{\sqrt{\bar{\alpha}_t}}$
  $L = \left\| \tilde{\boldsymbol{\mu}}_t(\mathbf{x}_t, \hat{\mathbf{x}}_0) - \tilde{\boldsymbol{\mu}}_t(\mathbf{x}_{t,n}', \hat{\mathbf{x}}_{0,n}') \right\|_2^2$
  $\boldsymbol{\delta}_{n+1} = \sigma(t)/N \cdot \text{sign}(\nabla_{\mathbf{x}_t} L)$
 **end for**
 $\boldsymbol{\delta} = \mathbb{P}(\boldsymbol{\delta}, -\sigma(t), \sigma(t))$
 $\mathbf{x}_t^{\text{adv}} = \mathbf{x}_t + \boldsymbol{\delta}$
 Sample $\zeta \sim \mathcal{N}(0, I)$
 $\mathbf{x}_{t-1} \leftarrow \boldsymbol{\epsilon}_\theta(\mathbf{x}_t^{\text{adv}}, t) + \mathbf{1}_{t>0} \, \sigma(t) \, \zeta$
**end for**

---

the end the perturbation in order to keep its values within the range $[-\sigma(t), \sigma(t)]$. These values have been chosen following the Gaussianity hypothesis of the intermediate MC states. Diffusion models model intermediate data through intermediate Gaussian distributions where the possible values would

have standard deviation $\sigma(t)$. In order not to diverge too much from data distribution and be in a suitable range of possible values, we decided to impose as a ray of the projection interval the same standard deviation, making it also adaptive to the considered timestep. The table Section 3.4 shows model performance under this PGD-like version of a diffusion model attack.

## A.4 EOT ATTACK

In this section, we present a PGD-based implementation of the Expectation over Transformation attack (EoT). We build on top of the previously introduced PGD attack (Algorithm 3) in order to define an EoT adapted version that would include stochasticity into the optimization of the adversarial noise. We define $e$ as the parameter setting the maximum number of samples to approximate the expectation. The implementation adopts the same outline as the one adopted by the PGD attack; as a

---

**Algorithm 4** EoT Adversarial Attack on a Diffusion Model.

**Input:** percentage of attacked timesteps $p$, total timesteps $T$, model $\epsilon_\theta$, scheduler values $\alpha_t$ and $\sigma(t)$, perturbation strength $\phi$, PGD iterations $\mathbf{N}$, the projection operator $\mathbb{P}$, EoT iterations $e$

$\mathbf{x}_T \sim \mathcal{N}(0, I)$

**for** $t = T$ to $0$ **do**

$\quad \boldsymbol{\delta}_0 \sim \mathcal{N}(0, \phi^2 \sigma^2(t) I)\,, \xi \sim \mathcal{N}(0, I)$

$\quad \mathbf{x}_{t-1} = \frac{1}{\sqrt{\alpha_t}} \left( \mathbf{x}_t - \frac{\sigma(t)}{\sqrt{1-\bar{\alpha}_t}} \epsilon_\theta(\mathbf{x}_t, t) \right) + \sigma_t \xi$

$\quad \boldsymbol{\delta} \sim \mathcal{N}(0, \phi^2 \sigma^2(t) I)$

$\quad$ **for** $n = 0$ to $\mathbf{N} - 1$ **do**

$\quad\quad \mathcal{G} \leftarrow [\,]$

$\quad\quad \mathbf{x}_t^{adv} = \mathbf{x}_t + \sqrt{\alpha_t} \boldsymbol{\delta}_n$

$\quad\quad$ **for** $i = 1$ to $e$ **do**

$\quad\quad\quad \zeta \sim \mathcal{N}(0, I)$

$\quad\quad\quad \mathbf{x}_{t-1}^{adv} = \frac{1}{\sqrt{\alpha_t}} \left( \mathbf{x}_t^{adv} - \frac{\sigma(t)}{\sqrt{1-\bar{\alpha}_t}} \epsilon_\theta(\mathbf{x}_t^{adv}, t) \right) + \sigma_t \zeta$

$\quad\quad\quad g_i \leftarrow \nabla_{\boldsymbol{\delta}_n} \left\| \mathbf{x}_t - \mathbf{x}_t^{adv} \right\|_2^2$

$\quad\quad\quad$ Append $g_i$ to $\mathcal{G}$

$\quad\quad$ **end for**

$\quad\quad \bar{g} \leftarrow \frac{1}{e} \sum_{g \in \mathcal{G}} g$

$\quad\quad \boldsymbol{\delta}_{n+1} \leftarrow \mathbb{P}(\boldsymbol{\delta}_n + \sigma(t)/N \cdot \text{sign}(\bar{g}), -\sigma(t), \sigma(t))$

$\quad$ **end for**

$\quad \boldsymbol{\delta} \leftarrow \boldsymbol{\delta}_{n+1}$

$\quad \mathbf{x}_{t-1} = \frac{1}{\sqrt{\alpha_t}} \left( \mathbf{x}_t + \sqrt{\alpha_t} \boldsymbol{\delta} - \frac{\sigma(t)}{\sqrt{1-\bar{\alpha}_t}} \epsilon_\theta(\mathbf{x}_t + \sqrt{\alpha_t} \boldsymbol{\delta}, t) \right) + \sigma_t \xi$

**end for**

---

consequence, all the previously described details about the attack notation still hold.

## A.5 RELATIONSHIP WITH SMOOTH DIFFUSION (GUO ET AL., 2024)

The method by Guo et al. (2024) also introduces a method for smoothing the latent space of Diffusion Models. While both papers utilize the term "smoothness", we emphasize that the underlying concept of smoothness, its enforcement mechanism, and our primary objectives fundamentally differ. In particular:

⋄ Guo et al. (2024) did not demonstrate that their optimization could inherently include being resilient to attacks, focusing more on smooth generation, interpolation, and inverse problems

⋄ Guo et al. (2024) does not mention any robustness to adversarial attacks

⋄ Guo et al. (2024) does not claim that it could be used to train with corrupted data

Table 3 reports the main differences between the two approaches.

| Component | Smooth Diffusion | **Ours** |
|---|---|---|
| Arch/Training | Stable.Diff. + LoRA | UNet with attention + scratch |
| Equation | $\left\|\left\|\nabla_\epsilon\left(\sqrt{1-\overline{\alpha_t}}\,\widehat{\boldsymbol{x}_0}(\epsilon)\cdot\Delta\widehat{\boldsymbol{x}_0}\right)\right\|\right\|_2$ | $\left\|\left\|\epsilon_\theta\left(\mathbf{x}t^{\text{adv}},t\right)-\left[\epsilon_\theta\left(\mathbf{x}_t,t\right)+\boldsymbol{\delta}\right]\right\|\right\|_2$ |
| Objective | Reduce gradient norm | Equivariance |
| Perturbation | normally sampled pix. int. normalized to unit length | adversarial under $\ell_\infty$ |
| Benefit | Smooth Latent | Resilient to adv. attacks |
| Benefit | Image inversion | Train on corrupted data |
| Benefit | Stable Interpolation | Faster sampling |

Table 3: Differences between ours methods and Guo et al. (2024)

## B    SUPPLEMENTARY MATERIAL

This supplementary material is intended to complement the main paper by providing further motivation for our assumptions and design choices, as well as additional ablation studies on the proposed datasets to demonstrate the effectiveness of our method. It is organized into the following sections.

Section C discusses the main differences among the considered approaches, offering a deeper analysis that includes both geometrical and empirical motivations behind the adopted design choices. It also clarifies the distinction between invariance and equivariance, and presents statistics on the adversarial perturbation $\delta$; Section D presents a more detailed analysis of the diffusion flow dynamics by examining the trajectories obtained from low-dimensional datasets under different conditions. Section E provides an extensive qualitative ablation across the real-world datasets introduced in the paper, showcasing a wide variety of samples and comparisons; Section F offers additional observations and insights into the proposed approach. **We encourage readers to zoom in and compare the results for a better understanding of their quality**.

## C    OBSERVATIONS AND MOTIVATIONS ON OUR ADVERSARIAL TRAINING FRAMEWORK

### C.1    EQUIVARIANT AND INVARIANT FUNCTIONS FOR ADVERSARIAL TRAINING

Adversarial training in classification has been widely studied over years Goodfellow et al. (2015); Madry et al. (2018); Zhang et al. (2019); Wang et al. (2020); Shafahi et al. (2019); Wong et al. (2020); Sriramanan et al. (2021; 2020); Wang et al. (2023); Zhu et al. (2021); Mujtaba Hussain et al. (2024) in different settings, threat models and under different perspectives. These methods share the objective to enforce invariance in the neural network $f_\theta$, since the final objective is to enforce the output of the network not to vary in the presence of minor changes in the network input. However, in generative modeling, particularly diffusion models (DMs), enforcing invariance hinders learning the correct distribution, making the model unable to take into account input changes in its prediction. Ignoring the adversarial perturbations applied during a perturbed training leads to deviations in trajectories, resulting in an inaccurate learned distribution. Conversely, training the model to incorporate the negative of the perturbation helps it recognize and manage potential deviations, enabling it to handle noise with broader standard deviations more effectively. In Nguyen & Raff (2019), the authors extend the concept of adversarial attacks to regression tasks, even though considering regression tasks on tabular datasets. Their proposed method addresses these attacks by introducing an adversarial training loss based on numerical stability, improving performance under adversarial conditions. Even though the latter bridges the concepts of regression and AT, an analysis of implications in the case of randomized and adversarial training applied to the generative model is still a topic to cover, particularly with reference to generative models. In this spirit, we propose a new training framework inspired by AT with the aim of shedding light on the concept of adversarial training for DMs, exploiting knowledge from both functional analysis and classification neural networks.

Formally defining the two properties, we can define both invariance and equivariance. Given a function $f : X \to Y$, as well as a specified group actions $A$, $f$ is said to be *equivariant* with respect to a transform $a \in A$ if and only if

$$f(a \circ x) = a \circ f(x), \quad x \in X \tag{12}$$

Given a function $f : X \to Y$, as well as a specified group action $A$, $f$ is said to be *invariant* with respect to an $a \in A$ transform if and only if

$$f(a \circ x) = f(x), \quad x \in X \tag{13}$$

In Fig. 9 we extend Fig.1 of the paper and depict what happens at the trajectory level if we enforce invariance instead of equivariance. The vector $\epsilon$ represents the noise that is added by the diffusion process, $\delta$ represents the added noise by the adversarial training. Finally, we will have two different versions of the noisy point, namely $\mathbf{x}_t$ and $\mathbf{x}_t + \delta$. The model, if unattacked, would like to regress a portion of noise equivalent to $-\epsilon$ so that it is able to correctly go back to $\mathbf{x}_0$. When applying $\delta$, the network's objective still has to be the same. The figure shows that if, given the noisy staring point $\mathbf{x}_t + \delta$, the model is enforced to learn again $-\epsilon$, so if the invariance is applied, the ending point would be some other point in the space different wrt. $\mathbf{x}_0$. On the contrary, if equivariance is applied, the network is forced to regress $-(\epsilon + \delta)$, making the model able to correctly regress $\mathbf{x}_0$.

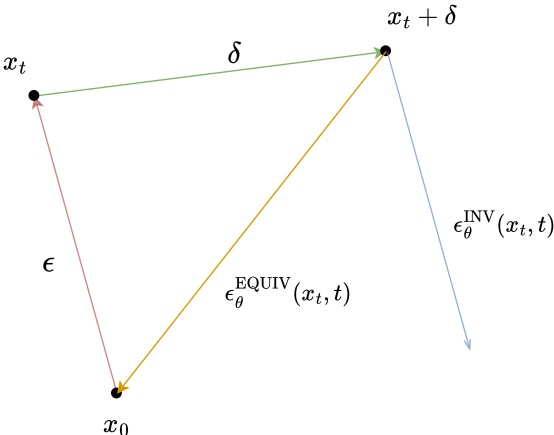

Figure 9: Not applying equivariance $\epsilon_\theta^{\text{EQUIV}}(\mathbf{x}_t, t)$, the model drifts and ends up in a different point of the space than the desired one, learning then the perturbation that we added as in $\epsilon_\theta^{\text{INV}\cdot}(\mathbf{x}_t, t)$

## C.2 INVARIANCE REGULARIZATION DOES NOT WORK

As empirical evidence of the inconsistency of invariance training in AT for DMs, we prove it on low-dimensional data. As a proof-of-concept, we consider the `oblique-plane` 3D dataset as data to train on, and then we impose adversarial training, following the same setting as in Algorithm 1, enforcing instead invariance by minimizing the loss function:

$$\mathcal{L}_{\text{AT}}(\mathbf{x}_t, \mathbf{x}_t^{\text{adv}}, t, \boldsymbol{\epsilon}) = \arg\min_\theta \underbrace{\left\| \boldsymbol{\epsilon}_\theta(\mathbf{x}_t, t) - \boldsymbol{\epsilon} \right\|_2^2}_{\mathcal{L}_{\text{DM}} \text{ to fit data distr.}} + \lambda_t \underbrace{\left\| \boldsymbol{\epsilon}_\theta(\mathbf{x}_t^{\text{adv}}, t) - [\boldsymbol{\epsilon}_\theta(\mathbf{x}_t, t)] \right\|_2^2}_{\mathcal{L}_{\text{reg}} \text{ to enforce invariance}} \tag{14}$$

We decided to implement this example on 3D data in order to have the possibility of observing the behavior of 3D trajectories. The plot shows it displaying side-to-side DDPM Ho et al. (2020) and invariance in the same data settings as the one displayed in Fig. 2. The model, by enforcing invariance, loses the ability to correctly reconstruct the data manifold, not being able to generate points in the data distribution, whereas the model trained through standard DDPM learns the data distribution but still suffers from learning the noise in case of noisy data. The same behavior can be observed when looking at the trajectories. This analysis clarifies even more what is the generation dynamics. The model creates sparse trajectories that do not tend to be clustered, neither at the beginning of the generation nor at the end, thereby causing generated samples to be completely off the data subspace.

A comparison between the plots in Fig. 10 and those in Fig. 11 further emphasizes the benefits of adversarial training with equivariance. The contrast shown in the compared trajectories clearly illustrates how our approach consistently produces trajectories that are more clustered, sharper, and better aligned with the underlying data manifold, thereby reinforcing the inadequacy of conventional adversarial training methods.

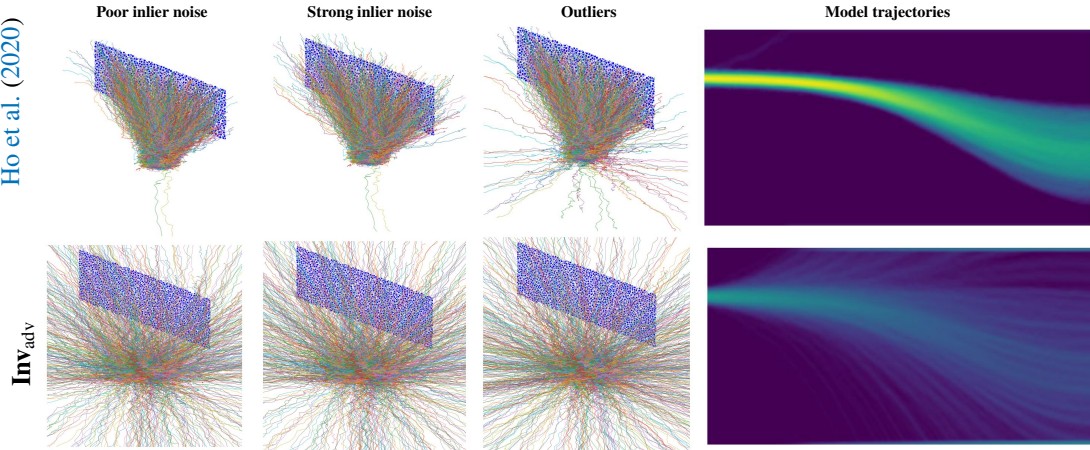

Figure 10: The application of invariance on 3D data highlights the incorrect behavior of the training procedure: the learnt data distribution is completely different from the reference one.

### C.3 NOTES ON DEFINITION OF THE ADVERSARIAL PERTURBATION

One of the main points of our work is defining a suitable perturbation $\boldsymbol{\delta}$ for unconditional diffusion models that aims at disrupting generation trajectories without relying on acting on the model's inputs. In order to craft this kind of attack, we focused on exploiting generation dynamics in order to correctly perturb it at each of its steps. Inspired by adversarial attacks with random start (such as R-FGSM Wong et al. (2020)), $\boldsymbol{\delta}$ is first initialized by randomly sampling from a uniform distribution, whose bounds are $[-r_\beta(\cdot), r_\beta(\cdot)]$. The initialization distribution is chosen to be a uniform distribution with varying bounds but always centered at zero. This choice ensures that the perturbation has zero mean, which is essential when applied within the diffusion process. A non-zero mean would not only bias the estimation of the noise but also violate the Gaussian transition assumption, which requires the noise to be zero-centered. The parameter $\beta$ is sampled from uniform distribution as follows $\beta \sim \mathcal{U}[0.5, 2]$. Its aim is to enhance the model's robustness to trajectory deviations by randomly varying the perturbation's bounds. Once the perturbation bounds are defined, it is straightforward to calculate the standard deviation of the initialization distribution. The mean value of $\boldsymbol{\delta}$ is given by:

$$\mathbb{E}[\boldsymbol{\delta}] = \frac{[(-r_\beta(\cdot)) + (r_\beta(\cdot))]}{2} = 0.$$

The variance of the distribution is defined as:

$$\mathbb{VAR}[\boldsymbol{\delta}] = \frac{[(-r_\beta(\cdot)) + (r_\beta(\cdot))]^2}{12} = \frac{(2r_\beta(\cdot))^2}{12} = \frac{r_\beta(\cdot)^2}{3}$$

The variance is consequently defined as a $\beta$-dependent quantity as it is rescaled batch-wise by this parameter, assuring a random dynamic change of the perturbation bounds.

## D    COMPREHENSIVE ANALYSIS OF THE DIFFUSION FLOW DYNAMICS

In order to better understand data behavior during the generation procedure, we report in this section further trajectory plots. The plots can only be visualized if the data taken into account is low-dimensional in order to properly track points' behavior in the generation. We exploit the low-dimensional datasets proposed in the paper to further investigate trajectory behavior. Additional qualitative samples of the Diffusion Flow are shown in Fig. 11, supplementing Fig. 4 and Fig. 6, which can be found in the main paper. Unlike this one, here we have the chance to show also the difference between models' behavior when data distribution is affected by strong inlier noise and outliers on both unimodal distribution `oblique-plane` and `3-gaussians`.

The plot shows that even though the DDPM model reaches the distribution of the final part of the trajectories, those are sparse and, even in the case of inlier noise, they appear not to be densely clustered, with some completely diverging from the data distribution. When applying the regularization, particularly in this case, the model is trained with adversarial noise, the density increases in the trajectories, defining sharper and clustered paths, strongly discouraging significant deviations from their central modes. This feature is especially useful when the initial data distribution is noisy, as it helps the model avoid learning erroneous points that stray from the true data distribution, preventing it from capturing the noise present in the starting data. In particular, when outlier noise is present, regularization minimizes its influence, resulting in denser and sharper trajectories that better align with the true data distribution clusters.

## E    ADDITIONAL QUALITATIVE SAMPLES UNDER MULTIPLE SETTINGS

### E.1    TRAINED ON CLEAN CIFAR-10 WITH $p = 0\%$

**DDPM vs Robust$_{\mathrm{adv}}$.** In Fig. 12 of this supplementary material, we extend Fig. 4 in the paper and show 300 samples from DDPM vs 300 samples from **Robust$_{\mathrm{adv}}$**, both trained on the original dataset. Although our method has not been designed to work directly with uncorrupted data, the images that ours generates result in smooth images, the clutter in the background has been canceled, yet the objects and animals are still clearly recognizable, and part of the noise in the background of CIFAR-10 has been removed. We think that it is reasonable to justify the drop we have in the FID with our method denoising action, which is, for example, the removal of part of the characteristic background noise proper of CIFAR-10. This effect can be the reason for the evaluation penalizing us.

**500 vs 1000 steps.** We expand the current section by including some samples that focus on enriching the paper's discussion about faster sampling. In Fig. 13 we offer on the left the results by **Robust$_{\mathrm{adv}}$** with 1000 inference steps trained on uncorrupted data. On the right instead, we show the qualitative samples still with **Robust$_{\mathrm{adv}}$** yet using a scheduler with 500 inference steps, thereby cutting 50% of the inference time. *Surprisingly, the faster sampling yields better FID. We get 28.68 FID with 1000 steps and 24.34 with 500 steps.* In terms of differences, taking more steps generates images with warmer and natural colors, whereas taking fewer steps seems to improve the details of the objects, and the colors look brighter and saturated, probably being closer to the actual CIFAR-10 images.

### E.2    TRAINED ON NOISY CIFAR-10 WITH $p = 90\%$, $\sigma = 0.1$

In Fig. 14 of this supplementary material, we provide additional figures not present in the main paper. The figures show 300 samples from DDPM vs **Robust$_{\mathrm{adv}}$** both trained on noisy CIFAR-10 with $p = 90\%$, $\sigma = 0.1$. The images that ours generates (right) are smooth, similar to the one in Fig. 12, inheriting the smoothing effect of the previous setting. In this case, the smoothing action helps absorb the Gaussian noise present in the dataset. This results in improved performance: unlike DDPM (left), ours is able to unlearn the noise and keep images still with natural colors.

### E.3    TRAINED ON NOISY CIFAR-10 WITH $p = 90\%$, $\sigma = 0.2$

In Fig. 15 of this supplementary material we extend Fig. 8 of the paper, enriching it with 300 more samples per method yet trained on noisy CIFAR-10 with $p = 90\%$, $\sigma = 0.2$. Looking at the standard deviation of the added noise, in this case $\sigma = 0.2$ represents a very strong one: it means we are adding 40% of the variability that is naturally present in CIFAR-10, being $\sigma_{\mathrm{data}} = 0.5$. Despite the

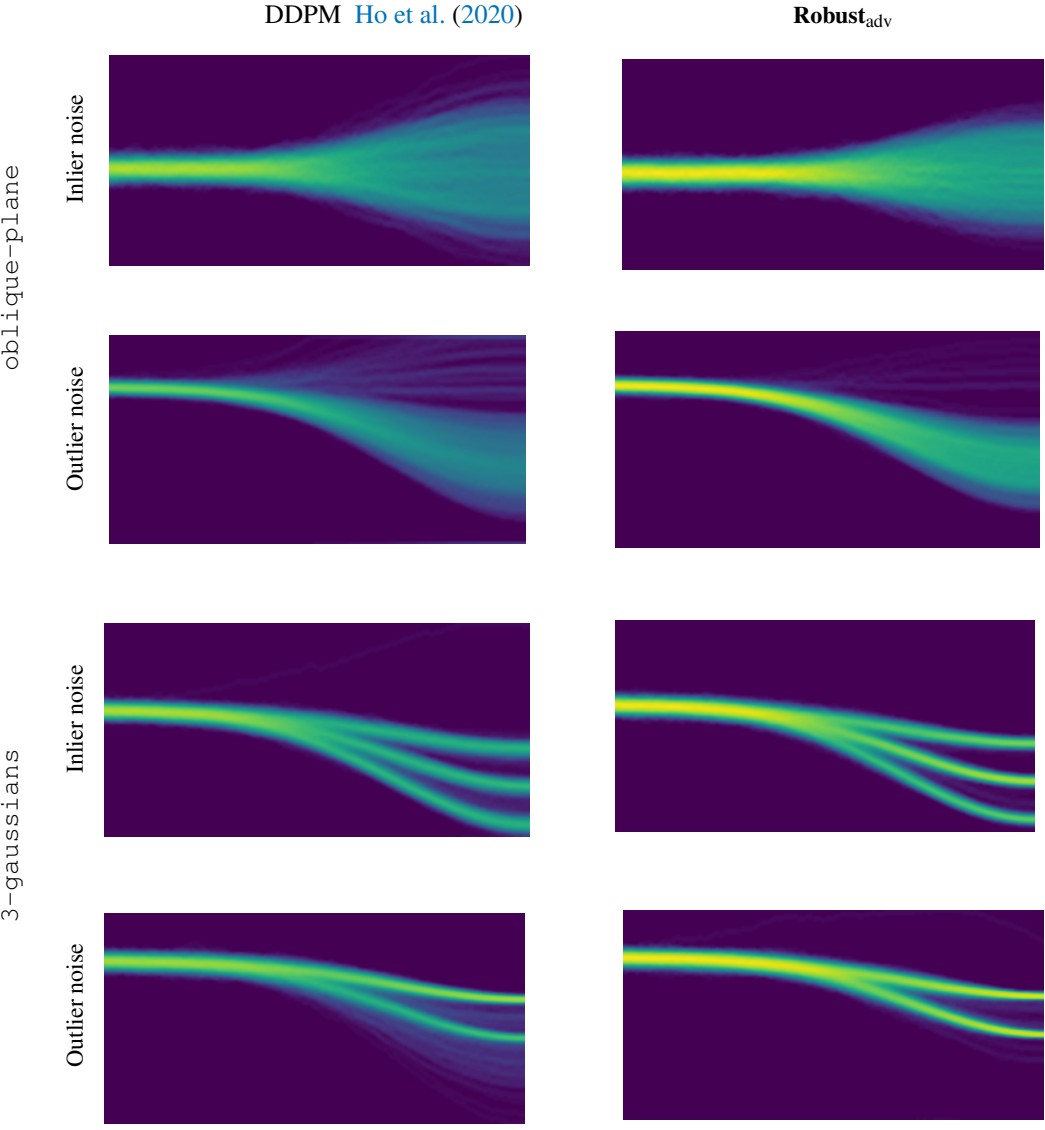

Figure 11: Diffusion flow: DMs vs **Robust**adv. Left column shows the results by Ho et al. (2020) under two different types of noise. Regular training tends to incorporate the noise inside the diffusion flow, making it more prone to generate undesirable and unexpected results; Right column is **Robust**adv that trades off variability for resilience. Indeed, heatmaps on the right are more concentrated, clear, and less faded.

clean CIFAR-10 with $p = 0\%$

DDPM Ho et al. (2020)                    **Robust**adv

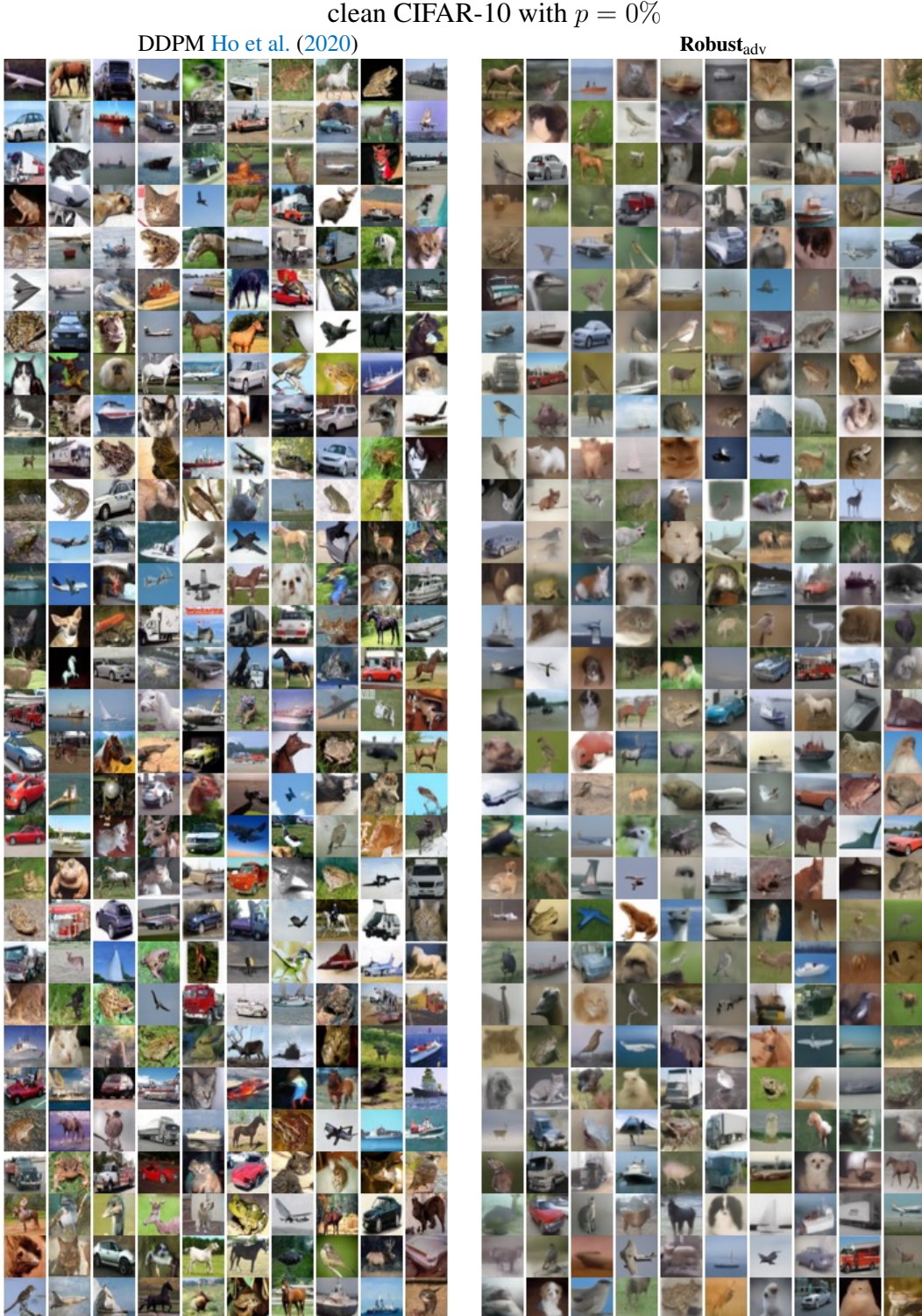

Figure 12: Trained on clean CIFAR-10 with $p = 0\%$. Despite the FID decreases once trained on clean data, generated images by **Robust**adv look smooth, and the clutter in the background has been canceled.

clean CIFAR-10 with $p = 0\%$

**Robust**adv w/ 1000 steps (28.68 FID ↓)          **Robust**adv w/ 500 steps (**24.34** FID ↓)

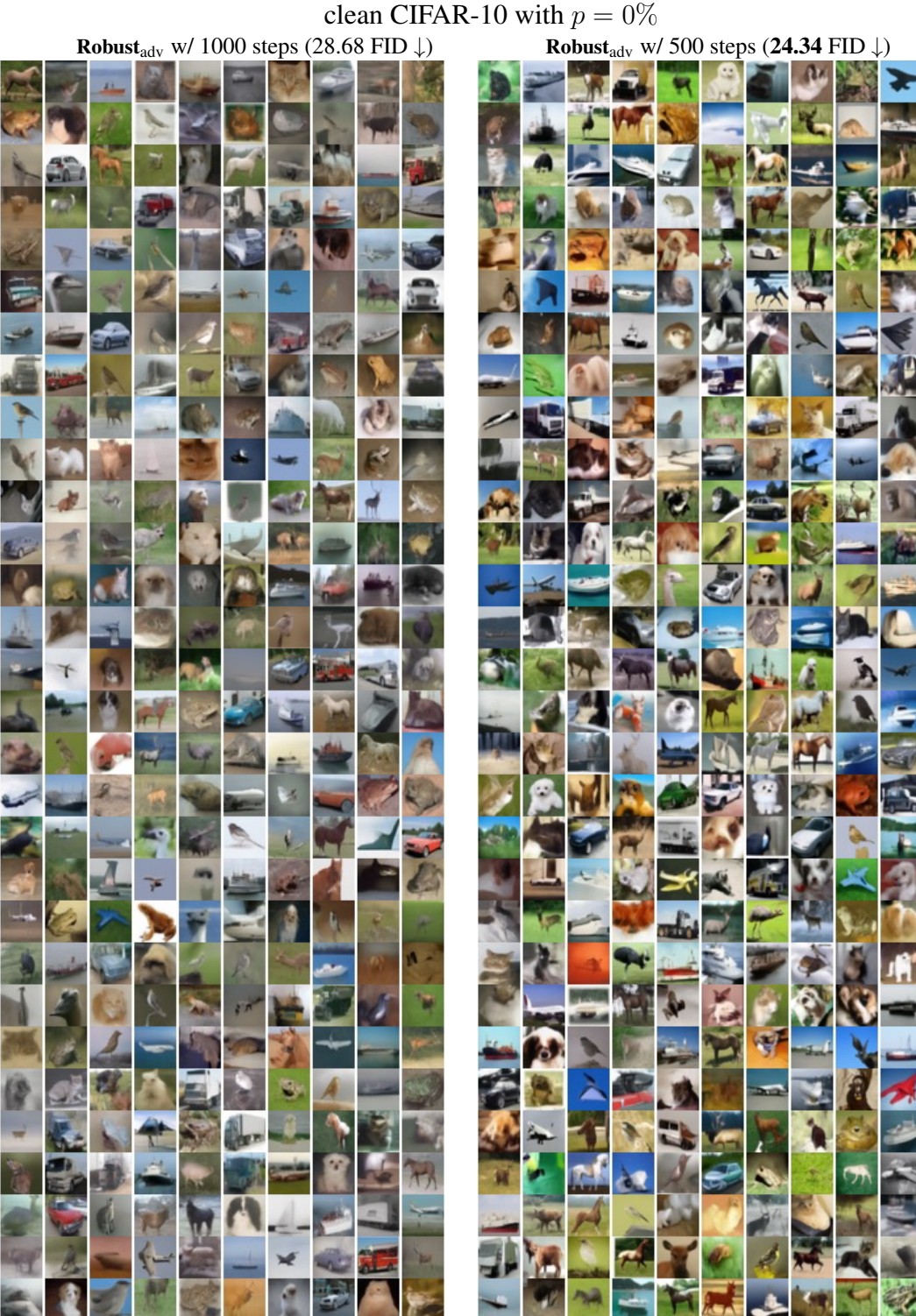

Figure 13: Trained on clean CIFAR-10 with $p = 0\%$ but comparing less steps (500) vs the default DDPM scheduler used for training (1000). Although we run **Robust**adv with a scheduler with fewer steps (500) and do not use it in training, the images on the right with 500 steps have better FID than with the original scheduler on the left.

noisy CIFAR-10 with $p = 90\%$, $\sigma = 0.1$

DDPM Ho et al. (2020)                    **Robust**$_{\text{adv}}$

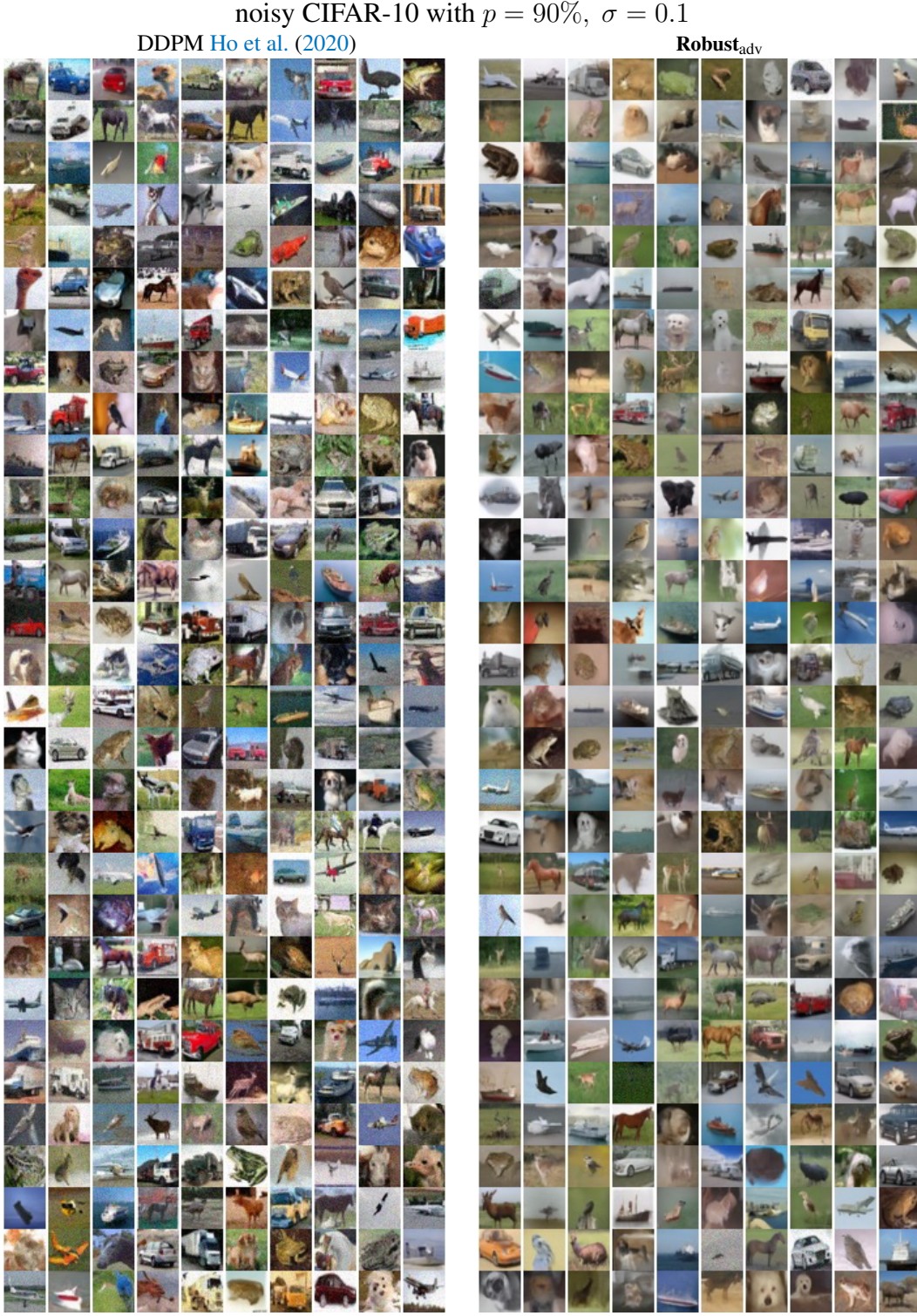

Figure 14: Trained on noisy CIFAR-10 with $p = 90\%$, $\sigma = 0.1$. Despite added noise, **Robust**$_{\text{adv}}$ images look smooth, and the clutter in the background has been canceled along with the Gaussian noise added. Instea,d DDPM on the left propagates the noise back in the output.

strong ambient noise, the images that ours generates (*right*) are smooth similar to the one in Fig. 12 and presence of the strong Gaussian is very rare. Unlike DDPM (*left*), ours is able to unlearn the noise and keep images still with natural colors.

### E.4 TRAINED ON NOISY CELEB-A WITH $p = 90\%$, $\sigma = 0.1$

We provide a more extensive qualitative analysis on the dataset CelebA Li et al. (2019) in Fig. 16 of this supplementary material. To further motivate the denoising effect, we here show same 300 samples per method yet trained on noisy Celeb-A with $p = 90\%$, $\sigma = 0.1$. The faces that ours generates ($right$) are smooth, but now instead of absorbing the Gaussian noise present in the dataset, unlike DDPM ($left$), ours is able to unlearn the noise and keep images still with natural colors.

### E.5 TRAINED ON NOISY CELEB-A WITH $p = 90\%$, $\sigma = 0.2$

We provide a more extensive qualitative analysis on the dataset CelebA Li et al. (2019) in Fig. 17 of this supplementary material. To further motivate the denoising effect, we here show same 300 samples per method yet trained on noisy Celeb-A with $p = 90\%$, $\sigma = 0.2$. The faces that ours generates ($right$) are smooth but now instead of absorbing the Gaussian noise present in the dataset, unlike DDPM ($left$), ours is able to unlearn the noise and keep images still with natural colors.

### E.6 TRAINED ON NOISY LSUN BEDROOM WITH $p = 90\%$, $\sigma = 0.1$

We provide a more extensive qualitative analysis on the dataset LSUN Bedroom Yu et al. (2015) in Fig. 19 of this supplementary material. To further motivate the denoising effect, we here show same 150 samples per method yet trained on noisy LSUN dataset with $p = 90\%$, $\sigma = 0.1$. The generated images by **Robust**$_{adv}$ (*right*) result to be smoother wrt. to the datasets ones and the DDPM generated ones (*left*) ones, but the smoothing effect allows absorbing the Gaussian noise present in the dataset: unlike DDPM, ours is able to unlearn the noise and keep images still with natural colors.

### E.7 TRAINED ON NOISY LSUN BEDROOM WITH $p = 90\%$, $\sigma = 0.2$

We further enrich the qualitative ablation on the dataset LSUN Bedroom Yu et al. (2015) in Fig. 20 of this supplementary material. To further motivate the denoising effect, we here show the same 150 samples per method yet trained on the noisy LSUN dataset with $p = 90\%$, $\sigma = 0.2$. The generated images by **Robust**$_{adv}$ (*right*) result to be smoother wrt. to the datasets ones and the DDPM generated ones (*left*) ones, but the smoothing effect allows absorbing the Gaussian noise present in the dataset: unlike DDPM, ours is able to unlearn the noise and keep images still with natural colors.

noisy CIFAR-10 with $p = 90\%$, $\sigma = 0.2$

DDPM Ho et al. (2020)                                   **Robust**$_{\text{adv}}$

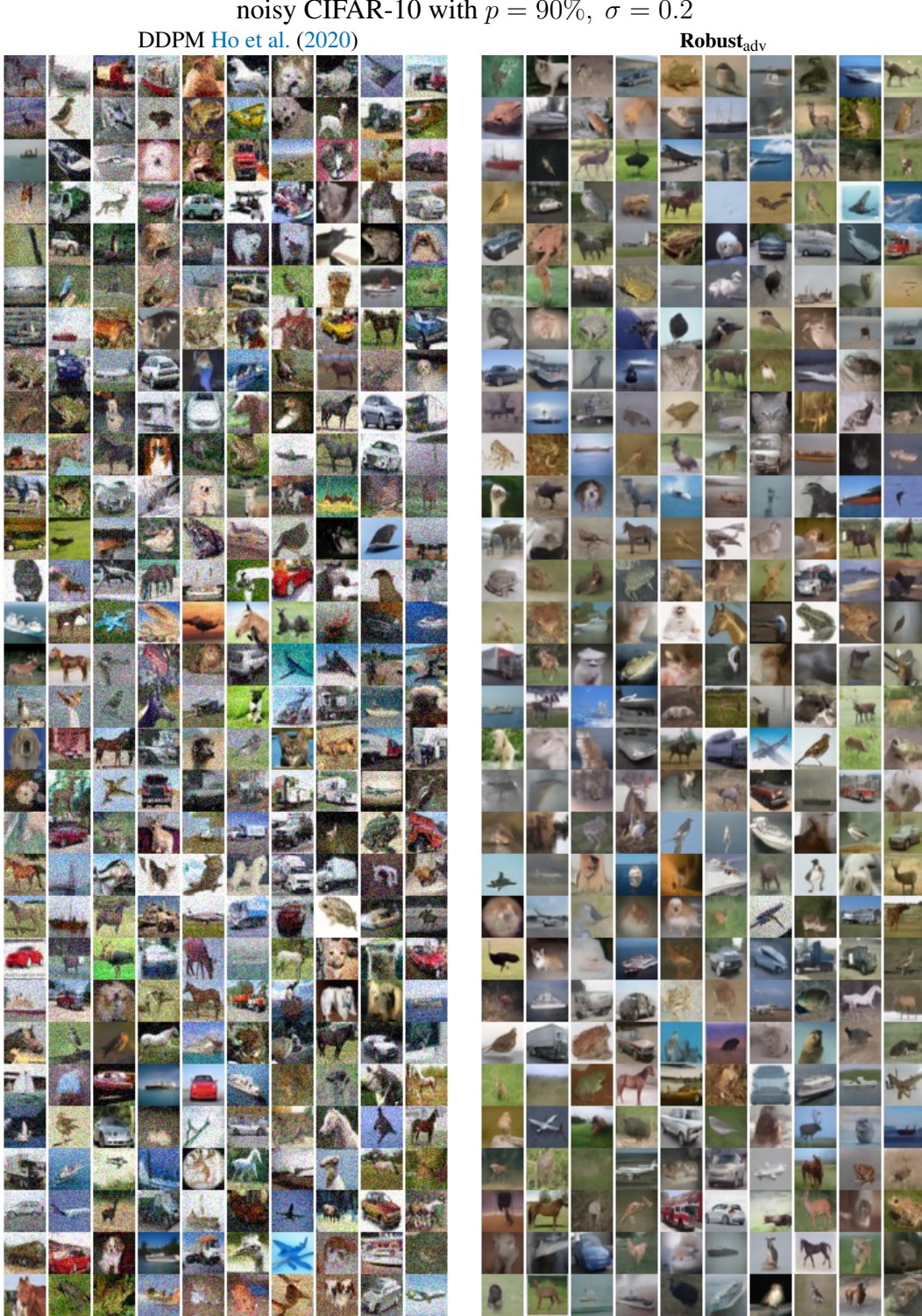

Figure 15: Trained on noisy CIFAR-10 with $p = 90\%$, $\sigma = 0.2$. Despite added noise, **Robust**$_{\text{adv}}$ images look smooth and the clutter in the background has been canceled along with the Gaussian noise added. Instead DDPM on the left propagates the noise back in the output.

noisy Celeb-A with $p = 90\%,\ \sigma = 0.1$

DDPM Ho et al. (2020)                                   **Robust**adv

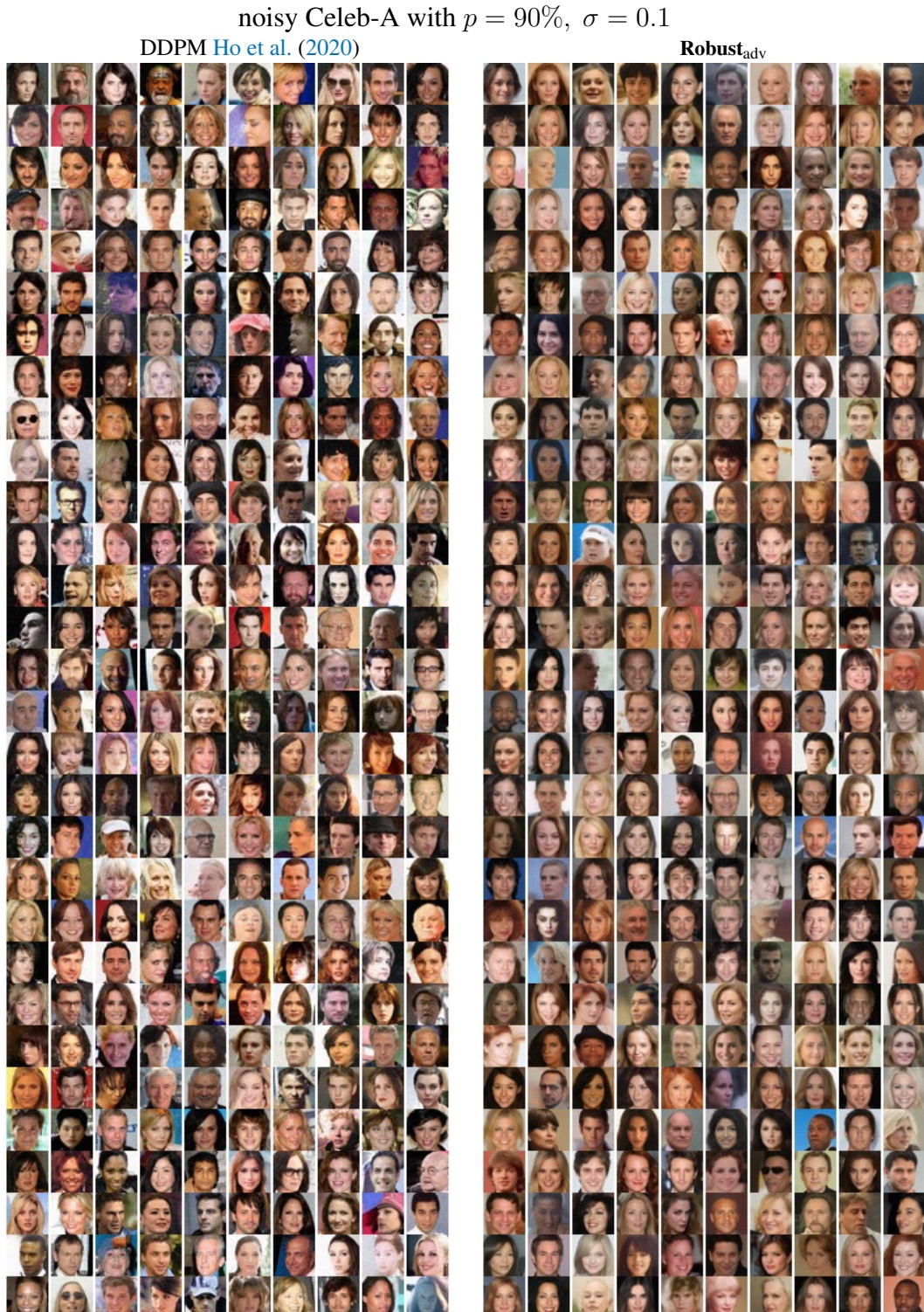

Figure 16: Trained on noisy Celeb-A with $p = 90\%,\ \sigma = 0.1$. Despite added noise, **Robust**adv faces look smooth and the clutter in the background has been canceled along with the Gaussian noise added. Instead, DDPM on the left propagates the noise back in the output.

noisy CelebA with $p = 90\%$, $\sigma = 0.2$

DDPM Ho et al. (2020)                    **Robust**adv

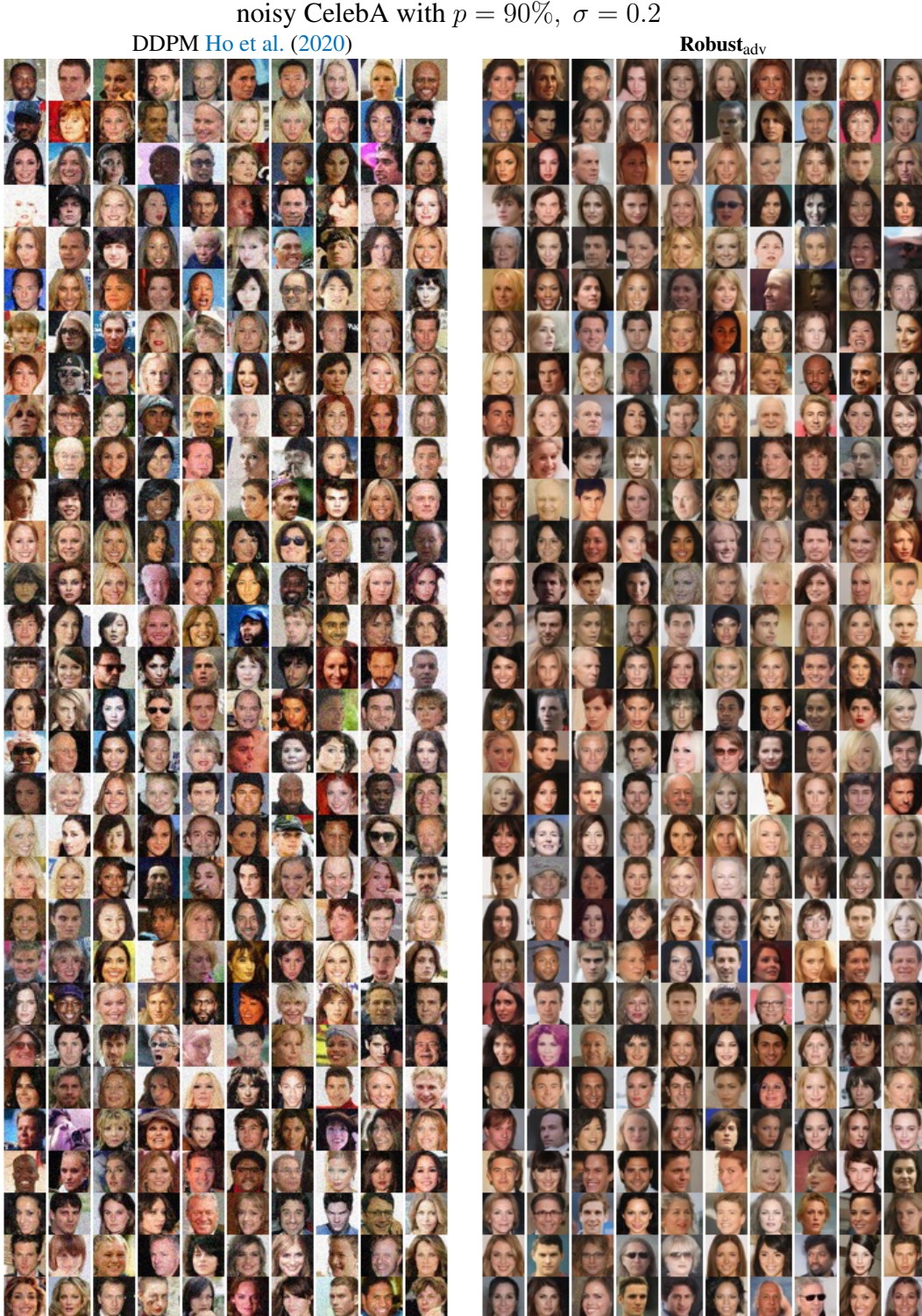

Figure 17: Trained on noisy Celeb-A with $p = 90\%$, $\sigma = 0.2$. Despite added noise, **Robust**adv faces look smooth, and the clutter in the background has been canceled along with the Gaussian noise added. Instead, DDPM on the left propagates the noise back in the output.

## LSUN Bedroom early stage training

DDPM Ho et al. (2020)    **Robust**adv

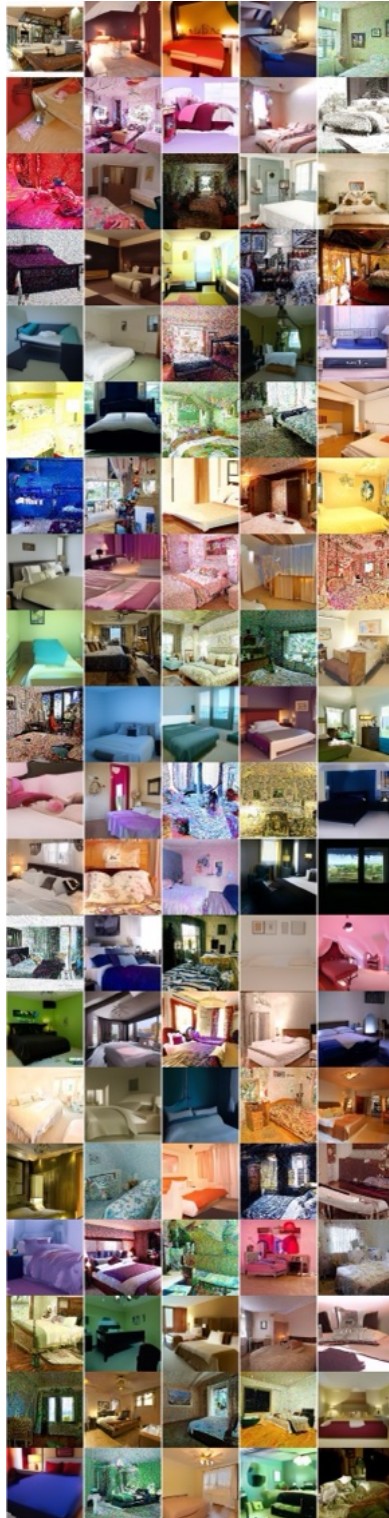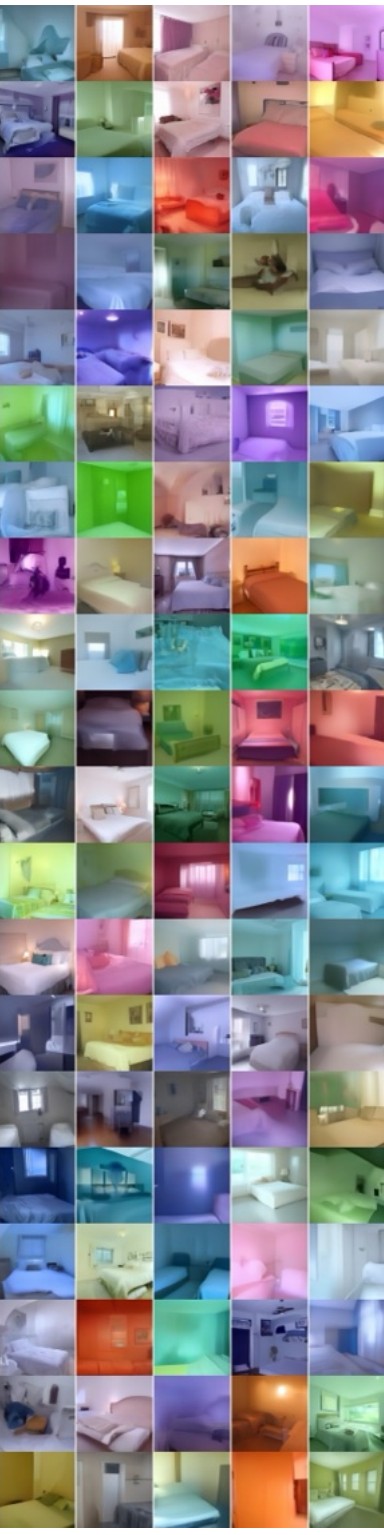

Figure 18: Trained on clean LSUN Bedroom. Despite the added noise, **Robust**adv produces images that appear smooth and exhibit fewer intricate details. When noise is absent from the training data, this smoothing effect results in the removal of fine-grained information from the learned distribution, ultimately reducing data variability.

LSUN Bedroom with $p = 90\%$, $\sigma = 0.1$, early stage training

DDPM Ho et al. (2020)                    **Robust**$_{\text{adv}}$

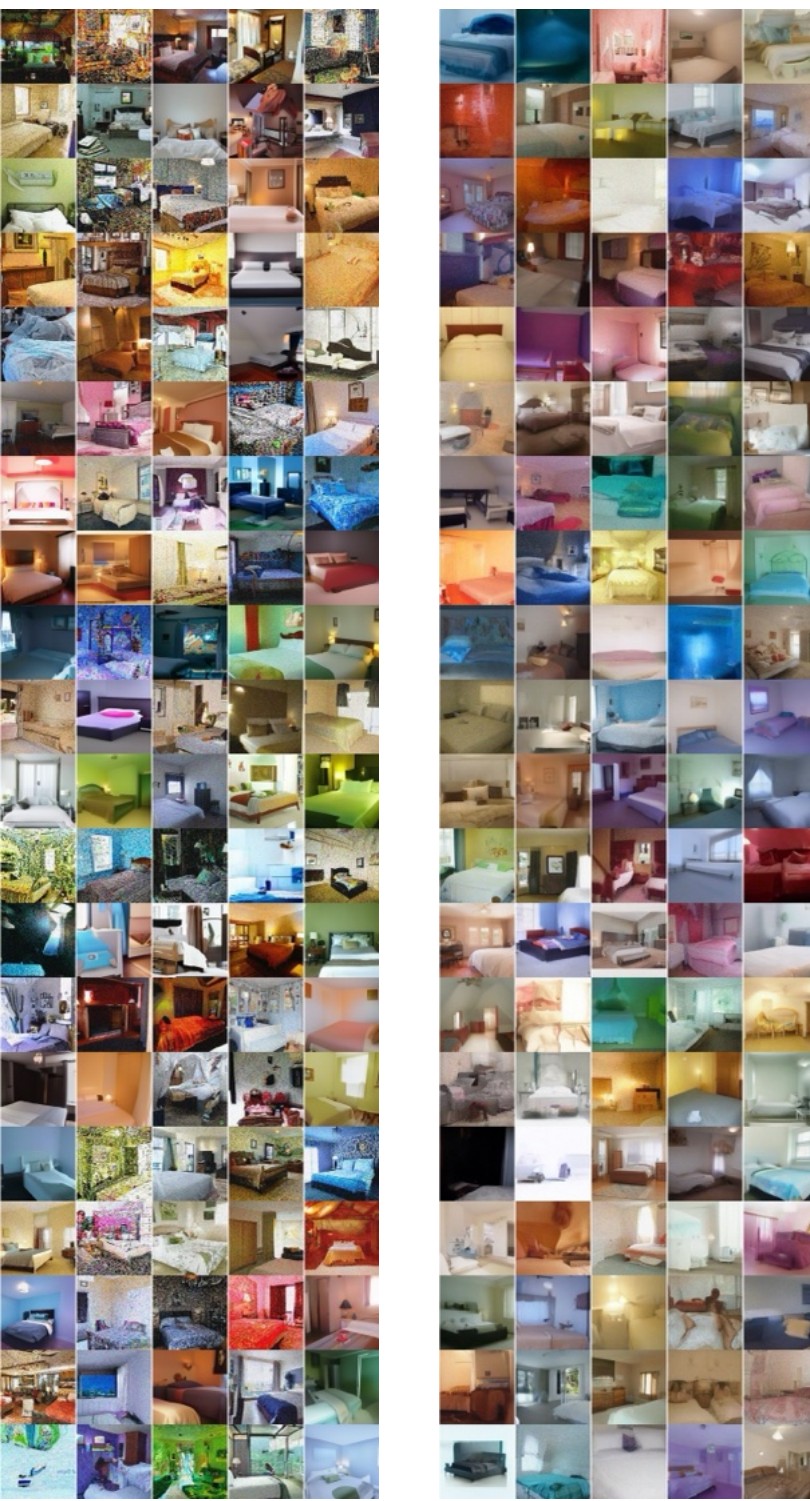

Figure 19: Trained on noisy LSUN Bedroom with $p = 90\%$, $\sigma = 0.1$. Despite added noise, **Robust**$_{\text{adv}}$ images look smooth and with fewer intricate details that have been canceled along with the Gaussian noise added. Instead, DDPM on the left propagates the noise back into the output.

LSUN Bedroom with $p = 90\%$, $\sigma = 0.2$, early stage training

DDPM Ho et al. (2020)                    **Robust**$_{\text{adv}}$

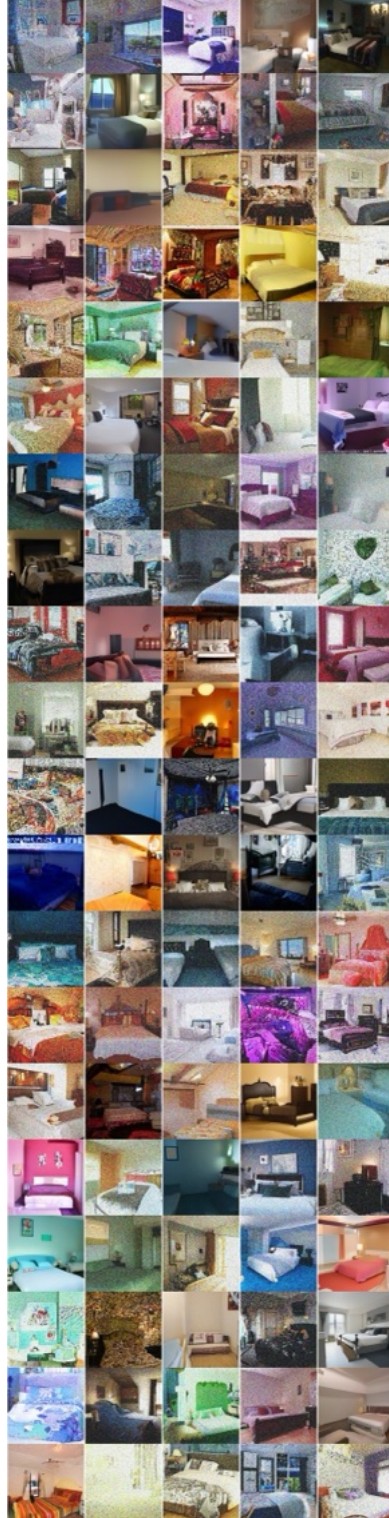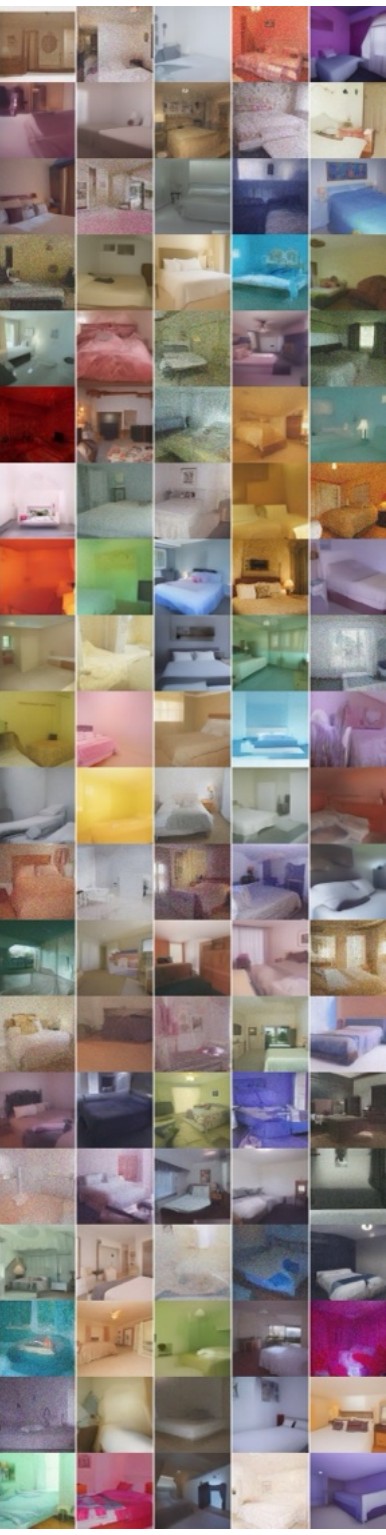

Figure 20: Trained on noisy LSUN Bedroom with $p = 90\%$, $\sigma = 0.2$. Despite added noise, **Robust**$_{\text{adv}}$ images look smooth and with less intricate details that have been canceled along with the Gaussian noise added. Instead, DDPM on the left propagates the noise back into the output.

LSUN Bedroom late training

DDPM Ho et al. (2020)                    **Robust**adv

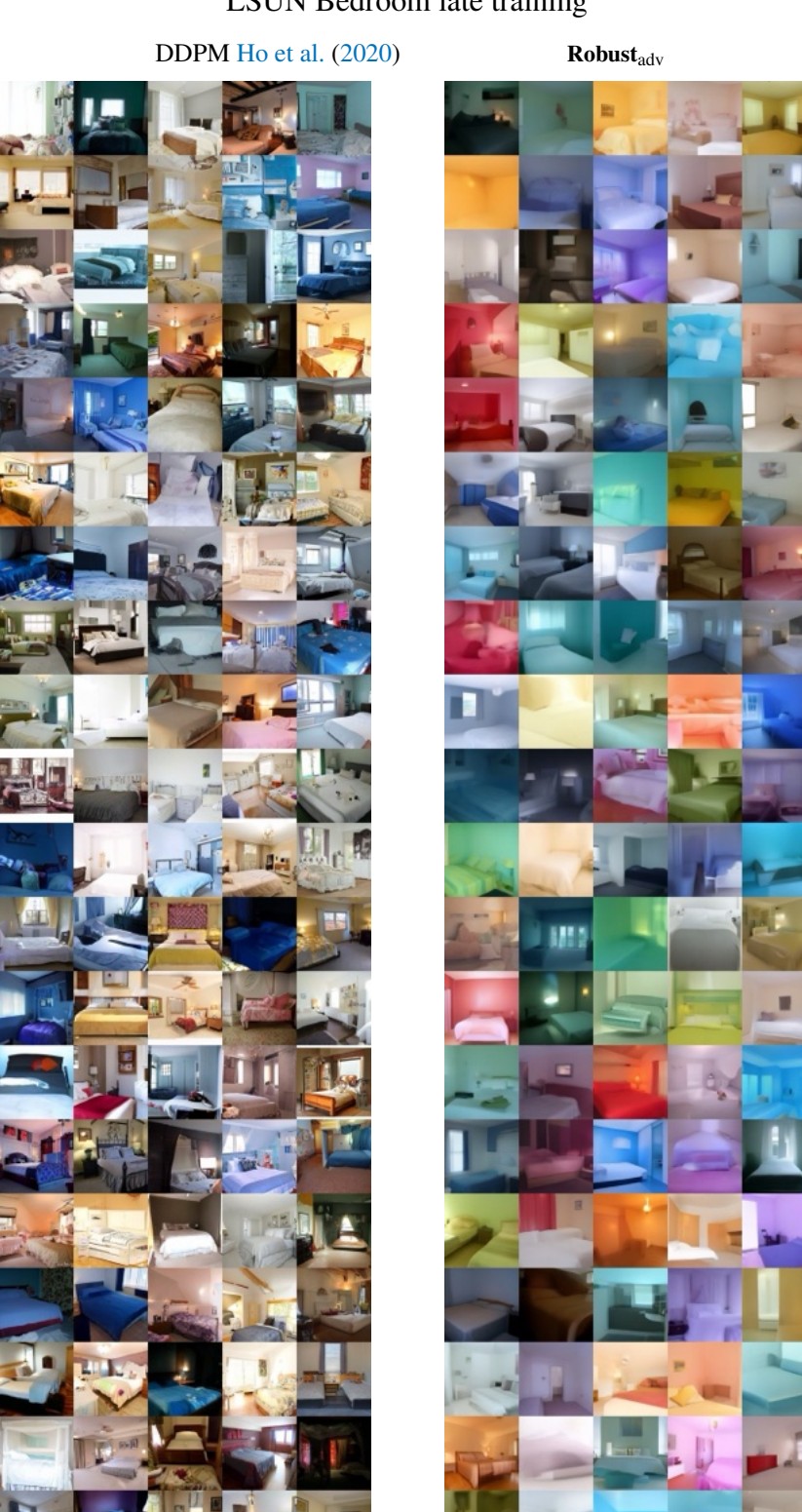

Figure 21: Trained on clean LSUN Bedroom. Despite added noise, **Robust**adv images look smooth and with less intricate details, even though more detailed than in earlier stages.

LSUN Bedroom with $p = 90\%$, $\sigma = 0.1$, late stage training

DDPM Ho et al. (2020)                    **Robust**$_\text{adv}$

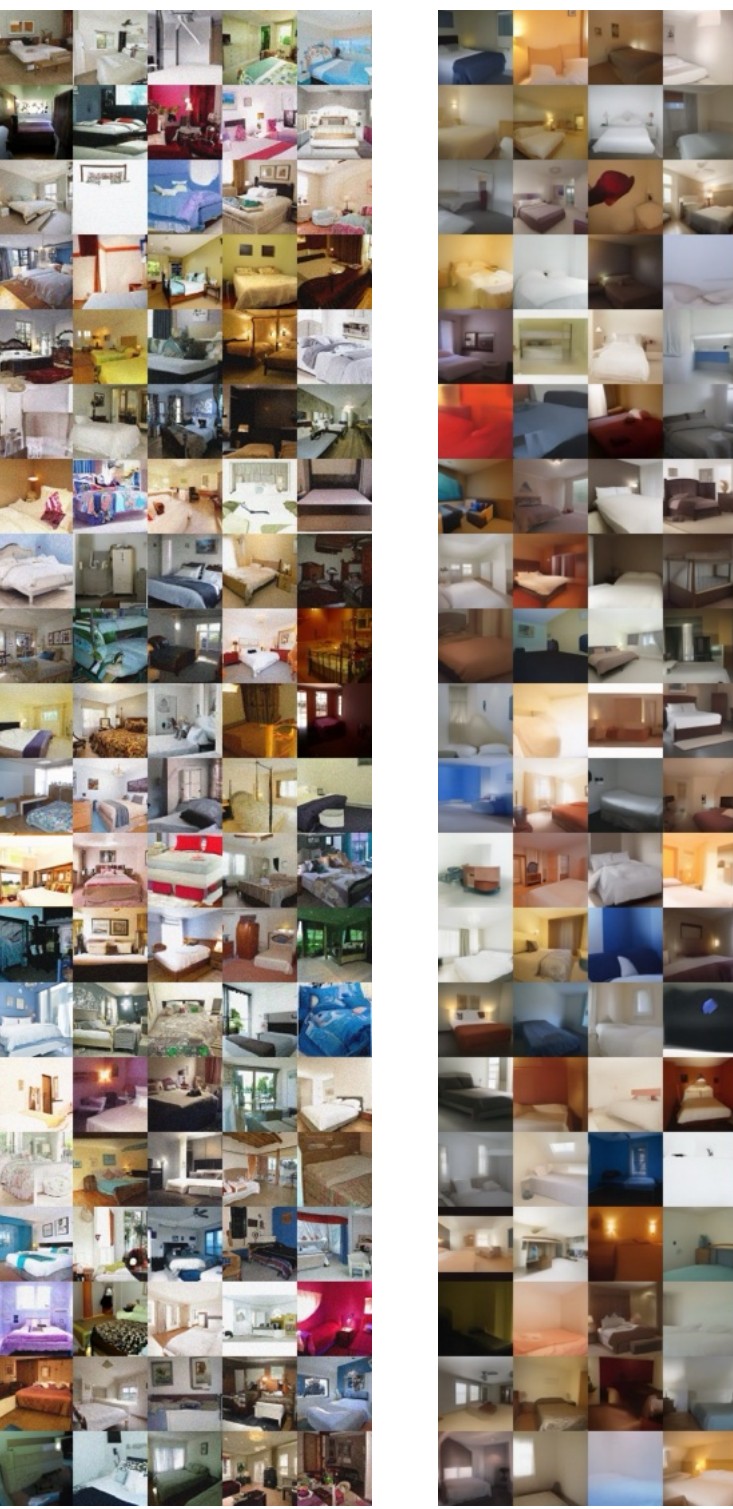

Figure 22: Trained on noisy LSUN Bedroom with $p = 90\%$, $\sigma = 0.1$. With extended training, **Robust**$_\text{adv}$ not only effectively removes the noise introduced into the dataset—in contrast to DDPM— but also restores fine details, resulting in multi-view images with natural colors and enhanced realism.

LSUN Bedroom with $p = 90\%,\ \sigma = 0.2$, late stage training

DDPM Ho et al. (2020)  **Robust**$_{\text{adv}}$

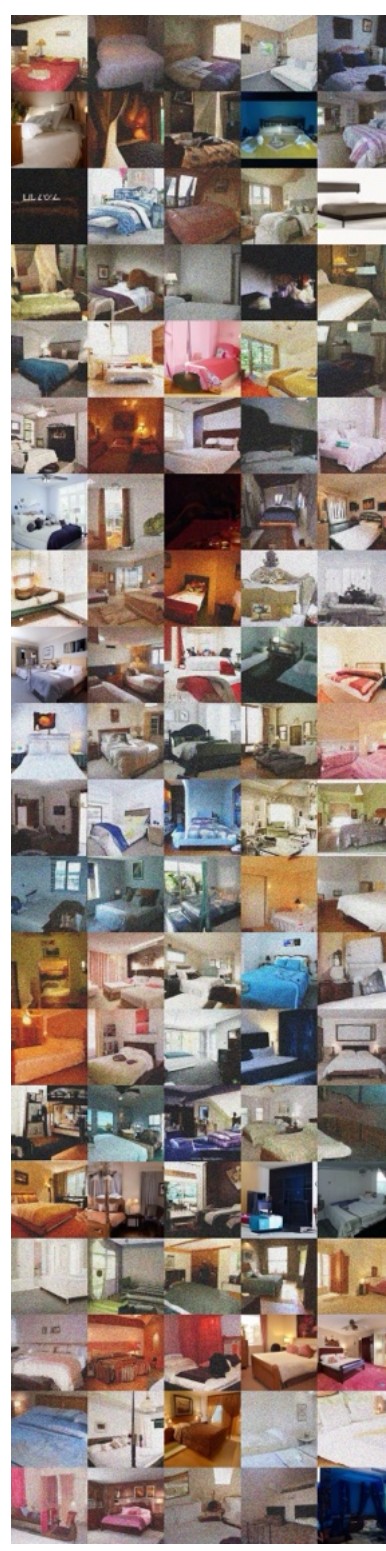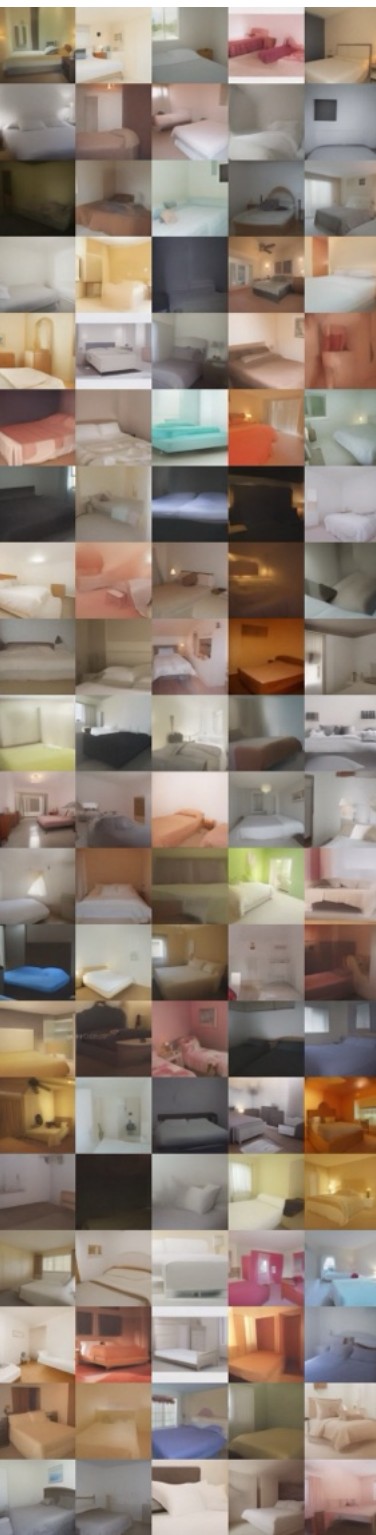

Figure 23: Trained on noisy LSUN Bedroom with $p = 90\%,\ \sigma = 0.2$. With extended training, **Robust**$_{\text{adv}}$ effectively removes the noise introduced into the dataset, in contrast to DDPM. As a result, it produces cleaner images, albeit with less intricate details.

## F ADVERSARIAL TRAINING ANALYSIS

This method aims at proposing an AT approach to the diffusion model's training whose design choices have been motivated extensively in previous sections as well as in the main paper. In this section, we want to highlight some interesting points we observed during the framework formulation.

### F.1 TRAINING DYNAMICS

Adversarial training diffusion models inevitably influences DM training dynamics. Indeed, the proposed regularization acts as a smoothing factor for the diffusion process in the trajectory space. In order to evaluate the training dynamics, we propose an ablation on DM generated samples at different training iterations. Fig. 24 is intended to show the evolution of generated samples by AT models at different training iterations. The first row shows samples generated by models trained in an early stage, while the second shows generations from models trained for longer. On the right column, the dataset has not been corrupted; the generations, after more training iterations, start losing the bright colors, tending towards more natural-looking colors. Moreover, the generated data starts acquiring its details. The same effects can be seen for models trained on corrupted data, $\sigma = 0.1$ in the middle and $\sigma = 0.2$ on the right (both with $p = 0.9\%$). In those cases, it is also possible to see that some generated samples, which at earlier epochs still resulted in being noisy, are completely denoised. This dynamic suggests that the model first focuses on fitting the overall data model, focusing more on the smoothing effect. Once done, the model goes back to learning the details of the data distribution, including some variability, but still not taking into account the noise present in the data. Furthermore, a clearer picture of the training dynamics can be obtained by examining images Figs. 18 to 23, that effectively compare the robust approach with the DDPM model at both early and late training stages.

### F.2 FINETUNING ANALYSIS

Adversarial training notably introduces a training time overhead. In our case, the increased training time is an investment for improved robustness and faster inference, which is particularly relevant in real-world pipelines where inference is repeated continually, while training is performed once for all. This is a standard trade-off in modern generative modeling, as seen in classifier guidance, which also increases training and complexity but is widely adopted.

In this section, we present preliminary results concerning the application of adversarial training in the fine-tuning setting. We point out that fine-tuning does help to alleviate the problem of training cost, so it could be a strong future improvement to allow the application of AT to heavier training pipelines. The table below shows FID and IS results evaluated on the CelebA dataset. For the fine-tuning, the model has been trained for all it training epochs according to the DDPM framework, except for the last 100 ones, when the adversarial regularization loss was applied. The table Table 4 showcases finetuning evaluation results. If compared with results in Table 2 we observe that we have similar results to the paper but with a fraction of the computational time.

| Configuration | Fine-tuned | | From Scratch | |
|---|---|---|---|---|
| $p\%/\sigma$ | DDPM | $\mathbf{Robust_{adv}}$ | DDPM | $\mathbf{Robust_{adv}}$ |
| 0.9 / 0.1 | 65.4 / 2.6 | **23.3 / 2.1** | 54.90 / 2.40 | **14.54 / 2.09** |
| 0.9 / 0.2 | 100.68 / 2.7 | **25.8 / 2.1** | 96.03 / 2.65 | **16.53 / 2.11** |

Table 4: Performance comparison (FID/IS) between DDPM and $\mathbf{Robust_{adv}}$ when finetuning on CelebA

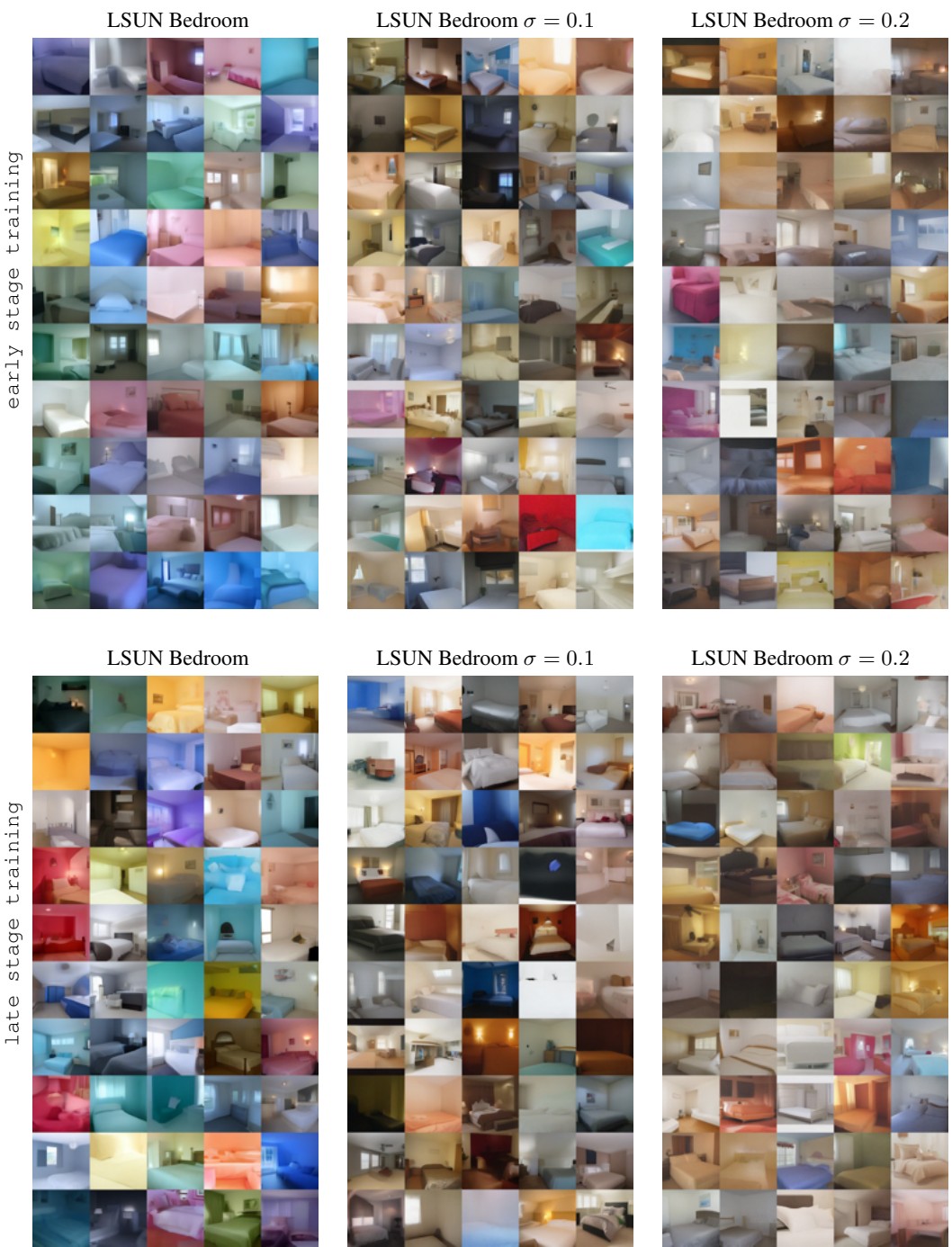

Figure 24: Qualitative results analysis on samples generated by **Robust**$_{\text{adv}}$ at different training stages.

### F.3    How $\lambda$ in Eq. (9) of the main paper influences the model's denoising capability

In the method section, we stress that the choice of the hyperparameter $\lambda$ heavily influences the model's smoothing ability. To further motivate the previous statement, we provide straightforward evidence of this by observing generated samples produced by different models, with the same architectures and minimum regularization ray among all the shown samples. The varying parameter is $\lambda$, which

LSUN Bedroom     LSUN Bedroom $\sigma = 0.1$     LSUN Bedroom $\sigma = 0.2$

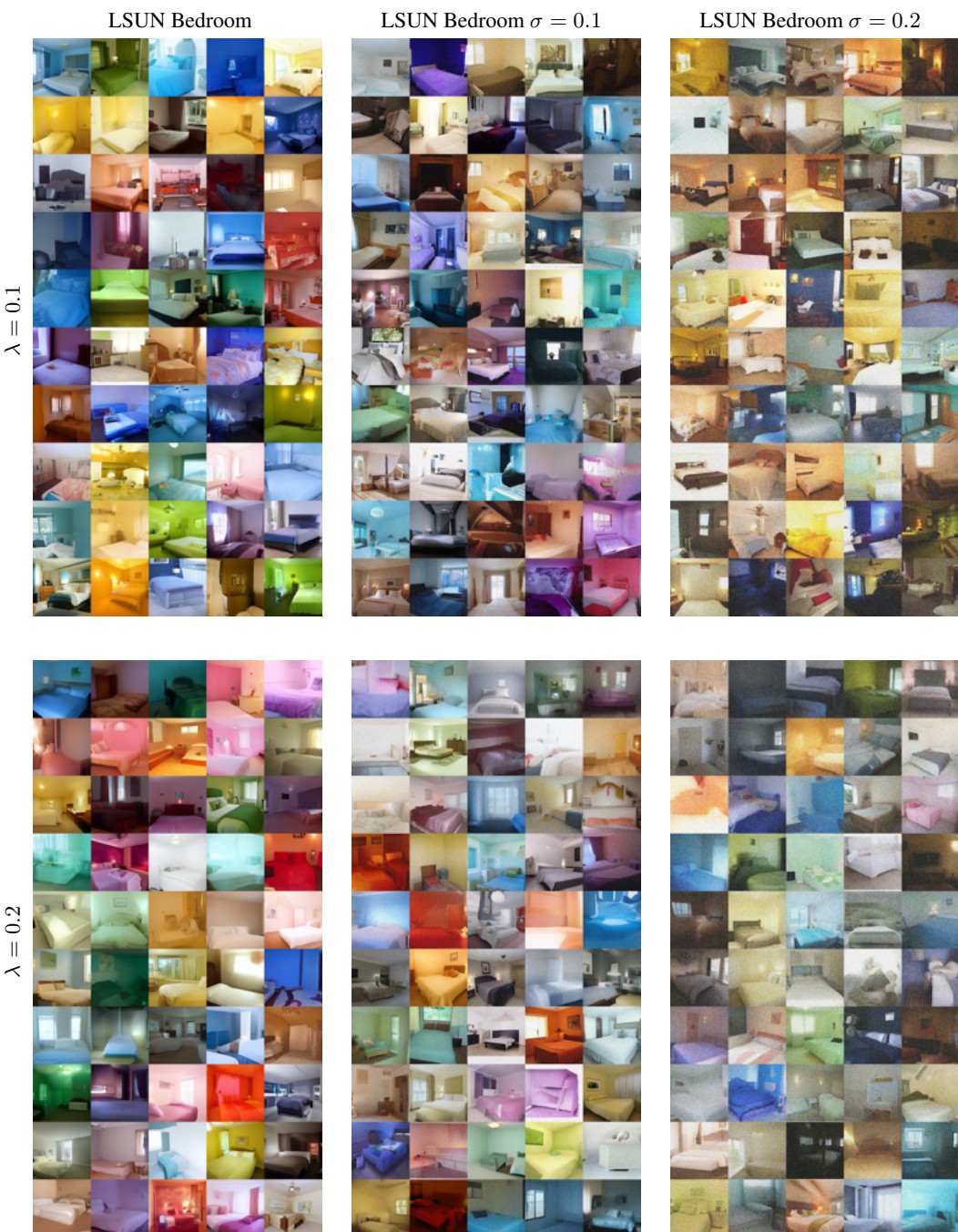

Figure 25: **Robust$_{\text{adv}}$** trained on LSUN Bedroom dataset, with different noisy data ($p = 90\%$, different $\sigma$ are visible in the image). The first row sets the regularization hyperparameter $\lambda$ to 0.1, the second to 0.2.

is set to the values $\{0.1, 0.2, 0.3\}$. Fig. 25 shows the results at an early training stage of the models. Despite being at an early stage, the $\lambda$ influence in models' performance already appears clear. When the data is not noisy (first column), increasing its value results in oversmoothing data, losing subject details, due to the smoothing factor introduced by the regularization. When the data becomes noisy, the regularization becomes fundamental in learning the correct distribution. In the first row, we see that the smoothing action is limited due to the small $\lambda = 0.1$, indeed the noise is still present in

the generated samples both in $\sigma = 0.1$ and $\sigma = 0.2$, whereas the noise decreases drastically when increasing $\lambda$ to $0.2$. In fact, the images shown in the bottom row show a minor presence of noise, which is expected to disappear in later training. On the other side, the increase of the parameter $\lambda$ also causes a loss of details in the image subjects. This phenomenon is due to the smoothing effect, which not only affects noise but also data variability. This smoothing effect becomes even more apparent when compared to Figs. 18 to 23, all generated with $\lambda = 0.3$. These comparisons further support the previous observations by extending the analysis across different levels of noise and training stages.

## G  COMPARISON WITH NOISE-AWARE DIFFUSION TRAINING

The primary objective of Daras et al. (2024a) is to develop noise-informed algorithms for training models in the presence of noisy training data. More in detail, the noise-informed training algorithm operates under two fundamental assumptions: (i) the assumption that the noise in the dataset is Gaussian and prior knowledge of the Gaussian variance, and (ii) identification of the specific training samples affected by noise corruption. To rigorously assess the robustness of this approach under the *unknown corruption* setting, we developed two distinct training configurations that relax these stringent assumptions and modify the original training framework of Daras et al. (2024a).

⋄ In *Conf. 1*, the method always knows the exact $\sigma$ level of the noise in the dataset but the assumption on which sample is clean $\mathbf{x} \in \mathcal{X}_{\text{clean}}$ and which sample is noisy $\mathbf{x} \in \mathcal{X}_{\text{noisy}}$ is forced to be correct only $(1 - p)\%$ of the time.

⋄ In *Conf. 2*, the assumption of knowing whether a sample is noisy or not is never considered, effectively forcing the same behavior for all the training data, when the data are noised with probability $p$.

For both configurations, we trained the models on CIFAR-10, considering clean data and noisy data with $\sigma = \{0.1, 0.2\}$ and $p = 90\%$. Quantitative results are summarized in Table 5 while qualitative examples are shown in Fig. 26.

| p % | $\sigma$ | Robust$_{\text{adv}}$ | Daras et al. (2024a) *Conf. 1* | Daras et al. (2024a) *Conf. 2* |
|-----|----------|----------------------|--------------------------------|--------------------------------|
| 0   | –        | 28.7                 | **14.0**                       | **14.9**                       |
| 0.9 | 0.1      | **24.7**             | 94.5                           | 102.7                          |
| 0.9 | 0.2      | **24.8**             | 109.7                          | 105.3                          |

Table 5: Experiments on CIFAR-10 under unknown corruption.

These experiments confirm that relaxing even one of the assumptions made in the noise-aware solution proposed by Daras et al. (2024a) reduces the method's robustness to unknown noise in the data, producing very high FID values. This confirms the practical limitations already highlighted by Daras et al. (2024a). On the contrary, our method is able to work in this more challenging setting, where the corruption is unknown, and achieves a stable trend in the FID across different $\sigma$ and $p$, without requiring access to clean/noisy labels or corruption parameters.

## H  LLM USAGE

Large language models were used exclusively for text polishing and minor exposition refinements. All substantive research content, methodology, and scientific conclusions were developed entirely by the authors.

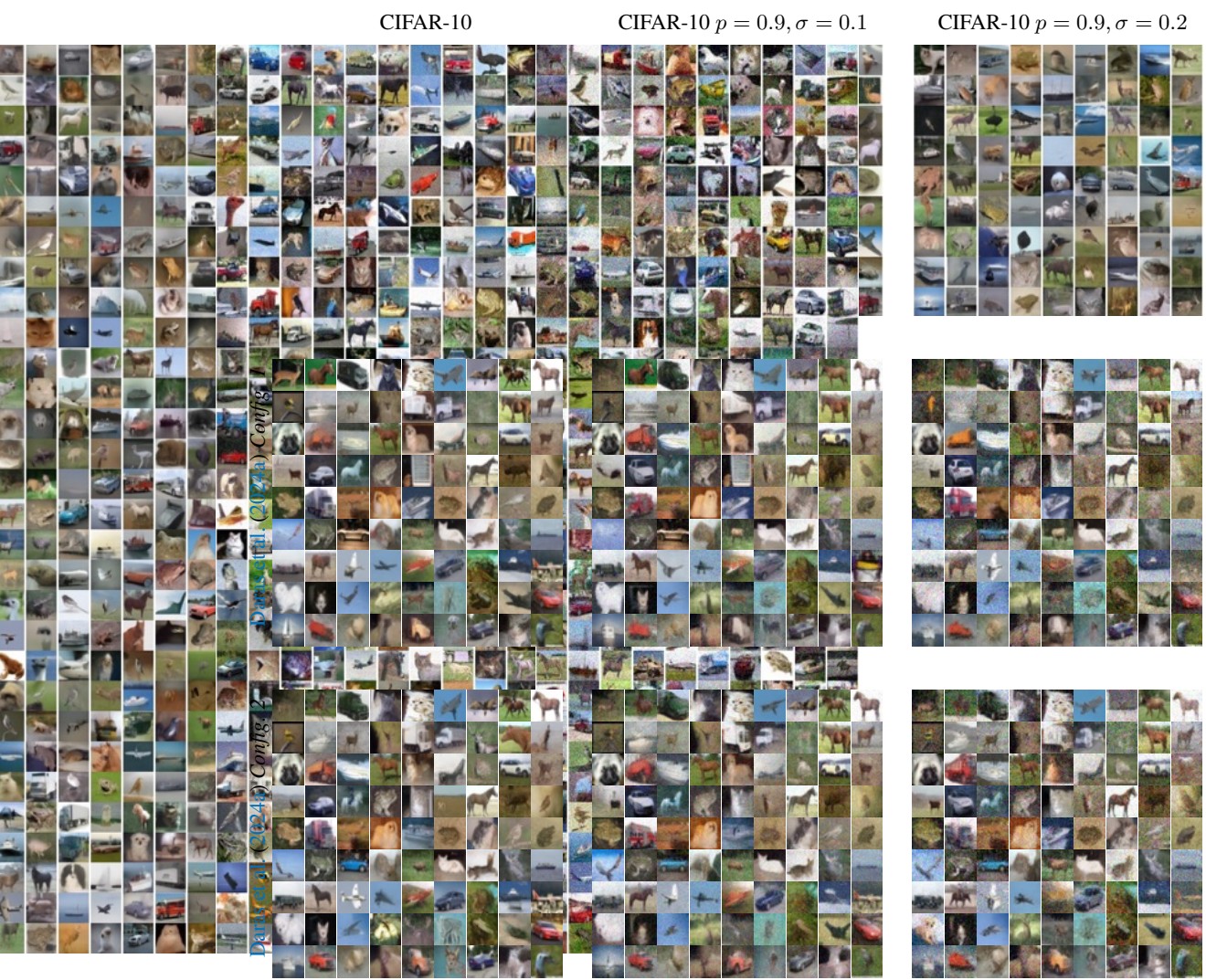

Figure 26: **Robust_adv** trained on CIFAR-10 dataset compared with the two proposed configurations of Daras et al. (2024a) in the unknown noise setting. Training data are either clean (*first column*) or noised with different noise levels ($p = 0.9, \sigma = \{0.1, 0.2\}$). The images demonstrate that **Robust_adv** effectively learns the target data distribution, ignoring the applied noise, even without making any assumptions about the applied perturbation.

