# OpenReview forum: "Why Adversarially Train Diffusion Models?"
_ICLR.cc/2026/Conference — ICLR 2026 Poster_

### Official Review · Reviewer_PUAf · 2025-10-26

**Soundness:** 4
**Presentation:** 4
**Contribution:** 4
**Rating:** 8
**Confidence:** 3

**Summary:**

This paper proposes an adversarial training framework for diffusion models that incorporates adversarial regularization directly into the denoising objective to improve robustness against noisy training data and input perturbations during the diffusion process. The proposed method (Robustadv) enforces equivariance by ensuring that small perturbations to noisy samples lead to proportionally consistent responses, while the perturbation magnitude is adaptively bounded to preserve the diffusion process’s stability. Experiments on synthetic and real datasets such as CIFAR-10, CelebA, and LSUN Bedroom demonstrate that the method enhances resistance to corruption, reduces memorization, and achieves smoother diffusion trajectories that allow faster sampling, all while maintaining high image quality even under heavy noise conditions.

**Strengths:**

1. The paper introduces a novel perspective on adversarial training for diffusion models by shifting the focus from classifier-level invariance to score-field equivariance. This insight effectively redefines robustness in the context of generative diffusion processes.

2. The proposed method is simple yet well grounded, requiring only a small modification. The time-dependent perturbation bound maintains compatibility with the diffusion process.

3. The paper presents extensive experiments across synthetic and real-world datasets, clearly showing improvements in robustness against noisy training data, reduced memorization, and smoother diffusion trajectories. The analysis of faster sampling and resilience to trajectory attacks further supports the method’s practical value.

**Weaknesses:**

The paper is overall well executed, with clear motivation, principled formulation, and comprehensive experiments. The only minor weakness is that the evaluation focuses mainly on DDPM and DDIM frameworks, leaving open whether the proposed adversarial regularization generalizes to newer diffusion models.

**Questions:**

- Given that the proposed objective enforces local smoothness in the score field, could this formulation be extended to improve adversarial purification or other defense mechanisms?
- Would the proposed adversarial regularization also improve robustness when training on datasets with natural corruptions?

---

> ### Author Response · Authors · 2025-11-21
> **Answer to Reviewer PUAf**
>
> ## Strengths and merits of the paper
>
> Thank you for the detailed and positive feedback. We are pleased that Reviewer `PUAf` highlighted several key strengths of our work, many of which were also independently emphasized by the other reviewers:
>
> - the **novel conceptual perspective** of shifting from classifier-style invariance to **score-field equivariance** for diffusion models, which aligns with the conceptual clarity noted by `Djvf` and the originality emphasized by `2wFr`;
> - the assessment that our method is **simple, principled, and well-grounded**, requiring only minimal modifications to standard DDPM training — a point similarly recognized by `Djvf` (plug-in simplicity) and `2wFr` (clean integration and clear formulation);
> - the recognition that our experiments are **extensive and compelling**, demonstrating improved robustness under heavy corruption, reduced memorization, smoother trajectories, and even **faster sampling**, consistent with `Djvf`’s observations on improved FID and score-field stability and with `2wFr`’s emphasis on robustness and practical relevance;
> - the acknowledgment that the paper is **well executed and clearly presented**, reinforcing the positive assessments on clarity and structure from `2wFr` and partially echoed by `Djvf`.
>
> We appreciate Reviewer `PUAf`’s strong endorsement and are encouraged by the consistent agreement across reviewers regarding the novelty, clarity, and empirical strength of our approach.
>
> ---
> We now address the few remaining remarks.
>
> > W1) The only minor weakness is that the evaluation focuses mainly on DDPM and DDIM frameworks, leaving open whether the proposed adversarial regularization generalizes to newer diffusion models
>
> We thank the reviewers for this comment. The main aim of the approach was to propose a foundational AT framework for denoising diffusion models, grounded on the principles of the generative mechanism itself. The paper indeed does not assume any specific DM implementation; the only assumption that is made is that the network infers the noise to be removed iteratively from the data that is being generated. In other words, it is an $\epsilon$-predicting network.
> The application of the method to newer approaches is something that we further aim to investigate in future works. As stated in Section 5 - Conclusions and Future Work, we aim to extend the approach to $\mathbf{x}$-predicting network, methods that predict the whole sample, and also $\mathbf{v}$-predicting network; please see the EDM paper (Karras et al., 2022).
>
> Regarding the application to larger architectures, the same motivation applies: models such as StableDiffusion 1.4 or 2.1 rely on an $\epsilon$-prediction objective, which makes them well-suited for extending our approach.
>
>
> ---
>
> ### Q1) Extensions to adversarial purification or other defense mechanisms
> > Q1) Given that the proposed objective enforces local smoothness in the score field, could this formulation be extended to improve adversarial purification or other defense mechanisms?
>
> In Section 4. **"Adversarial defenses with denoising or randomized smoothing"**, we highlighted the correlation between the denoising effect of the proposed approach and the adversarial purification task.
> We believe that our approach can also be applied as a reconstruction network, which, given a classifier and a pool of adversarial images, can reconstruct the inputs while avoiding the adversarial noise introduced during the attack.
> Another possible application is in the context of jailbreak attacks.
> As shown in Figure 2, the network tends to avoid learning outliers.
> Jailbreak attacks intentionally introduce outlier noise to associate a certain behaviour with a particular outlier.
> We believe that our approach could help mitigate this threat by leveraging its inherent ability to avoid such outliers during training.
>
> ---
> ### Q2) Improve robustness on datasets with natural corruptions
> > Q2) Would the proposed adversarial regularization also improve robustness when training on datasets with natural corruptions?
>
> The proposed regularization is conceived as an adversarial robustness-oriented regularization, but, as shown, it can also function in other corruption-agnostic cases.
>
> The denoising capability developed by robust generative models, therefore, does not require knowledge of the noise distribution beforehand; consequently, we would assume that the denoising capability would still hold in cases of naturally corrupted data.
>
> Sensitive domains, such as MRI and sensor data generation, undoubtedly suffer from noisy measurements.  These fields are suitable for applying the proposed regularisation, and we appreciate the reviewer’s suggestion.

---

> > ### Comment · Reviewer_PUAf · 2025-11-26
> >
> > I thank the authors for their detailed response. I remain convinced of the paper's novelty and solid contribution to the field. Therefore, I will maintain my initial rating.

---

> ### Author Response · Authors · 2025-11-27
>
> We thank the reviewer for their insightful feedback and are happy that our additional explanations in the rebuttal helped clarify and deepen their understanding of the topic.

---

### Official Review · Reviewer_2wFr · 2025-10-30

**Soundness:** 4
**Presentation:** 3
**Contribution:** 3
**Rating:** 6
**Confidence:** 3

**Summary:**

This paper proposes an adaptation of Adversarial Training (AT) for Diffusion Models, framing it as an equivariance constraint aligned with denoising dynamics. The method aims to improve model robustness against noise, data corruption, and adversarial attacks, while also reducing memorization. The authors provide empirical validation on both low-dimensional synthetic data and standard image benchmarks.

**Strengths:**

Originality: Successfully adapts Adversarial Training, a discriminative technique, to the generative setting of Diffusion Models via a novel equivariance constraint.

Quality & Significance: Comprehensive experiments show the method confers clear advantages: robustness to noise/corruption, reduced memorization, and security against adversarial attacks. This is highly relevant for safe and reliable deployment of DMs.

**Weaknesses:**

- The paper focuses on proof-of-concept and standard benchmarks. Testing on more complex, large-scale datasets (e.g., ImageNet) would further strengthen the claims of general applicability.

- The computational overhead of adversarial training during the diffusion process is not thoroughly discussed.

- related work: The related work section lacks engagement with several key areas of relevant research. This includes: 1) foundational work on robust denoising autoencoders [1], 2) the parallel and distinct line of research on adversarial purification using diffusion models [2], and 3) recent analysis specifically on the nature of adversarial vulnerabilities in DMs [3]. Addressing these would better contextualize the paper's contributions. I think that is not much work for the rebuttal.


References
1. "Time-based Sampling and Reconstruction of Non-bandlimited Signals" - https://ieeexplore.ieee.org/document/8682626
2. "Diffusion Models for Adversarial Purification" - https://arxiv.org/abs/2205.07460
3.  ''Adversarial Examples are Misaligned in Diffusion Model Manifolds'' - https://arxiv.org/abs/2401.06637

**Questions:**

1. What is the estimated computational overhead (training time, memory) of your adversarial training method compared to standard diffusion model training?

2. Have you explored the trade-offs between the "random perturbation" (smoothing) and "adversarial perturbation" variants of your method in terms of robustness gain vs. computational cost?

---

> ### Author Response · Authors · 2025-11-21
> **Answer to Reviewer 2wFR (part 1/2)**
>
> ## Strengths and merit of the paper
> Thank you for the thoughtful and constructive feedback. We are pleased that Reviewer `2wFr` highlighted several important strengths of our work, many of which were independently noted by the other reviewers as well:
>
> - the **originality** of adapting adversarial training to diffusion models through a principled **score-field equivariance formulation**, which also aligns with the conceptual insights emphasized by `Djvf` and `PUAf`;
> - the recognition that our empirical evaluation is **comprehensive**, demonstrating clear gains in **robustness to noise and corruption**, **reduced memorization**, and **resilience to adversarial attacks** — strengths also emphasized by `Djvf` (improved FID under heavy corruption) and `PUAf` (robustness across synthetic and real datasets);
> - the acknowledgement that our method’s contributions are **significant and practically relevant**, especially for improving the **safety and reliability of diffusion models**, an impact also noted explicitly by `PUAf` (faster sampling, smoother trajectories) and implicitly by `Djvf` (tighter score fields and better PSNR/subspace recovery).
>
> We appreciate Reviewer `2wFr`’s positive assessment and are encouraged by the strong alignment across reviewers regarding the conceptual novelty, empirical robustness, and practical utility of our approach.
>
> ---
>
> We now address the few remaining remarks, hoping that they can improve the evaluation of our work.
>
> ### (W1) More complex, large-scale datasets like ImageNet (W1, Q1).
>
> > W1) The paper focuses on proof-of-concept and standard benchmarks. Testing on more complex, large-scale datasets (e.g., ImageNet) would further strengthen the claims of general applicability
>
> We thank the reviewer for this comment. As highlighted by all reviewers, the proposed approach is supported through a comprehensive experimental evaluation conducted on multiple datasets and different data configurations, including multiclass data of varying resolution and size. More in detail, we consider a progression of real-world datasets of increasing difficulty: CIFAR-10 (32×32, 50K), CelebA (64×64, 202.6K), and LSUN Bedroom (256×256, 303.1K).
>
>
> Nevertheless, following the reviewer's suggestion, we conducted some additional experiments considering the suggested large-scale and more complex ImageNet, to further strengthen our evaluation.
>
> **The ImageNet evaluation added at this stage of the rebuttal is considering early checkpoints of the training process.**
> It is possible to appreciate that even at an early stage in the training process, our robust model achieves lower FIDs on different configurations, compared to DDPM.
>
> | p% | σ | DDPM | Robust$_\text{adv}$ |
> |----|---|------|-----------|
> |  0    | 0    |    **84.8**  |  88.6        |
> |  0.9  | 0.1  |  97.6    |     **83.8**    |
> |  0.9  | 0.2  |    129.4  |  **80.3**        |
>
>
> We added these results in the text in section 3.1 in the paragraph **Resistant to noise by design.**
>
> ### (W2, Q1) Computational overhead
>
> > W2) The computational overhead of adversarial training during the diffusion process is not thoroughly discussed.
> > Q1) What is the estimated computational overhead (training time, memory) of your adversarial training method compared to standard diffusion model training?
>
> As concerns the time complexity, in section 3.1 of the original manuscript at the paragraph **Time complexity**, we analyse the time complexity of the approach, which reports an average time increase of 150%, with AT running approximately 2.5 times slower than the standard diffusion model training.
>
> Furthermore, in the appendix section F.2 of the revised manuscript, we further discuss the impact of AT on the time complexity of the algorithm, providing as an alternative **a robust finetuning approach** to **mitigate the computational overhead**. This offers almost similar performance to training from scratch, but greatly improves the training complexity. The preliminary analysis shows that, by finetuning only for the last 100 epochs of the training pipeline, the smoothing capability of the model still holds, reducing the FID on noisy data from $65.4$ to $23.2$ for $90$% of noisy data at $\sigma=0.1$ and from $100.68$ to $25.9$ for $90$% of noisy data at $\sigma=0.2$.
>
> In terms of **memory usage**, our method does not introduce any additional overhead. The attack requires backpropagation in the network only once, and the gradients are not stored or accumulated. Once computed, those are directly added to the input $\mathbf{x}_t$, therefore the approach does not overload the GPU memory.

---

> ### Author Response · Authors · 2025-11-21
> **Answer to Reviewer 2wFR (part 2/2)**
>
> ### (W3) Additional related work papers
>
> > W3) The related work section lacks engagement with several key areas of relevant research. This includes: 1) foundational work on robust denoising autoencoders [1], 2) the parallel and distinct line of research on adversarial purification using diffusion models [2], and 3) recent analysis specifically on the nature of adversarial vulnerabilities in DMs [3].
>
> We thank the reviewer for the valuable suggestions that allowed us to better contextualize the paper. We have added the reference [3] as _Lorenz et al. (2024)_ in the minisection **Adversarial Robustness**, mentioning the finding that adversarial attacks do not align well with the manifold learned by DMs.
>
> We also note that the suggested work "Diffusion Models for Adversarial Purification" https://arxiv.org/abs/2205.07460 is already cited and discussed in the paper as _Nie et al., 2022_ in the paragraph **Adversarial defenses with denoising or randomized smoothing**.
>
> We also reviewed the article [1]: “Time-based Sampling and Reconstruction of Non-bandlimited Signals”. However, we were unable to identify any direct connection to robust denoising autoencoders, nor did we find citations to this work in the foundational literature on denoising autoencoders. We would be very grateful if the reviewer could provide additional guidance to help us appropriately position this reference within our related work.
>
>
> ---
> ### (Q2) Trade-off random vs adversarial perturbations
>
> > Have you explored the trade-offs between the "random perturbation" (smoothing) and "adversarial perturbation" variants of your method in terms of robustness gain vs. computational cost?
>
>
> We appreciate the reviewer’s point. The original manuscript includes a dedicated discussion in Section 3.1 - **Random or adversarial perturbation?** that clarifies the motivation behind choosing an adversarial rather than a random perturbation. We also report that adversarial perturbation can guarantee a much stronger denoising effect than random, yet is more expensive for training.
>
> In particular, **Table 1 (top)** directly compares random noise injection, sampled from a uniform distribution, called **Robust**$\_\text{rand}$  and adversarial perturbation **Robust**$\_\text{adv}$  during training.
>
> Both settings introduce additional perturbations during the denoising process, but **only the adversarial one is explicitly optimized, thus requiring an additional backpropagation over the input** while the random perturbation does not need it.
>
> Although the adversarial improves, it comes with higher computational complexity in the training, while random perturbation has basically the same complexity as standard training.
>
> As shown in Table 1 - CIFAR-10, injecting random noise **Robust**$_\text{rand}$ leads to much worse sample quality with an FID of $79.21$ with $\sigma = 0.1$, whereas the adversarial setting yields a substantially improved FID, down to $24.7$ at $\sigma = 0.1$ and an increase in Inception Score.
>
> The table below provides the requested information on the robustness gain vs. computational cost:
>
> | p% 0.9, $\sigma=0.1$              | FID ↓  | Time increase wrt std training |
> |---------------------|--------| ---|
> | DDPM | 102.68 | 1$\times$  |
> | **Robust**$_\text{rand}$        | 79.21  | $\approx$ 1$\times$ |
> | **Robust**$_\text{adv}$ | **24.7**  | $\approx$ 2.5$\times$ |

---

> ### Comment · Reviewer_2wFr · 2025-11-22
> **upgrade decision**
>
> I appreciate the authors' detailed response. The new ImageNet experiments are a welcome addition and go a long way in addressing my concern about scalability. The clarifications on computational cost and the related work are also helpful.
>
> While the ImageNet results are preliminary, they are convincing enough to support the authors' claims for the scope of this conference. The core strengths of the paper - originality and demonstrated robustness - remain solid.
>
> The rebuttal has alleviated my main concerns. I have therefore revised my score upward, finding the paper now clearly above the acceptance threshold.

---

> > ### Author Response · Authors · 2025-11-24
> >
> > We would like to thank the reviewer for the thoughtful follow-up and for revising the score. We appreciate your recognition of the paper’s originality and robustness, and we are glad that the additional ImageNet experiments and the further clarifications helped address your concerns regarding scalability, computational cost, and the positioning within the related literature.
> >
> > Your insights have helped us improve the clarity and strength of our work. We will update the manuscript with the results on ImageNet computed using the final checkpoint.

---

### Official Review · Reviewer_Djvf · 2025-11-03

**Soundness:** 2
**Presentation:** 2
**Contribution:** 2
**Rating:** 4
**Confidence:** 3

**Summary:**

The authors argue that adversarial training (AT) for classifiers, which enforces invariance of predictions under small input perturbations, does not transfer directly to diffusion models (DMs), whose training objective is regression to injected noise. They propose that AT for DMs should instead enforce local equivariance of the noise predictor: when the trajectory state $x_t$ is perturbed by $\delta$, the predicted noise should change by (approximately) $\delta$. Formally, they add to the standard DDPM loss an equivariance regularizer (Eq. (6)) and train either with random perturbations or adversarial FGSM‑style perturbations. A time-dependent perturbation radius $r_{\beta}(t)$ is scheduled to respect the forward‑process variance. The adversarial step uses a variance-aware projection.

Empirically, on a synthetic 3D and a linearized butterflies dataset, the method produces tighter score fields and trajectories, better PSNR and subspace recovery under corruption. On CIFAR‑10, CelebA, LSUN Bedroom, when 90% of the training data are corrupted with Gaussian noise, Robustadv reduces FID versus DDPM/DDIM. Conversely, when training on clean data, Robustadv degrades FID with smoother but less detailed images. The paper also reports fewer near-duplicates by DINOv2 similarity, better tolerance to trajectory-space attacks and sometimes better FID.

**Strengths:**

1. The paper carefully argues that classifier-style invariance is misaligned for diffusion’s noise-regression objective, and instead enforces local equivariance of the score under trajectory perturbations; the formulation is explicit and contrasted with a naive invariance loss, with toy-world visual evidence that invariance drifts off-manifold
2. The training recipe is simple and can be seen as a plug-in. The proposed regularizer can be easily integrated with standard DDPM training. The time-dependent perturbation radius and adversarial update are demonstrated clearly.
3. With 90% of training samples noised ($\sigma = 0.1/0.2$), Robustadv sharply improves FID over DDPM/DDIM on CIFAR-10/CelebA/LSUN, which are supported by qualitative samples as well.

**Weaknesses:**

1. The paper positions itself against noise-aware diffusion training but does not compare to them experimentally. This leaves competitiveness under “unknown corruption” unestablished.
2. The penalty on clean data is substantial yet under-analyzed. With $p = 0\%$ noise, FID worsens markedly. The authors do not provide a trade-off analysis in depth.
3. The ELBO derivation with adversarial sub‑steps (Appendix A.1.3) is heuristic and does not yield quantitative guarantees in theory; also, assumptions about Gaussian intermediate states under non‑Gaussian perturbations are not formalized.
4. The perturbation radius $r_{\beta}(t)$ and scaling of $\lambda_t$ are primarily hand-crafted. No principled guidance on stability/consistency or dataset-noise dependence, which should have been provided, given that the adversarial training recipe may have altered the optimization landscape of the diffusion model.
5. The training is ~2.5x slower than standard diffusion model training, which is however already notoriously slow. In practice, this computational overhead may be unacceptable, especially provided with the scale of data used to train a diffusion model successfully.

**Questions:**

1. Can you compute-matched comparisons to noise-aware diffusion training methods on CIFAR-10/CelebA/LSUN for $p \in \{ 0.3, 0.6, 0.9 \}$ and $\sigma \in \{ 0.05, 0.1, 0.2 \}?
2. The evaluation in terms of the robustness to adversarial attacks are quite limited and do not match the common practice in the adversarial defense community. Can you evaluate EOT over diffusion stochasticity and adaptive attacks.

---

> ### Author Response · Authors · 2025-11-21
> **Answer to Reviewer Djvf (part 1/4)**
>
> ### Strengths and merit of the paper
>
> Thank you for the detailed feedback and constructive criticism. We are pleased that Reviewer `Djvf` recognized several core strengths of our work, many of which were independently highlighted by the other reviewers as well:
>
> - the observation that **classifier-style invariance is fundamentally misaligned with diffusion’s noise-regression objective**, and that our proposed **score-field equivariance** is a principled alternative — also emphasized by reviewers `2wFr` and `PUAf`;
> - the fact that our method is **simple, plug-in, and integrates cleanly into standard DDPM training**, with the perturbation schedule and adversarial update **clearly demonstrated** — a point likewise noted by reviewers `2wFr` and `PUAf`;
> - the strong empirical result that **under 90% corrupted training data**, Robustadv **sharply improves FID** over DDPM/DDIM on CIFAR-10, CelebA, and LSUN — corroborated also by reviewer `2wFr` (robustness, reduced memorization) and `PUAf` (improved trajectories and corruption tolerance);
> - the broader appreciation that our approach meaningfully tackles **key limitations of diffusion models**, including:
>     -  **noise sensitivity**,
>     -  **memorization**, and
>     -  **instability under perturbations**, which both `2wFr` and `PUAf` highlight as practically relevant for robust generative modeling.
>
> We appreciate Reviewer `Djvf`’s careful reading and are encouraged by the strong alignment across reviewers on the conceptual clarity, practical simplicity, and demonstrated robustness benefits of our method.
>
> ---
>
> ###  (W1, Q1) Relationship to noise-aware diffusion training.
> > W1) The paper positions itself against noise-aware diffusion training but does not compare to them experimentally. This leaves competitiveness under “unknown corruption” unestablished.
>
> > Q1) Can you compute-matched comparisons to noise-aware diffusion training methods on CIFAR-10/CelebA/LSUN for and $\sigma \in { 0.05, 0.1, 0.2 }$?
>
>
> We thank the reviewer for this comment. The authors from Daras et al. (2024a, 2024c, 2024d) themselves admit that their works are greatly limited by strong assumptions on the corrupting noise distribution, making them not suitable to work in the **unknown corruptions** setting. They acknowledge this as a fundamental limitation in the limitations sections of the respective papers and recognise that assuming prior knowledge of the data composition and noise pattern is unrealistic.
>
>
> Nonetheless, we agree that, although our method is motivated by the limitations present in these noise-aware diffusion training approaches, a comparison with these solutions under **unknown corruption** would strengthen our evaluation.
> To this end, we are conducting an additional set of experiments in which we modified the training procedure of Daras et al. (2024a) to remove any reliance on prior knowledge, such as:
> - **the noise level used to corrupt each image** (e.g., Gaussian with a specific $\sigma$), thereby avoiding the injection of exact corruption into the training objective; and
> - whether **a training sample is clean  $\mathbf{x} \in \mathcal{X}_{\text{clean}}$ or noisy** $\mathbf{x} \in \mathcal{X}_{\text{noisy}}$.
>
> Consequently, for this set of experiments, all samples are treated uniformly during the training of the U-NET-based consistency model used in Daras et al. (2024a), as is with our solution. For additional details, please refer to Algorithm 1 reported in Daras et al. (2024a).
>
>
> For these additional experiments, we are focusing on the CIFAR-10 dataset, mainly due to computational constraints and the time required to train each model, varying $\sigma$ and $p$.
>
> We will provide the results of this new set of experiments as soon as they are available, and we will update the manuscript accordingly.

---

> ### Author Response · Authors · 2025-11-21
> **Answer to Reviewer Djvf (part 2/4)**
>
> ### (W2) Trade-off analysis on the clean data
>
> > The penalty on clean data is substantial yet under-analyzed. With
>  noise, FID worsens markedly. The authors do not provide a trade-off analysis in depth.
>
>
> Following your remark, in the revised version of the paper, we present a more in-depth discussion about the smoothing effect when training on the clean data. This can be found in the main paper in section 3.3, renamed *Implications of smooth diffusion flow* and the newly added paragraph **Trade-off analysis on clean and noisy data**, offering also the new Fig. 7.
>
> We also provide additional analysis in Appendix D, and qualitative samples in Fig. 24, where we show the effect on the generated data of different training with different lambdas $\lambda=\{0.1,0.2,0.3\}$. Here we see that relaxing $\lambda$ effectively reduces the penalty on the clean data, yet it has a diminishing effect on suppressing the noise. Once $\lambda$ approaches zero, the baseline quality is recovered.
>
> Finally, the literature regarding adversarial robustness in the context of discriminative networks introduces a *similar* trade-off between robust and clean accuracy on data.
> We are the first to show that this trade-off arises in generative diffusion models: we believe that the smoothing is responsible for the penalty that generative models introduce when trained adversarially.
>
>
>
> ### (Q2) EoT over stochasticity and adaptive attacks
>
> > The evaluation in terms of the robustness to adversarial attacks are quite limited and do not match the common practice in the adversarial defense community. Can you evaluate EOT over diffusion stochasticity and adaptive attacks.
>
> #### Adaptive Attacks
>
> In Diffusion Models, different attacks [d-e] have been proposed, each of them targeting a specific task, and none of them targeting specifically the diffusion trajectory. In this work, we propose a fundamental analytical attack that, by optimizing the perturbation at each diffusion timestep (i.e., effectively providing a worst case), represents already an **adaptive attack** procedure (see Algorithm 2) aimed at disrupting the model’s trajectory definition.
>
> The choice of using PGD as an attack stems from previous papers analysing the effectiveness of various attacks in the context of sample purification using DMs. These papers specifically targeted the diffusion process.
> The analysis provided in [a] shows that, when comparing the two gradient-based attack methods, AutoAttack [b] and PGD [c], PGD is found to be more effective.
>
> #### EoT
>
> We thank the reviewer for raising the point about including the EoT attack in our pipeline. We recognize it as a valid point because the diffusion process itself, whenever the inference is done through DDPM, involves stochasticity.
>
> We formulate the EoT [f] attack building on the previously tested PGD attack presented in section 3.4. In the same section, we now describe the EoT attack setting, while implementation details are provided in Appendix A.4.
>
> The attack shows effectiveness against the considered models and in particular, coherently with the previously tested attacks we can observe that 1) the Robust model greatly outperforms DDPM baseline by mitigating the attack disruption, achieving in all settings a lower FID compared to DDPM 2) Robust$_\text{adv}$ remains robust against the EoT attack, with similar results as the other PGD attack.
>
> |  Attacked steps | 250  | 500   | 750   | 1000    |
> |-----------|-------|-------|--------|--------|
> | DDPM      | 57.6  | 132.7 | 195.99 | 248.6  |
> | Robust$_\text{adv}$| **25.9**  | **61.2**  | **100.2**  | **132.6**  |
>
>
> [a] Robust Evaluation of Diffusion-Based Adversarial Purification
>
> [b] F. Croce and M. Hein, “Reliable evaluation of adversarial robustness
> with an ensemble of diverse parameter-free attacks,” in ICML, 2020
>
> [c] Aleksander Madry, Aleksandar Makelov, Ludwig Schmidt, Dimitris Tsipras, and Adrian Vladu. Towards deep learning models resistant to adversarial attacks. In ICLR, 2018
>
> [d] Shih, Chun-Yen, et al. "Pixel is not a barrier: An effective evasion attack for pixel-domain diffusion models." Proceedings of the AAAI Conference on Artificial Intelligence. Vol. 39. No. 7. 2025.
>
> [e] Hadi Salman, Alaa Khaddaj, Guillaume Leclerc, Andrew Ilyas, and Aleksander Mądry. 2023. Raising the cost of malicious AI-powered image editing. In Proceedings of the 40th International Conference on Machine Learning (ICML'23), Vol. 202. JMLR.org, Article 1240, 29894–29918.
>
> [f] Athalye, Anish, et al. "Synthesizing robust adversarial examples." International conference on machine learning. PMLR, 2018.

---

> ### Author Response · Authors · 2025-11-21
> **Answer to Reviewer Djvf (part 3/4)**
>
> ### (W3) ELBO Derivation
> > The ELBO derivation with adversarial sub‑steps (Appendix A.1.3) is heuristic and does not yield quantitative guarantees in theory; also, assumptions about Gaussian intermediate states under non‑Gaussian perturbations are not formalized.
>
> The ELBO derivation outlined in Appendix A.1.3. is an adaptation of the ELBO derivation from the paper Denoising Diffusion Probabilistic Models.
> We leverage it to formalize the insertion of another state at timestep $t$ and to analyze how the ELBO changes under an additional perturbation.
> The perturbation is added within the same timestep rather than as an auxiliary transition, representing a deviation of the model from its expected behavior at that point in the trajectory.
> Indeed, the perturbation is added to the DM within the same timestep $t$, not considering it as an auxiliary transition towards a different timestep, but instead as an adversarial transition where the model moves to a point of the trajectory where its behavior diverges from what should be learnt.
> The adversarial perturbation is crafted via FGSM at each timestep.
> Empirically, the effect of the attack can be interpreted as a shift of the mean of the intermediate state distribution. We would also highlight that the attack is added to the $x_t$ with a proper scaling that is dependent on the standard deviation of $x_t \sim\mathcal{N}(\sqrt{\bar{\alpha_t}} x_0 , (1-\bar{\alpha_t})I)$. The application of the rescaled attack would then not substantially cause the data to exceed the 3$\sigma$ boundary of the Gaussian distribution, which is also guaranteed by the attack value clamping.
> This allowed us to retain the standard Gaussian-based derivation of the ELBO as a heuristic tool to analyze the impact of perturbations at intermediate timesteps, while focusing on how the model responds to deviations in the trajectory.
>
> ### (W4) Heuristics and Radius Scheduler
> > The perturbation radius and scaling are primarily hand-crafted. No principled guidance on stability/consistency or dataset-noise dependence, which should have been provided, given that the adversarial training recipe may have altered the optimization landscape of the diffusion model.
>
> The definition of the radius and the scaling consists of heuristics motivated by the need for the regularization to be compatible with the diffusion process dynamics. The outline of the motivation is reported in section 2.4. The ray scheduling derivation is motivated as follows:
>
> Given the noisy sample $ x_t = \sqrt{\bar{\alpha_t}} x_0 + (1-\bar{\alpha_t}) \epsilon~~~ \text{where} ~~~\epsilon \sim \mathcal{N}(0,1)$
>
> This quantity is alternatively represented as $x_t \sim\mathcal{N}(\sqrt{\bar{\alpha_t}} x_0 , (1-\bar{\alpha_t})I)$ via reparametrization trick, which is a gaussian with variance $1-\bar{\alpha_t}$.
>
> With the adversarial component $\delta$, it becomes  $x_t^{adv} = \sqrt(\bar{\alpha_t}) x_0 + (1-\bar{\alpha_t}) (\epsilon +\delta)$.
>
> To maintain consistency with the noise scale at each timestep, we scale the perturbation so that its maximum effect matches the standard deviation of the noise.
> This ensures that $x_t^{adv}$ does not deviate unrealistically from the diffusion distribution.
> As later in the diffusion process $t \mapsto 0$, $x_t$ becomes closer to the real data distribution, so the perturbation ray cannot be wide because in this phase the model refines class-level details. At this stage, wide perturbations can disrupt generation or cause class collapse. Therefore, the ray must shrink to preserve generative fidelity.
> Our preliminary experiments confirmed this: without proper scheduling, generation quality degrades, often collapsing to class averages.

---

> ### Author Response · Authors · 2025-11-21
> **Answer to Reviewer Djvf (part 4/4)**
>
> ### (W5) Computational complexity
> > The training is ~2.5x slower than standard diffusion model training, which is however, already notoriously slow. In practice, this computational overhead may be unacceptable, especially provided with the scale of data used to train a diffusion model successfully.
>
>
> The training overhead arises from adversarial sample crafting and an additional forward pass, as discussed in Section 3.1 (Time Complexity).  In our case, the increased training time is an investment **for improved robustness and faster inference**, which is particularly relevant in real-world pipelines where inference is repeated continuously, while training is performed once for all.
> The presence of this overhead is a standard trade-off in robust training; also, when adversarially training classifiers, this trade-off affects the training procedure. Not only examples from classifiers include this overhead, but also brand new adversarial training applications on multimodal models, such as the Robust CLIP paper [1], are affected by this computational overhead.
>
> Finally, we point out that **adversarially fine-tuning** those models is also an option that helps to alleviate the problem of training cost. Examples can be found in the paper in Appendix F.2, present in the original submission.
>
> For additional details, please refer to the response to (W2, Q1) at reviewer `2wFr`.
>
> [1] Schlarmann et al. _"Robust CLIP: Unsupervised Adversarial Fine-Tuning of Vision Embeddings for Robust Large Vision-Language Models"_, ICML 2024

---

> ### Author Response · Authors · 2025-11-26
> **Additional Results for Answer to Reviewer Djvf (part 1/4)**
>
> ### Experiments for (W1, Q1) Relationship to noise-aware diffusion training
>
> > W1) The paper positions itself against noise-aware diffusion training but does not compare to them experimentally. This leaves competitiveness under “unknown corruption” unestablished.
>
> > Q1) Can you compute-matched comparisons to noise-aware diffusion training methods on CIFAR-10/CelebA/LSUN for and $\sigma \in \{ 0.05, 0.1, 0.2 \}$?
>
> For this experiment, we devised two different approaches to thoroughly evaluate the behaviour of _Daras et al. (2024a)_ under **unknown corruption**:
> 1. In the first configuration ***Conf. 1***, the method always knows the exact $\sigma$ level of the noise in the dataset but the assumption on which sample is clean  $\mathbf{x} \in \mathcal{X_\text{clean}}$ and which is noisy $\mathbf{x} \in \mathcal{X}_{\text{noisy}}$ is forced to be correct only $(1-p)$% of the time.
> 2. In the second configuration ***Conf. 2***, the assumption of knowing whether a sample is noisy or not is never considered, effectively forcing the same behavior for all the training data, when the data are noised with probability $p$.
>
> For both configurations, due to the limited time for the rebuttal, we trained the models on CIFAR-10 considering clean data and noisy data with $\sigma=\{0.1, 0.2\}$ and $p=90$%, as in our original manuscript. The results are summarized in the table below (reported in Tab.5 of the paper):
>
> **Table: Experiments on CIFAR-10 under Unknown Corruption**
>
> | $\sigma$ | $p~($\%$)$  | **Ours** | Daras et al. (2024a) *Conf. 1* | Daras et al. (2024a) *Conf. 2* |
> |----------|------|----------|-------------|-------------|
> | --       | 0    | 28.7      | **14.0**   |**14.9**    |
> | 0.1      | 0.90 | **24.7** | 94.5        |102.7       |
> | 0.2      | 0.90 | **24.8** | 109.7       |105.3       |
>
>
>
> These experiments confirm that relaxing even one of the assumptions made in the noise-aware solution proposed by _Daras et al. (2024a)_ reduces the method's robustness to unknown noise in the data, producing very high FID. This confirms the practical limitations already highlighted by _Daras et al. (2024a)_, as detailed in the response provided above.
>
> On the contrary, our method is able to work in this more challenging setting where the corruption is unknown, achieving a stable trend across different $\sigma$ and $p$, without having access to clean/noisy labels or corruption parameters. Additionally, our solution can work in an **adversarial setting**, which is something that "noise-aware" models are unable to address.
>
>
> We again thank the reviewer for the valuable suggestions, which have helped us improve the clarity and strength of our work. With the additional comparison included in the Appendix in **Section G - Comparison with noise-aware diffusion training (Table 5 and Figure 26)**, we demonstrate that our approach is valuable and overcomes the limitation of noise-aware methods in the unknown corruption setting.
>
> We hope that these additional results and our responses will positively influence the final evaluation of our paper.

---

> > ### Comment · Reviewer_Djvf · 2025-11-27
> >
> > Thanks for the response. I think the authors have addressed my concerns, for which I am willing to increase the score.

---

> ### Author Response · Authors · 2025-11-27
>
> We are grateful to the reviewer for their valuable insights and the important points raised during the rebuttal process.
> The additional results we have included have further strengthened both the paper and its contribution to the field. We are delighted that our findings successfully addressed the concerns expressed during the rebuttal period and are pleased with the reviewer's positive evaluation of our work.

---

### Author Response · Authors · 2025-12-01
**Message to the Area Chair (part 1/2)**

Dear Area Chair,

We would like to briefly summarize the contribution of our paper, along with the recognized strengths by the reviewers and the outcome of the rebuttal.

Our work was consistently recognized in first place across reviewers (`Djvf`, `2wFr`, and `PUAf`) for **its strong conceptual foundations, practical simplicity, and robust performance, and was also acknowledged for its quality, novelty, and practical relevance**. The core idea of our paper is to introduce the concept of Adversarial Training (AT) for Diffusion Models (DMs) from both theoretical and practical perspectives, and to investigate their emerging properties.

The paper grounds on the observation that the classifier-style invariance is fundamentally misaligned with diffusion’s noise-regression objective (Eq. 3). As a consequence, we **introduce a score-field equivariance alternative to effectively adapt the adversarial training framework to DMs**, proposing it as a plug-in regularization component in the diffusion loss (Eq. 6). All reviewers highlighted this contribution being a **simple, principled, and well-grounded novel conceptual perspective, emphasizing that the method cleanly integrates into standard DDPM training**, with a clear formulation and demonstration. Reviewers also agreed that the **evaluation is comprehensive and compelling, demonstrating sharp improvements in robustness under heavily corrupted data** (including 90% corrupted training data), **reduced memorization, and increased tolerance to adversarial perturbations**,  with substantial FID gains across diverse datasets such as CIFAR-10, CelebA, and LSUN. All the reviewers praised the **clarity and execution of the paper and noted that the contributions are practically relevant for enhancing the safety, reliability, and stability of diffusion models**, reflecting strong alignment on the novelty, clarity, and empirical strength of our approach.

Beyond the original version of the paper, the rebuttal period enabled us to further expand the experimental evidence and clarify some minor theoretical questions as follows:

- We assessed the generality of our method by adding to the pool of considered datasets also **ImageNet**, requested by rev. `2wFr`, where again our method shows improved performance.
- As suggested by the rev. `Djvf`, beyond the already proposed FGSM and PGD attack frameworks, we also **included a new attack setting based on the Expectation over Transformation (EoT) attack paradigm**, in order to take into account the stochasticity of the diffusion process. Even in this case, the models trained following the proposed approach result to be robust against this threat, showing less degraded performance compared to the standard Diffusion Models.
- Furthermore, learning from *generic* noisy data is one of the properties of adversarially trained Diffusion Models. **We compared our approach with a state-of-the-art training noise-aware approach** in the **unknown corruptions** setting. We show that the latter makes really strong and impractical assumptions on the applied noise model and data composition, as highlighted in the limitations of the original article. When **working in the challenging and realistic unknown noise setting, our method achieves lower and more stable FID**, effectively ignoring the noise applied to the data without relying on any assumptions, while the noise-aware method performs poorly.

---

> ### Author Response · Authors · 2025-12-01
> **Message to the Area Chair (part 2/2)**
>
> The rebuttal process clearly improved the assessments and confidence from reviewers.
>
> - Reviewer `2wFr` **(Rating 6$\rightarrow$8, updated on November 22nd, Confidence 3\)** praises the originality of the work as well as the quality and significance of the conducted experiments, defining the work as highly relevant for safe and reliable deployment of DMs. The reviewer's main suggestion was to test our method on the ImageNet dataset, which we did in the rebuttal period, providing a table in the replies highlighting the generalization capability of our approach. Moreover, it was asked to integrate some suggested papers in the related work section. We address this by discussing the referenced papers in the textual replies and adding the related references. Finally, we addressed the reviewer's final concerns about computational complexity and the choice definition of the applied adversarial noise by further clarifying these points in **Section 3.1**. The reviewer explicitly increased its score while maintaining its confidence, commenting:
>
>     > The rebuttal has alleviated my main concerns. I have therefore **revised my score upward**, finding the paper now clearly above the acceptance threshold.
>
> - Reviewer `PUAf` **(Rating 8$\rightarrow$8, confirmed on November 26th, Confidence 3\)** acknowledged our work as a novel perspective on adversarial training, redefining robustness in the context of generative diffusion processes. It also praises the simple yet well-grounded approach validated with extensive experiments across synthetic and real-world datasets. Moreover, we would also like to point out that the reviewer evaluated the paper's soundness, presentation, and contribution as excellent. The reviewer overall does not point out major weaknesses, asking instead for possible broader application of this approach to two different problems: adversarial purification and learning from data with natural corruptions. We address the first point both via textual reply and in **Section 4** and **Figure 2** of the paper; the second point is addressed textually in the rebuttal. Moreover, following the reviewer's remark, we leave as future work, as included in **Section 5**, the application of this framework to newer diffusion models. The reviewer was again convinced by our replies, keeping its score unchanged and commenting:
>
>     >I **remain convinced** of the **paper's novelty and solid contribution** to the field. Therefore, I will **maintain my initial rating**.
>
> - Reviewer `Djvf` **(Rating 4$\rightarrow$6, updated on early November 27th, Confidence 3\)** praised the approach and its simple and clear recipe, acknowledging also the sharp improvements of our approach. The reviewer was initially concerned about the comparisons with the existing noise-aware approaches in the **unknown corruptions** setting. We addressed this point in the rebuttal period by adding supplementary experiments showing that our method performs best in this context with respect to the noise-informed approaches. The experiments have been included in **Appendix G**. Moreover, the reviewer proposed to include an EoT attack to test the robustness of the model. We formulated the attack (**algorithm in A.4**), and the adversarially trained models resulted in being also robust against this additional challenging setting, as discussed in **Section 3.4** and shown in the new table in **Figure 8(b)**. Finally, the reviewer asked for further clarifications about some theoretical insights of the proposed method and about the computational overhead, which we replied to textually by adding both rebuttal replies and further analysis in **Appendix F**. The reviewer explicitly increased its score while maintaining its confidence, commenting:
>
>     > The authors **have addressed my concerns**, for which I am **willing to increase the score**.
>
> The rebuttal addressed all the reviewer’s main concerns and we believe the outcome demonstrates the work’s solidity, novelty, and impact, along with the improvements made during the rebuttal.
> The interaction has been conducted in a sound, ethical and constructive manner, and all evidence from the discussions has led to real and concrete changes in the revised paper. We are pleased that the quality of our work has improved as a result of our interaction with the reviewers.
>
> We appreciate your consideration and remain available for any clarification.

---

### Meta-Review · Area_Chair_kjCr · 2026-01-07

**Summary:**

This paper mainly introduces the concept of Adversarial Training (AT) for Diffusion Models (DMs) from both theoretical and practical perspectives, and investigates their emerging properties. The paper grounds on the observation that the classifier-style invariance is fundamentally misaligned with diffusion’s noise-regression objective (Eq. 3). As a consequence, the authors introduce a score-field equivariance alternative to effectively adapt the adversarial training framework to DMs, proposing it as a plug-in regularization component in the diffusion loss (Eq. 6). Basic evaluation is made in the original submission.

**Reviewer Concerns:**

Overall, this paper is well-written and has clear theoretical and empirical contributions. The reviewers' major concerns are 1) large-scale dataset experiments and 2) additional computational overhead.

For the first one, the authors provided results from ImageNet, and reviewers appreciate the results provided. For the second one, the authors acknowledge this drawback, yet provide a more advanced version to avoid this issue in the appendix.

Thus, the main concerns from reviews are well-addressed. For minor concerns, the authors also provide detailed responses, which is satisfying.

**Reviewer Scores:**

Because all concerns are addressed well, this paper has consistent support from reviewers (with confirmed responses). In the end, this paper's score comes to 8,8,6, which is a clear acceptance to me.

---

### Decision · Program_Chairs · 2026-01-26

Accept (Poster)